

# A 250 plastome phylogeny of the grass family (Poaceae): topological support under different data partitions

Jeffery M. Saarela[1], Sean V. Burke[2], William P. Wysocki[3], Matthew D. Barrett[4,5], Lynn G. Clark[6], Joseph M. Craine[7], Paul M. Peterson[8], Robert J. Soreng[8], Maria S. Vorontsova[9] and Melvin R. Duvall[2]

[1] Beaty Centre for Species Discovery and Botany Section, Canadian Museum of Nature, Ottawa, ON, Canada
[2] Plant Molecular and Bioinformatics Center, Biological Sciences, Northern Illinois University, DeKalb, IL, USA
[3] Center for Data Intensive Sciences, University of Chicago, Chicago, IL, USA
[4] Botanic Gardens and Parks Authority, Kings Park and Botanic Garden, West Perth, WA, Australia
[5] School of Biological Sciences, The University of Western Australia, Crawley, WA, Australia
[6] Department of Ecology, Evolution and Organismal Biology, Iowa State University, Ames, IA, USA
[7] Jonah Ventures, Manhattan, KS, USA
[8] Department of Botany, National Museum of Natural History, Smithsonian Institution, Washington, DC, USA
[9] Comparative Plant & Fungal Biology, Royal Botanic Gardens, Kew, Richmond, Surrey, UK

Corresponding authors
Jeffery M. Saarela,
jsaarela@mus-nature.ca

Melvin R. Duvall,
mel-duvall@niu.edu

## ABSTRACT

The systematics of grasses has advanced through applications of plastome phylogenomics, although studies have been largely limited to subfamilies or other subgroups of Poaceae. Here we present a plastome phylogenomic analysis of 250 complete plastomes (179 genera) sampled from 44 of the 52 tribes of Poaceae. Plastome sequences were determined from high throughput sequencing libraries and the assemblies represent over 28.7 Mbases of sequence data. Phylogenetic signal was characterized in 14 partitions, including (1) complete plastomes; (2) protein coding regions; (3) noncoding regions; and (4) three loci commonly used in single and multi-gene studies of grasses. Each of the four main partitions was further refined, alternatively including or excluding positively selected codons and also the gaps introduced by the alignment. All 76 protein coding plastome loci were found to be predominantly under purifying selection, but specific codons were found to be under positive selection in 65 loci. The loci that have been widely used in multi-gene phylogenetic studies had among the highest proportions of positively selected codons, suggesting caution in the interpretation of these earlier results. Plastome phylogenomic analyses confirmed the backbone topology for Poaceae with maximum bootstrap support (BP). Among the 14 analyses, 82 clades out of 309 resolved were maximally supported in all trees. Analyses of newly sequenced plastomes were in agreement with current classifications. Five of seven partitions in which alignment gaps were removed retrieved Panicoideae as sister to the remaining PACMAD subfamilies. Alternative topologies were recovered in trees from partitions that included alignment gaps. This suggests that ambiguities in aligning these uncertain regions might introduce a false signal. Resolution of these and other

critical branch points in the phylogeny of Poaceae will help to better understand the selective forces that drove the radiation of the BOP and PACMAD clades comprising more than 99.9% of grass diversity.

## INTRODUCTION

Grasses (Poaceae) are the fifth largest family of flowering plants in the world, with some 11,500 species and about 768 genera (*Soreng et al., 2017*), and the family is economically important because it includes wheat (*Triticum* L.), rice (*Oryza* L.) and corn (*Zea* L.), as well as numerous forage, bamboo and biofuel species. Grasses grow on all continents in tropical, temperate and Arctic zones. Grasses are common and often dominant components of open ecosystems (prairies, pampas, steppes, veldts), but also occur in association with forests, and they have diversified to inhabit aquatic to desert habitats. The deep phylogenetic framework for Poaceae is well established (*Grass Phylogeny Working Group, 2001*; *Duvall et al., 2007*; *Bouchenak-Khelladi et al., 2008*; *Saarela & Graham, 2010*; *Grass Phylogeny Working Group II, 2012*). Three small, deeply diverging subfamilies (Anomochlooideae, Pharoideae, Puelioideae) are the successive sister groups of a major clade comprising two lineages, the BOP and PACMAD clades. The BOP clade consists of three subfamilies: Bambusoideae (bamboos or bambusoids), Oryzoideae (rices or oryzoids) and Pooideae (cool season or pooids). The PACMAD clade consists of six subfamilies: Panicoideae (panicoids), Arundinoideae (arundinoids), Chloridoideae (chloridoids), Micrairoideae (micrairoids), Aristidoideae (aristidoids) and Danthonioideae (danthonioids).

The systematics of grasses have been studied throughout the history of botany, with the first dedicated classification by *Brown (1814)*, and classifications of the whole family or parts thereof have been continually proposed through time as new information has accumulated. The current subfamily classification was proposed by *Grass Phylogeny Working Group (2001)* and modified by *Sánchez-Ken et al. (2007)* and *Sánchez-Ken & Clark (2010)*. The phylogenetic structure of the family was synthesised and reconciled with morphological and developmental genetic data by *Kellogg (2015)*. A recent worldwide classification of grasses based explicitly on molecular phylogenetic evidence recognized 12 subfamilies, six supertribes, 52 tribes and 90 subtribes of grasses, with 21 genera unplaced to tribe and 39 unplaced to subtribe (*Soreng et al., 2017*); this is an update of a classification proposed two years earlier by *Soreng et al. (2015b)*.

Complete plastid genomes from across the land plant tree of life are being sequenced rapidly (*Tonti-Filippini et al., 2017*), including those from grasses. *Saarela et al. (2015)* summarized all publications of grass plastomes published as of September 2014, and many new plastomes have since become available. Recent grass plastome sequences have been variously published in short contributions (*Myszczyński et al., 2015*; *Wang & Gao,*

*2015, 2016*; *Lu et al., 2016*; *Perumal et al., 2016*) or in the context of detailed phylogenomic analyses of different grass lineages, including the PACMAD clade (*Cotton et al., 2015*; *Teisher et al., 2017*), Bambusoideae (*Wu et al., 2015*; *Wysocki et al., 2015*; *Attigala et al., 2016*; *Vieira et al., 2016*; *Zhang & Chen, 2016*), Aristidoideae (*Besnard et al., 2014*), Brachypodieae (*Sancho et al., 2017*), early diverging grasses (*Burke et al., 2016a*), Panicoideae (*Burke et al., 2016b*), Chloridoideae (*Duvall et al., 2016*), *Zea* (*Orton et al., 2017*), Micrairoideae (*Duvall et al., 2017*), Pooideae (*Saarela et al., 2015*) and Oryzeae (*Kim et al., 2015*; *Liu et al., 2016*; *Wu & Ge, 2016*; *Zhang et al., 2016a*, *2016b*). Phylogenomic analyses of plastomes have contributed increased resolution and support for many relationships within and among grass subfamilies compared with earlier single- and multi-gene plastid studies. However, the large number of publicly available grass plastome sequences have not been combined in a single study.

One issue of plastome phylogenomics relates to data partitioning. A few plastome phylogenomic studies of grasses have investigated different ways to partition complete plastome sequences, primarily by comparing analyses of coding, noncoding, and coding plus noncoding regions. These studies failed to find unambiguous evidence to support any specific partitioning strategy, but this may reflect insufficient taxonomic sampling (*Zhang, Ma & Li, 2011*; *Burke, Grennan & Duvall, 2012*; *Ma et al., 2014*; *Saarela et al., 2015*). Gapped sites in an alignment are another partition that can be analyzed, and that may be problematic for phylogenetic inference (*Warnow, 2012*). The effects of including vs. excluding gapped sites on phylogenetic reconstruction has not been fully explored in grass phylogenomic analyses. Some gapped sites in an alignment reflect evolutionary history (e.g., indels), but others may represent suboptimal alignment possibly resulting in spurious phylogenetic results. Indeed, as plastome-scale data sets rapidly increase in size, manual curation of alignments is increasingly difficult, especially as phylogenetic breadth of taxon sampling increases. Generating alignments automatically for these large data sets is practical, but such alignments are likely to include some ambiguously aligned regions, particularly among the most rapidly-evolving parts of the plastome for which it is necessary to introduce alignment gaps.

Yet another way to partition plastomes for phylogenetic reconstruction is through comparison of positively selected vs. nonselected nucleotide sites. The plastome is a mosaic of selected and nonselected nucleotide sites. Sites under selection are found in polycistronic protein and RNA coding regions together with their associated promoter and intron-processing regions. These regions constitute approximately 45% of the grass plastome, although many third codon positions in protein coding loci are unconstrained by selection. Positive selection can bias reconstruction of phylogenetic relationships (*Christin et al., 2012*). In grasses, positive selection has been characterized in *rbcL* (*Christin et al., 2008b*) and in all plastome protein genes across the PACMAD clade (*Piot et al., 2018*). The latter study identified positive selection in 25 of 76 plastid genes, and found that the multiple origins of $C_4$ photosynthesis in the PACMAD clade were accompanied by positive selection in *rbcL* but not in other plastid genes. Potential bias in plastome phylogenetic reconstruction when positively selected sites are included in analysis has not been explored previously in grasses.

Here, we report a phylogenomic study of 250 plastomes broadly representing the subfamilies, families, tribes and subtribes of grasses, including 15 new plastomes. We characterize phylogenetic signal in 14 plastome partitions, including a three-gene partition, coding regions, noncoding regions, and complete plastomes, each variously including or excluding gapped sites and positively selected sites in the alignment. With these analyses we test the hypotheses that selection, alignment ambiguities, or other characteristics of specific partitions contribute to difficulties in retrieving consistent topologies in grass phylogenies when taxonomic sampling is consistent. In particular, the problematic deep relationships among the PACMAD subfamilies are investigated (i.e., what is the branching order among Aristidoideae, Panicoideae, Chloridoideae + Danthonioideae, and Arundinoideae + Micrairoideae). We also compare and contrast the results of our phylogenomic analyses with current grass classifications and with existing knowledge of phylogenetic relationships in grasses derived from plastid/plastome and nuclear analyses.

## MATERIALS AND METHODS

### Taxon sampling

Taxon sampling includes 15 new (Table 1) and 235 plastomes obtained from GenBank (*Asano et al., 2004*; *Saski et al., 2007*; *Bortiri et al., 2008*; *Diekmann et al., 2009*; *Leseberg & Duvall, 2009*; *Wu et al., 2009*; *Wu & Ge, 2012*; *Morris & Duvall, 2010*; *Nock et al., 2011*; *Young et al., 2011*; *Zhang, Ma & Li, 2011*; *Zhang et al., 2016b*; *Burke, Grennan & Duvall, 2012*; *Burke et al., 2014*, *2016a*, *2016b*; *Waters et al., 2012*; *Besnard et al., 2013*, *2014*; *Hand et al., 2013*; *Gornicki et al., 2014*; *Jones, Burke & Duvall, 2014*; *Lee et al., 2014*; *Ma et al., 2014*, *2015*; *Mariac et al., 2014*; *Middleton et al., 2014*; *Ye et al., 2014*; *Cotton et al., 2015*; *Gogniashvili et al., 2015*; *Kim et al., 2015*; *Lundgren et al., 2015*; *Rousseau-Gueutin et al., 2015*; *Saarela et al., 2015*; *Wambugu et al., 2015*; *Wang & Gao, 2015*; *Wysocki et al., 2015*; *Attigala et al., 2016*; *Duvall et al., 2016*, *2017*; *Gao & Gao, 2016*; *Gao, Li & Gao, 2016*; *Nah et al., 2016*) (Table S1). Plastomes not available on GenBank as of 19 November 2015 were not included in our analyses, unless generated by us. The voucher specimen for the plastome published as *Microstegium vimineum* (Trin.) A. Camus (GRIN, PI 659331) in *Burke et al. (2016b)* was mis-identified; its correct identity is *Arthraxon prionodes* (Steud.) Dandy (E. Kellogg, 2016, personal communication). The same seed accession was sequenced in *Estep et al. (2014)*, as *A. prionodes*, and in *Liu et al. (2014)*, as *M. vimineum*. We have corrected this error in GenBank (accession KU291471). For new plastomes, we obtained DNA from either fragments of herbarium mounted leaf tissues or silica-dried seedlings germinated in the greenhouse. For the latter, voucher specimens were prepared from greenhouse material when it reached a flowering stage suitable for identification, and these were deposited in the herbarium (DEK) of the Biological Sciences Department, Northern Illinois University, DeKalb, Illinois. Tissue was homogenized manually in liquid nitrogen before extraction. The DNA extraction protocol was followed using the Qiagen DNeasy Plant Mini Kit (Qiagen Inc., Valencia, CA, USA).

We follow the subdivisional classification of Poaceae by *Soreng et al. (2017)* with one exception: following the results of *Burke et al. (2016b)*, we consider *Whiteochloa* C.E.

**Table 1 Vouchers (standard herbarium codes are used) or seed sources of plant material for 15 new plastomes, GenBank accession numbers, and other sequencing and assembly details.**

| Species | GenBank accession number | Voucher/seed source | Single end library prep. method | Sequence mode | Assembler | Annotation reference |
|---|---|---|---|---|---|---|
| *Agrostis gigantea* Roth | MF460976 | PI 619538 | Nextera | High output | Iterative velvet | NC_027468 |
| *Alopecurus arundinaceus* Poir. | MF460977 | PI 380664 | Nextera | High output | Iterative velvet | NC_027468 |
| *Aristida ternipes* Cav. | MF460978 | MSB 98474 | Nextera XT | Rapid mode | SPAdes | NC_025228 |
| *Connorochloa tenuis* (Buchanan) Barkworth, S.W.L. Jacobs & H.Q. Zhang | MF460979 | PI 531685 | Nextera XT | Rapid mode | SPAdes | NC_021761 |
| *Drepanostachyum falcatum* (Nees) Keng f. | MF460981 | *L. Clark 1756* (ISC) | Nextera | High output | Iterative velvet | NC_024725 |
| *Lamarckia aurea* (L.) Moench | MF460982 | PI 378959 | Nextera XT | Rapid mode | SPAdes | NC_0274373 |
| *Leptochloa pluriflora* (E. Fourn.) P.M. Peterson & N. Snow | MF460983 | PI 337598 | Nextera XT | Rapid mode | SPAdes | NC_027650 |
| *Oxychloris scariosa* (F. Muell.) Lazarides | MF460971 | PI 238262 | Nextera XT | Rapid mode | SPAdes | NC_024262 |
| *Prosphytochloa prehensilis* (Nees) Schweick. | MF460972 | *G. Guala 1689* (ISC) | Nextera | High output | Iterative velvet | NC_026967 |
| *Rytidosperma pallidum* (R. Br.) A.M. Humphreys & H.P. Linder | MF460980 | *P. Linder 5664* (BOL) | Nextera | High output | Iterative velvet | NC_025232 |
| *Stipagrostis uniplumis* (Licht. ex Roem. & Schult.) De Winter var. *uniplumis* | MF460973 | *N. Snow & M. Chatukuta 6853* (MO) | Nextera | High output | Iterative velvet | NC_025228 |
| *Taeniatherum caput-medusae* (L.) Nevski | MF460974 | PI 561092 | Nextera XT | Rapid mode | SPAdes | NC_009950 |
| *Triodia stipoides* (S.W.L. Jacobs) Crisp & Mant | MF460970 | *Barrett 3523* (PERTH) | Nextera | High output | Iterative velvet | NC_024262 |
| *Triodia wiseana* C.A. Gardner | MF460975 | *Peterson et al. 14384* (US) | Nextera | High output | Iterative velvet | NC_024262 |
| *Zingeria biebersteiniana* (Claus) P.A. Smirn. | MF460984 | W6 19209 | Nextera XT | Rapid mode | SPAdes | NC_009950 |

**Note:**
PI, Plant Introduction number; W6, West Regional PI group; U.S. National Plant Germplasm System (https://npgsweb.ars-grin.gov/); MSB, Millenium Seed Bank, Kew (http://apps.kew.org/seedlist/SeedlistServlet).

Hubb. as part of Panicinae rather than Cenchrinae. Generic classification also follows *Soreng et al. (2017)*, including for plastomes published under different names, as noted in Table S1. Major lineages in the bamboo tribe Arundinarieae have been referred to as clades I–XII (*Triplett & Clark, 2010*; *Zeng et al., 2010*; *Yang et al., 2013*; *Attigala et al., 2014*; *Zhang et al., 2016c*), and we here follow this informal naming system. The plastome sampling represents all 12 subfamilies, 44 tribes, 63 subtribes, 179 genera and 250 species of grasses.

## Plastome sequencing, assembly, annotation and alignment

Plastome sequencing methods generally followed those of *Burke et al. (2016b)*. Single end libraries were prepared depending on different starting quantities of DNA with the

Nextera or Nextera XT methods (Illumina, San Diego, CA, USA). All sequencing was done on the Illumina HiSeq 2500 platform at the core DNA Facility, Iowa State University, Ames, IA, USA. Details are given (Table 1). Illumina reads were assembled into complete plastid chromosomes with de novo assembly methods. For Nextera data the Velvet software package (*Zerbino & Birney, 2008*) was used iteratively, loading contigs from the previous step into the assembler multiple times with stepwise increasing kmer lengths (see details in *Wysocki et al. (2014)*). For Nextera XT data, SPAdes v.3.5.0 (http://bioinf.spbau.ru/spades) was used for de novo assembly (*Bankevich et al., 2012*). Contigs were scaffolded with the anchored conserved region extension method (*Wysocki et al., 2014*), which queries contig sets for regions that are invariant across Poaceae. Any remaining gaps in the scaffolds were resolved by in silico genome walking. A final verification was performed by mapping reads to the newly assembled plastome to detect and correct inconsistencies. Annotations were determined in Geneious Pro v9.1.6 (Biomatters Ltd., Auckland, New Zealand) (*Kearse et al., 2012*) initially using the pairwise align function. A published reference for each new plastome was obtained from a closely related grass species. Annotations from the reference were transferred to the newly assembled plastome. The boundaries of coding sequences (CDSs) were adjusted to preserve reading frames. The endpoints of the large inverted repeats (IR) were located using the methods of *Burke, Grennan & Duvall (2012)*. Data partitions were aligned in Geneious Pro with the MAFFT v7.017 (*Katoh & Standley, 2013*) plugin using the auto function for the algorithm and other settings as defaults. In the interest of reproducibility no manual adjustments were made to the alignments. This approach may have discarded informative microstructural mutations or possibly allowed regions with micro inversions to be misaligned. However, in the context of whole plastome alignments, these events are overwhelmed by unambiguously aligned coding and noncoding regions and so have minimal effect.

## Purifying/positive selection

Purifying/positive selection detection and the removal of positively selected sites were as follows: alignments for each of the 76 protein CDSs, excluding duplicated copies of loci in one of the IRs, for the 250 taxa were individually extracted. Each extracted CDS was imported into Mega6 v6.06 (*Tamura et al., 2013*) and aligned by codon via Muscle (*Edgar, 2004*). The computed overall mean distance function determined the mean nonsynonymous substitutions per non-synonymous site (dN) and the mean synonymous substitutions per synonymous site (dS) values for each CDS based on default parameters. Each CDS was then tested for positive or purifying selection using the codon-based $Z$-test of selection (*Nei & Kumar, 2000*) on default parameters (Dataset S1).

After the predominant type of selection for each locus was identified, the specific number of positively selected codons in each locus was determined using the following methods. Each extracted CDS was then first analyzed with the codon alignment tool in HyPhy v2.22016030beta (MP) (*Pond & Frost, 2005*), under the "Data File Tools" (option 4), under the following encoded conditions: 1 (CDS aligned to reference using protein translations), 1 (default BLAST BLOSUM62 matrix), 1 (prefix and suffix indels were not penalized), 2 (the longest sequence in the data file was used as the reference), and

**Table 2 Descriptions of the plastome partitions analyzed.**

| Code | Data partition | Gapped sites stripped | Positively selected sites removed | Matrix length |
|------|----------------|------------------------|-----------------------------------|---------------|
| A | *rbcL, ndhF, matK* | – | – | 7,195 |
| B | *rbcL, ndhF, matK* | + | – | 3,476 |
| C | *rbcL, ndhF, matK* | – | + | 6,722 |
| D | *rbcL, ndhF, matK* | + | + | 5,077 |
| E | Plastome coding | – | – | 59,299 |
| F | Plastome coding | + | – | 46,707 |
| G | Plastome coding | – | + | 55,851 |
| H | Plastome coding | + | + | 44,975 |
| Q | Plastome noncoding | – | n/a | 143,401 |
| R | Plastome noncoding | + | n/a | 26,307 |
| W | Complete plastome | – | – | 197,529 |
| X | Complete plastome | + | – | 78,714 |
| Y | Complete plastome | – | + | 197,332 |
| Z | Complete plastome | + | + | 71,140 |

**Note:**
Partitions variously included a subset of coding regions, all coding regions, noncoding regions and the complete plastome, including coding and noncoding partitions. In the alignments of each of these four core partitions we variously included or excluded gapped sites and positively selected sites in partitions including coding regions.

1 (there were no reference coordinate sequences; i.e., no external standard). Each alignment was then analyzed with "Data File Tools" (option 4) following the options of: 6 (convert sequence names to HyPhy valid identifiers if needed and replace stop codons with gaps in codon data if any are present), and 1 (universal code), to remove stop codons and convert sequence names just for the following analysis. Each CDS alignment was then tested for positive selection at individual codon sites using mixed effects model of evolution (MEME) (*Murrell et al., 2012*). This was done through the "Positive Selection" option in HyPhy following the options of 9 (quickly test for positive selection using several approaches), 1 (universal code), 1 (new analysis), 1 (default, use HKY85 and MG94xHKY85), 1 (neutral dN/dS = 1), and 11 (MEME to search for evidence of episodic selection at individual sites). Based on this output, for each data partition that had positively selected sites removed, these sites were manually removed for each CDS.

## Phylogeny estimation

We analysed 14 different data partitions (Table 2), which variously included a subset of plastome coding regions (partitions A–D), all plastome coding regions (E–H), plastome noncoding regions (Q–R) and complete plastomes (coding and noncoding partitions) (W–Z). The subset of coding regions (*rbcL, ndhF, matK*, and the *trnK* intron) was selected to compare against previously published family wide analyses such as *Grass Phylogeny Working Group II (2012)*. In each of these four core partitions we variously included or excluded gapped sites in the alignment and positively selected sites in partitions including coding regions when applicable, so that each core partition had four variants except the noncoding partition, from which there were no positively selected sites to exclude.

Alignment gaps found in one or more sequences were removed using the "remove gaps" command in Geneious. Alignment files are presented in Datasets S2–S15. We refer to the four trees based on the reduced datasets as "three-gene trees," and to the ten trees based on large partitions of whole plastomes as "plastome trees."

We conducted only maximum likelihood (ML) analyses, given the extensive partitioning that was conducted and the large number of comparisons that needed to be made among the many clades in trees from different partitions. Since our study is focused on plastome phylogeny of the entire grass family, it would be ideal to root the tree with non-grass taxa. The closest relatives of grasses are the families Joinvilleaceae, Ecdeiocolaceae and Flagellariaceae. Plastome data based on 77 coding regions and some smaller data sets identify Ecdeiocolaceae and grasses as sister taxa (*Bremer, 2002*; *Chase et al., 2006*; *Graham et al., 2006*; *Givnish et al., 2010*; *Barrett et al., 2016*), while other studies have identified a clade of Ecdeiocolaceae + Joinvilleaceae as the grass sister group (*Marchant & Briggs, 2007*; *Christin et al., 2008a*; *Saarela & Graham, 2010*). Of these three nongrass graminid families, complete plastome data are available only for Joinvilleaceae (*Wysocki et al., 2016a*). Moreover, there are major rearrangements in the large single copy (LSC) regions of grass plastomes compared to other Poales and all other angiosperms (*Doyle et al., 1992*; *Katayama & Ogihara, 1996*; *Michelangeli, Davis & Stevenson, 2003*; *Burke et al., 2016a*; *Wysocki et al., 2016a*). In a recent plastome phylogenomic study of grasses, including a highly rearranged Poales outgroup (*Joinvillea* Gaudich. ex Brongn. and Gris) resulted in the loss of substantial data, especially in noncoding regions (*Burke et al., 2016a*). Using *Joinvillea* as an outgroup in the current study would similarly result in a loss of data, which would be detrimental to our partitioned analyses. Thus, considering the difficulty of aligning grasses with nongrasses, and consistent with the purpose of the present analysis, we opted to root our trees along the branch leading to Anomochlooideae (*Anomochloa marantoidea* Brong. and *Streptochaeta spicata* Schrad. ex Nees), given the well-established sister group relationship of Anomochlooideae to the rest of the grasses

A substitution model was selected using jModelTest v. 2.1.3 (*Guindon & Gascuel, 2003*; *Darriba et al., 2012*). For all partitions, the GTR + I + G model was selected under the Akaike information criterion (*Akaike, 1974*). ML analyses were performed using RAxML-HPC2 on XSEDE v. 8.1.11 (*Stamatakis, 2014*) at the CIPRES Science Gateway (*Miller, Pfeiffer & Schwartz, 2010*). The number of bootstrap replicates was automatically halted by the "autoMRE" function. All model parameters were estimated.

To compare and contrast the phylogenetic trees generated from each of the 14 partitions, we compared branch support for identical (i.e., shared) clades across all trees (Dataset S16). First, we chose a reference tree, named all clades on that tree by assigning a unique number to each branch subtending a clade, and then recorded, in a spreadsheet, BP for each clade when ≥50%. Second, we recorded BP ≥50% for the same clades in all other trees. When we encountered a clade in a non-reference tree not present in the reference tree, we named that clade and recorded BP for it, when ≥50%, across all trees. We chose the tree resulting from analysis of partition X as our reference tree, for the following reasons: (1) a priori partition exclusion is minimized in X (i.e., it includes coding and noncoding regions), so we are not arbitrarily choosing one of those two major
partitions for our reference tree; (2) stripping gapped sites is necessary since accurate ML estimation cannot be guaranteed, even with long sequences, when indels are included and treated as missing data (*Warnow, 2012*); (3) erring on the side of including more data, as found in complete plastomes, keeps our large analysis from running into an insufficient phylogenetic information wall. We present the ML reference tree (X) in multiple figures, including a summary tree showing relationships among subfamilies, with within-subfamily sampling collapsed to a single branch.

We performed six pairwise comparisons of trees generated from data partitions differing only by the inclusion or exclusion of positively selected codons, to gain insight into the relative effect of these characters on the inferred phylogenies. We conducted similar comparisons of trees differing only by the inclusion or exclusion of gapped sites. To characterize the effect of datasets comprising three-genes and all plastome coding genes on tree topology and support we conducted pairwise comparisons of trees A vs. E, B vs. F, C vs. G and D vs. H.

Descriptions of phylogenetic results focus on the plastome trees (not the three-gene trees), and clades with support <50% are not discussed. We use the terms "weak or poor," "moderate," and "strong" in reference to clades that received BP values of 50–70%, 71–90% and 91–100%, respectively, and "unsupported" for clades with BP <50%.

## RESULTS

### Purifying/positive selection

Selection analyses of 76 protein coding loci in the grass plastome indicated that all were under selection. The predominate selective force identified for all loci was purifying selection (all 76 significant at $p < 0.05$). However, positively selected codons could be identified in 65 out of the 76 loci (Fig. 1). Loci without positively selected codons included two subunits of the cytochrome b6/f complex (*petL*, *petN*), seven photosystem II proteins (*psbE*, *psbF*, *psbI*, *psbL*, *psbN*, *psbT*, *psbZ*) and two ribosomal proteins (*rpL36*, *rps19*). All 11 were relatively short loci (mean length <150 bases). Of the 65 loci with positively selected codons, the greatest number were found in *rpoC2* (184 codons), *matK* (74), *ndhF* (67) and *rbcL* (47).

### Comparison of phylogenetic trees from partitioned datasets

The entire 250 plastome alignment, including gapped sites, is 197,529 bp in length, and there is considerable variation in the lengths of the plastome partitions considered (Table 2). The plastome coding, plastome noncoding and complete plastome alignments including gapped sites are 24–32%, 445% and 151–177% longer, respectively, than when gapped sites are excluded. The plastome coding and complete plastome alignments including positively selected sites are 5.8% and 0.1% greater in length than when positively selected sites are excluded.

Maximum likelihood cladograms and phylograms of all 14 analyses are provided in Fig. S1. Mean BP across all branches with support ≥50% in each of the 14 trees ranges from 90.1% to 97.9%. Support is generally lower in the three-gene trees than in the plastome trees (Fig. 2). Among the four three-gene trees, mean support is highest in tree

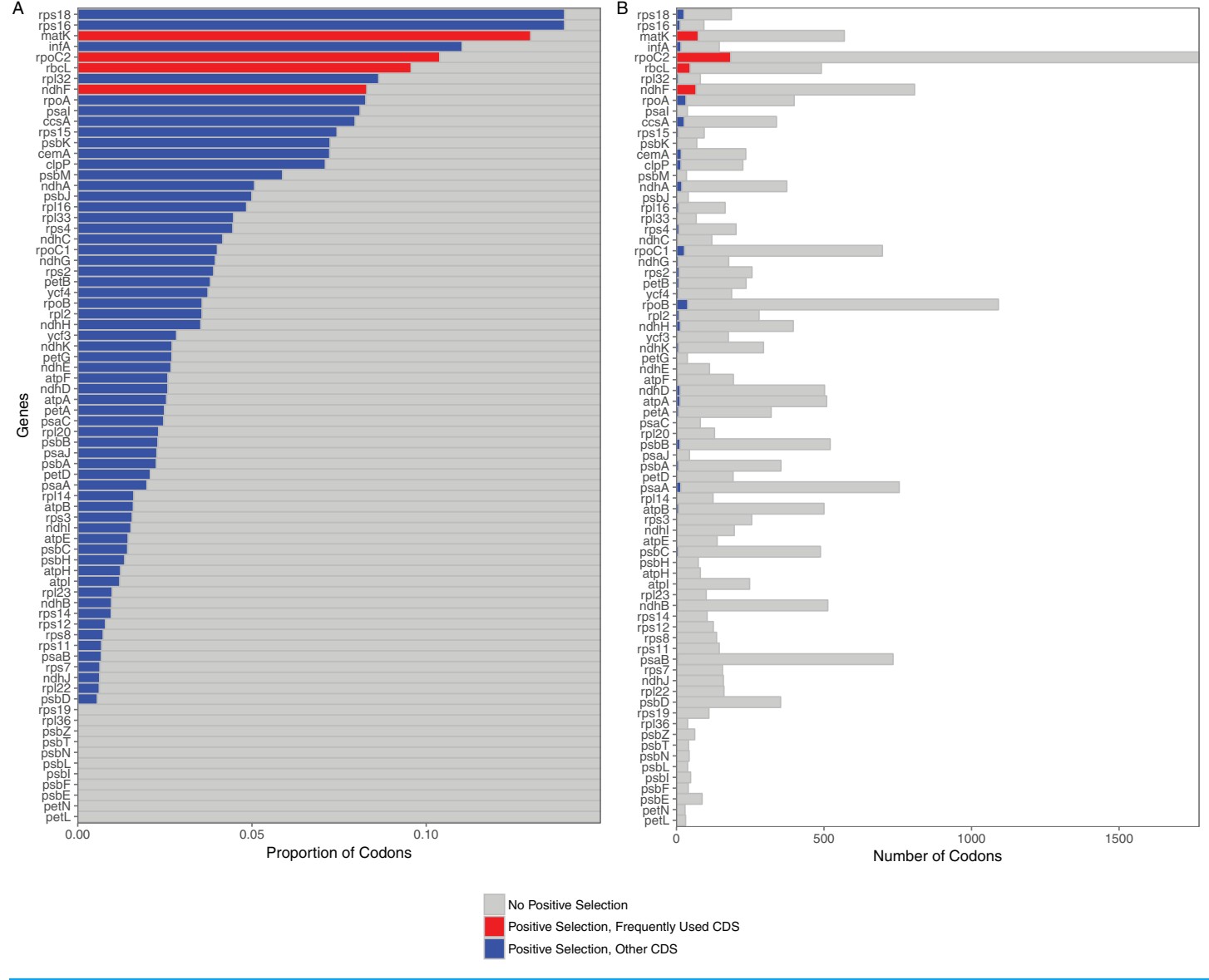

**Figure 1 Proportion and raw numbers of positively selected codons for each plastome protein coding sequence (CDS).** (A) Proportion of codons. (B) Number of codons. Red represents positive selection in CDS that are commonly used in phylogenetic studies, while blue represents positive selection in other CDS. Proportionate data are only represented up to 0.15 selected codons for clarity of illustration.

A, and that support is significantly different from mean support in trees B–D (Fig. 2). Among the 10 plastome trees, support is generally lower in tree R compared to the others, but mean support in tree R is not significantly different than that in trees E–H. Mean branch support is highest in trees Q and W–Z, but is not significantly different among these trees (Fig. 2).

The reference tree X (Figs. 3–8) includes 242 clades with BP ≥50%, and the 14 trees include 309 clades with support ≥50% in at least one tree (Dataset S16). Eighty-five clades are maximally supported in all 14 trees and 144 in all plastome trees, 231 clades are
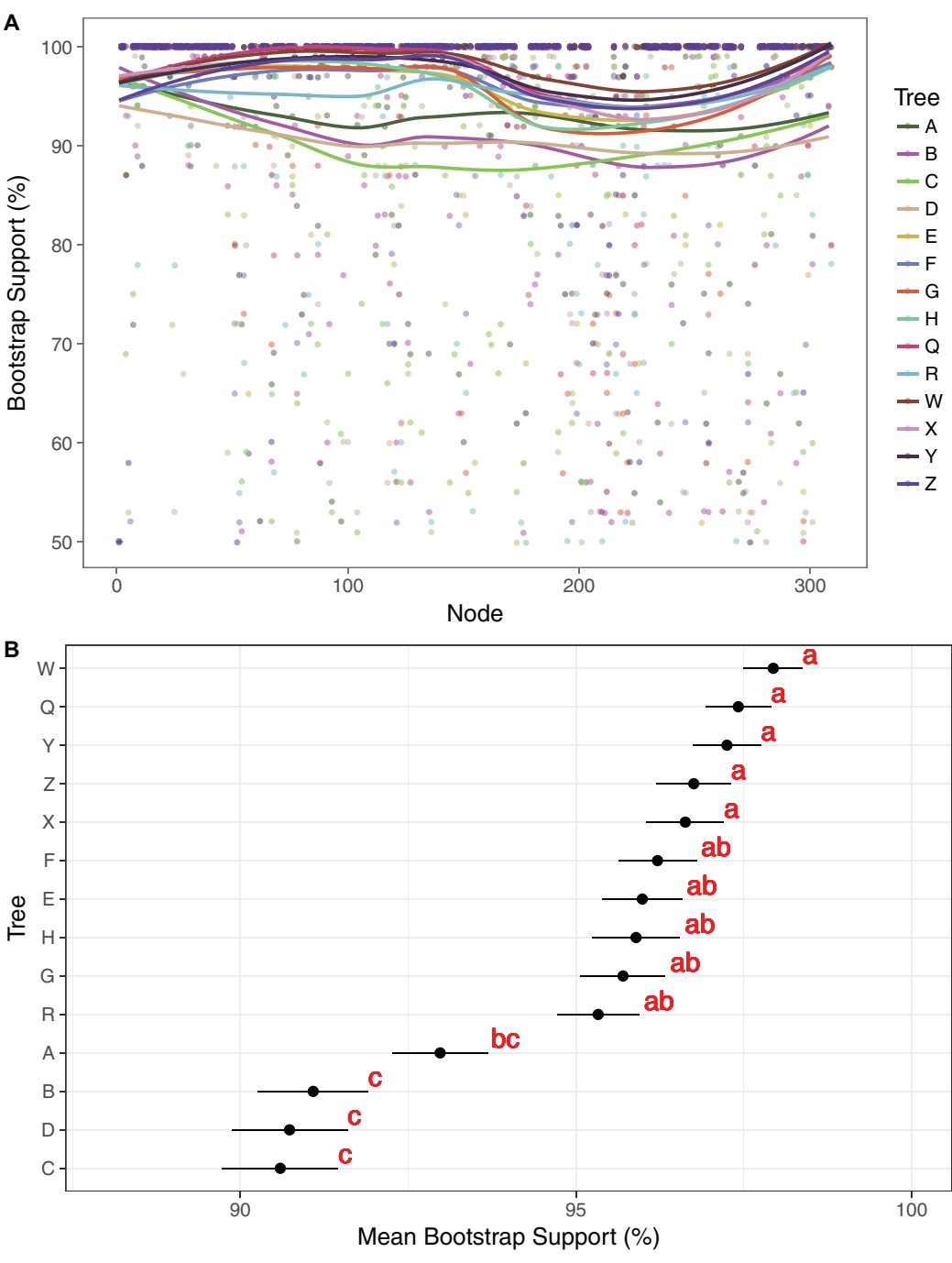

**Figure 2 Comparison of bootstrap support across trees.** (A) Support values and trend lines for all clades identified in at least one tree with support ≥50%. Trend lines were plotted in R, using the command geom_smooth in ggplot2 with the method "LOESS." Clade no. corresponds to numerical clade identifiers as noted in Dataset S16. (B) Mean support values with standard error bars for each tree. Trees were grouped by a least significant difference test with a Bonferonni correction. Groups are identified by the analysis and labeled with one or two letter designators showing overlap in some cases.

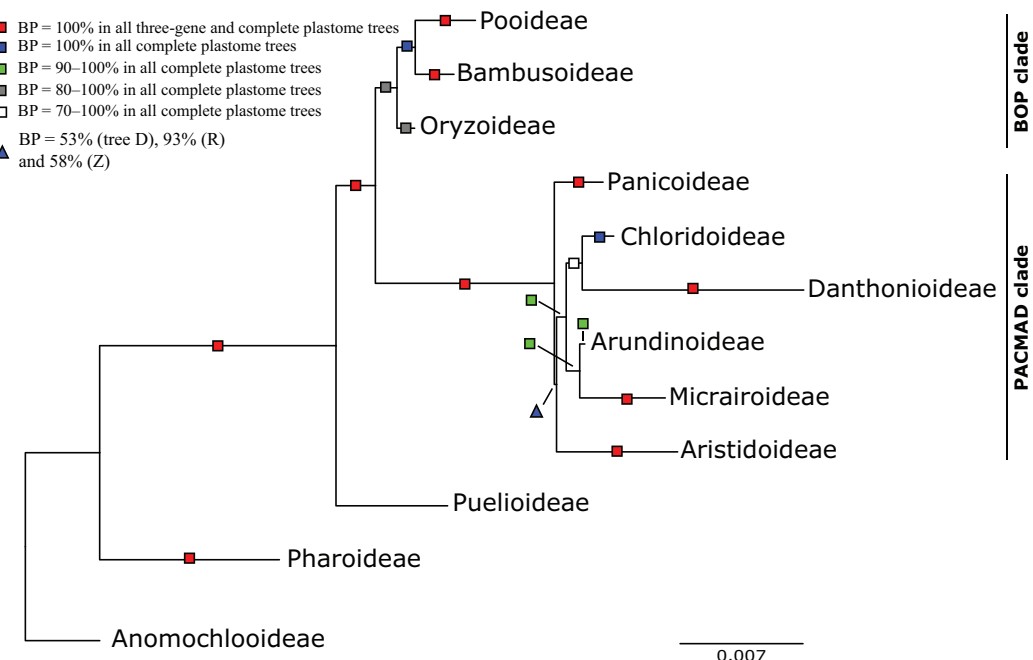

**Figure 3 Maximum likelihood phylogram from analysis of complete plastomes excluding gapped sites and including positively selected sites (tree X) showing relationships among major lineages of Poaceae.** Subfamilies are collapsed and only the branch subtending each subfamilial clade is shown. Bootstrap support is indicated along branches, according to the legend on the upper left. Two alternative topologies within the PACMAD clade, not shown in the figure, are identified in our trees: (1) Aristidoideae are sister to the rest of the PACMAD clade, with BP = 69–100%, in trees A, C, Q, W and Y; and (2) Aristidoideae and Panicoideae form a weakly supported clade (BP = 52%) in tree F that is sister to the rest of the PACMAD clade.

maximally supported in at least one plastome tree, and 78 clades are not maximally supported in any tree. There are 197 clades with support ≥50% in at least one plastome tree and 24 clades with support ≥50% in only single plastome trees. The number of clades in each tree with support ≥50% ranges from 206 to 246. There are fewer clades with support ≥50% in the three-gene trees (206–215 clades, $\bar{x} = 212$) than in the plastome trees (224–246, $\bar{x} = 239$).

In pairwise comparisons of trees generated from data partitions differing by the inclusion or exclusion of positively selected codons (Table 3), the number of shared clades (i.e., identical clades with support ≥50%) ranges from 192 to 238 and the number of shared clades with maximal support from 90 to 203. When positively selected sites are excluded, the number of shared clades with increased support in one of the compared trees ranges from 11 to 46 ($\bar{x} = 22 \pm 12$), with identical support in the compared trees from 100 to 205 ($\bar{x} = 166 \pm 42$), and with decreased support in one of the compared trees from 12 to 58 ($\bar{x} = 32 \pm 15$). In all comparisons, the ranges of support differences for shared clades are considerable (up to 48%), and some clades identified in each tree with BP ≥50% are unsupported in the other. Overall, however, excluding positively selected codons has relatively little effect on support values, with average support differences for shared clades in each comparison ranging from 1% to 4%. The largest effect of positively

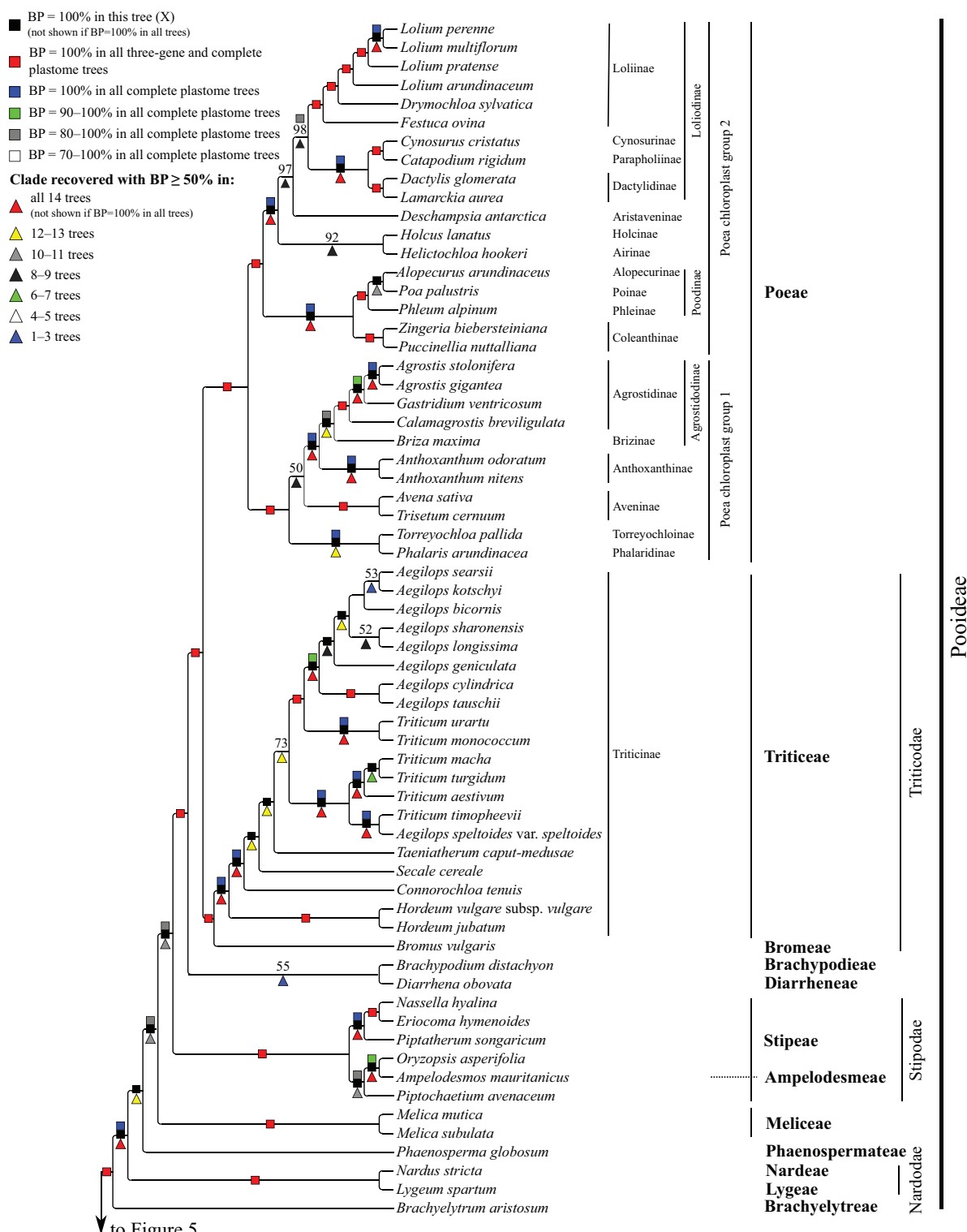

**Figure 4 Pooideae portion of the maximum likelihood tree inferred from complete plastomes excluding gapped sites and including positively selected sites (tree X).** Bootstrap support, when ≥50%, for clades in this tree and clades shared among this and other trees, is summarized along branches, according to the legend. Numbers along branches are bootstrap support values in tree X.

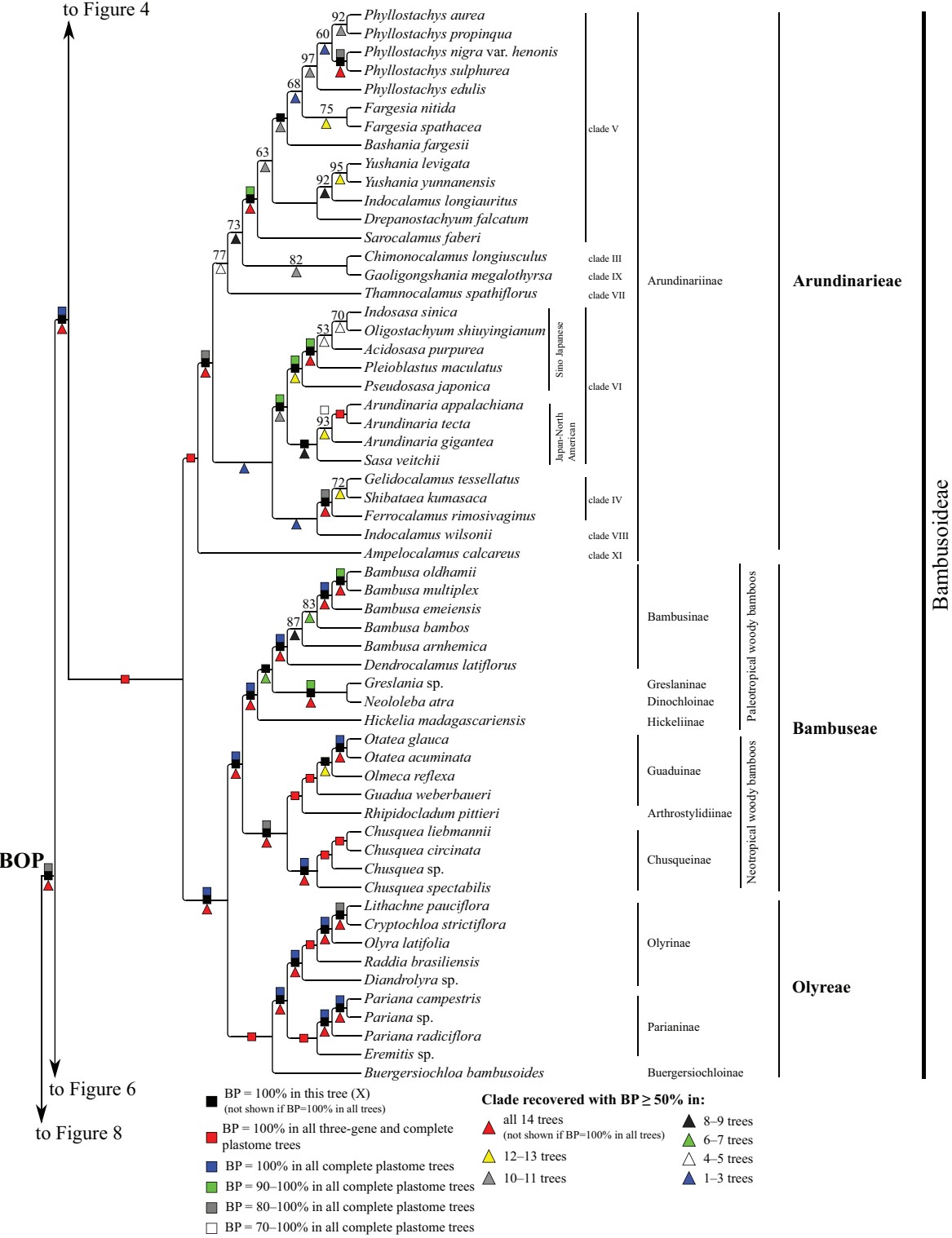

**Figure 5 Bambusoideae portion of the maximum likelihood tree inferred from complete plastomes excluding gapped sites and including positively selected sites (tree X).** Bootstrap support, when ≥50%, for clades in this tree and clades shared among this and other trees, is summarized along branches, according to the legend. Numbers along branches are bootstrap support values in tree X.

to Figure 5

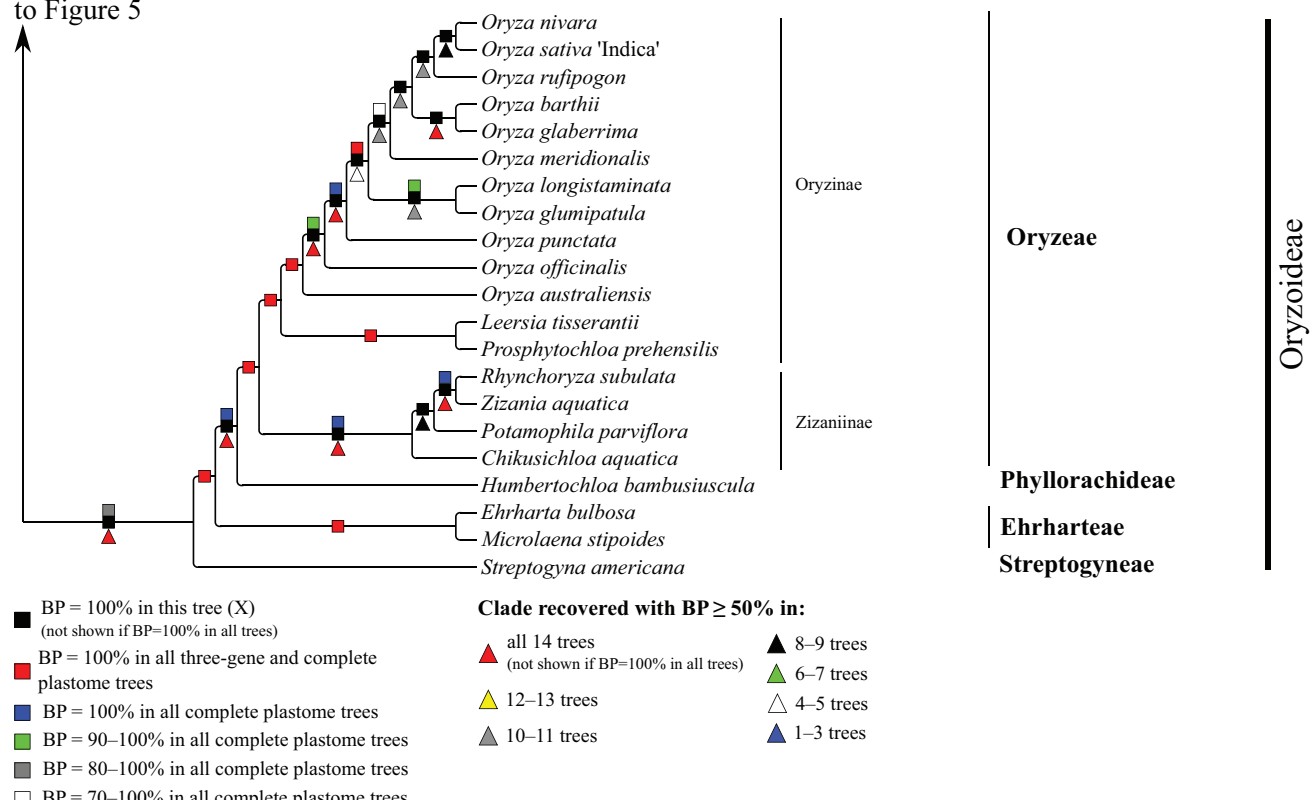

**Figure 6 Oryzoideae portion of the maximum likelihood tree inferred from complete plastomes excluding gapped sites and including positively selected sites (tree X).** Bootstrap support, when ≥50%, for clades in this tree and clades shared among this and other trees, is summarized along branches, according to the legend. Numbers along branches are bootstrap support values in tree X.

selected sites is in comparison B vs. D. These are three-gene data sets, with all genes known to have positively selected sites.

In pairwise comparisons of trees generated from data partitions differing only by inclusion or exclusion of gapped sites (Table 3), the number of shared clades ranges from 197 to 230 and the number of shared clades with maximal support from 95 to 200. When gapped sites are excluded, the number of shared clades with increased support in one of the compared trees ranges from 3 to 32 ($\bar{x} = 15 \pm 9$), with identical support in the compared trees from 103 to 201 ($\bar{x} = 164 \pm 38$), and with decreased support in one of the compared trees from 19 to 68 ($\bar{x} = 38 \pm 13$). In all comparisons, the ranges of support differences for shared clades are considerable (up to 44%), and, as in the comparisons excluding positively selected sites, some clades identified in each tree with BP ≥50% are unsupported in the other. Overall, excluding gapped sites has relatively little effect on support values, with average support differences for shared clades in each comparison ranging from 2% to 4%. The largest effect of gapped sites are in comparisons Y vs. Z (complete plastome data sets excluding positively selected sites) and Q vs. R (noncoding plastome data).

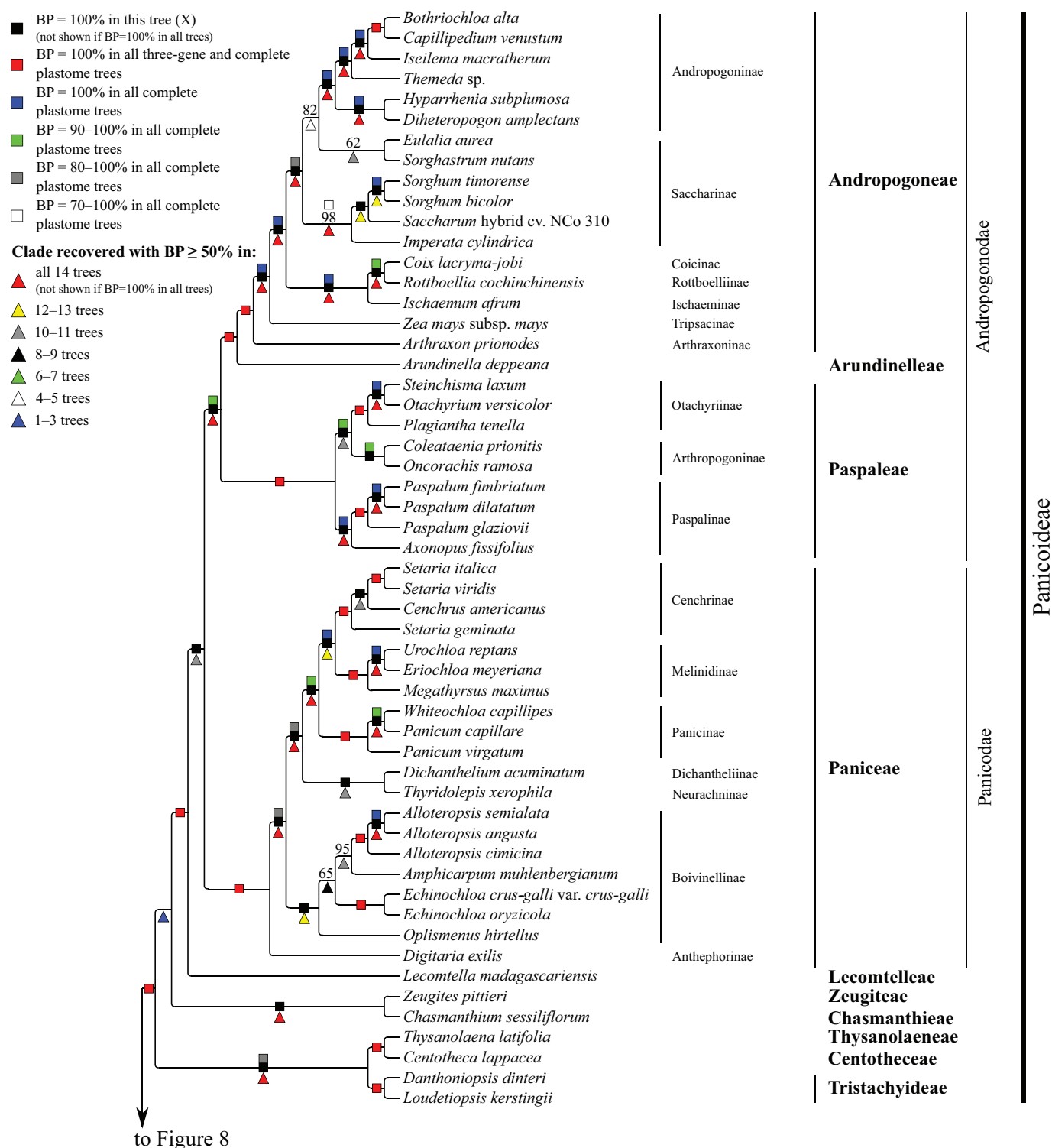

**Figure 7 Panicoideae portion of the maximum likelihood tree inferred from complete plastomes excluding gapped sites and including positively selected sites (tree X).** Bootstrap support, when ≥50%, for clades in this tree and clades shared among this and other trees, is summarized along branches, according to the legend. Numbers along branches are bootstrap support values in tree X.

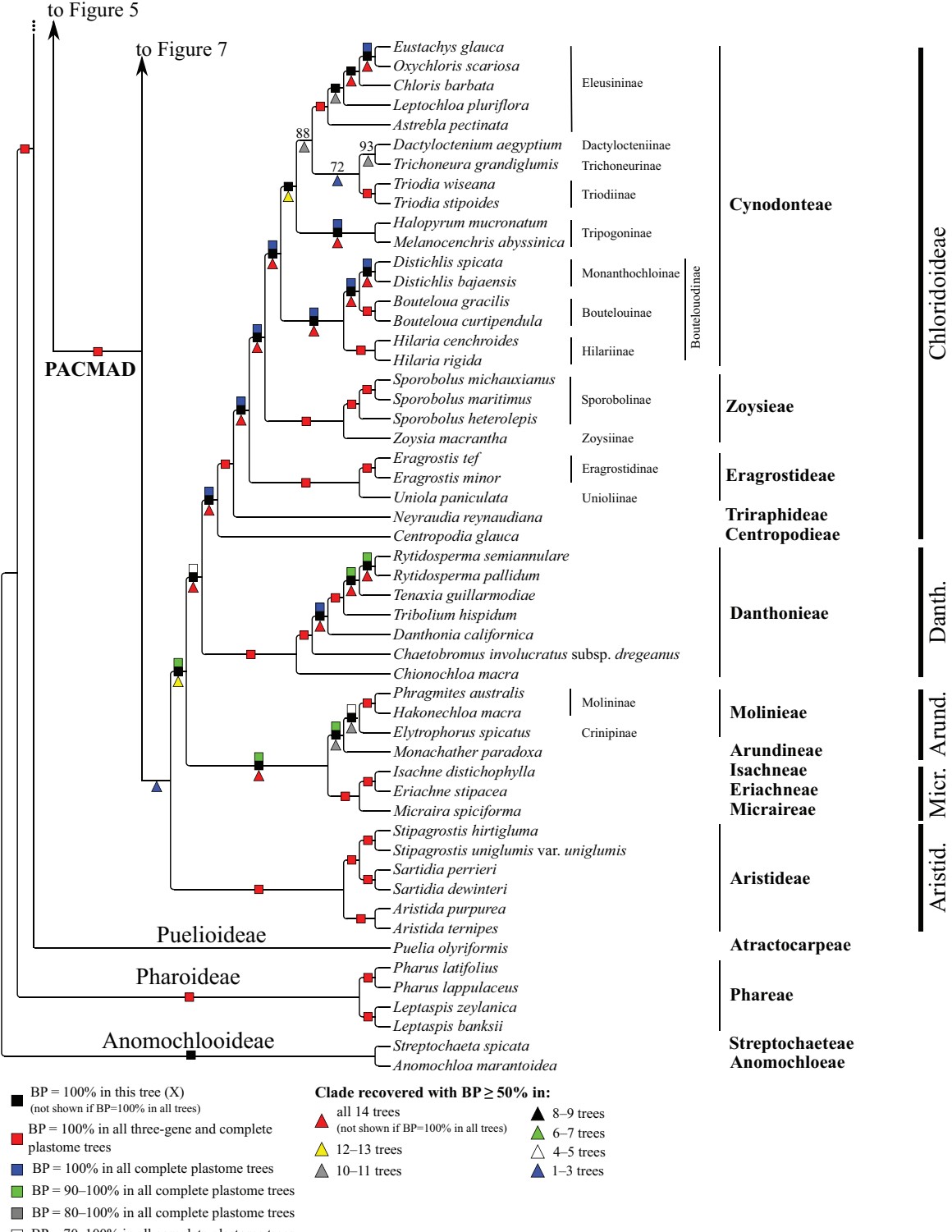

Figure 8 **Anomochlooideae, Pharoideae, Puelioideae, Arundinoideae, Chloridoideae, Danthonioideae and Micrairoideae portion of the maximum likelihood tree inferred from complete plastomes excluding gapped sites and including positively selected sites (tree X).** Bootstrap support, when ≥50%, for clades in this tree and clades shared among this and other trees, is summarized along branches, according to the legend. Numbers along branches are bootstrap support values in tree X.

**Table 3  Pairwise comparisons of trees inferred from plastome partitions differing by the inclusion or exclusion of positively selected sites and gapped sites.**

Positively selected sites included or excluded

|  | A vs. C | B vs. D | E vs. G | F vs. H | W vs. Y | X vs. Z | $\bar{x}$ |
|---|---|---|---|---|---|---|---|
| Number of shared clades with BP ≥50% | 201 | 192 | 227 | 229 | 233 | 238 | 220 ± 17 |
| Number (%) of shared clades with BP = 100% | 106 (53) | 90 (47) | 185 (81) | 185 (81) | 203 (87) | 198 (83) | 160 ± 45 |
| Number of clades with BP ≥50% (≥70%) in one tree and <50% in the other | 13(4) in A, 14 (2) in C | 19(5) in B, 14 (6) in D | 12(4) in E, 10 (3) in G | 9(5) in F, 8 (2) in H | 11(10) in W, 15 (11) in Y | 4(0) in X, 2(1) in Z |  |
| Range of support difference | −39 to 41 | −29 to 40 | −27 to 26 | −30 to 22 | −20 to 30 | −48 to 40 |  |
| Mean support difference ± s.d. [absolute mean support difference ± s.d.] | 2 ± 9 [4 ± 9] | 1 ± 9 [4 ± 8] | 1 ± 5 [2 ± 5] | 0 ± 4 [1 ± 4] | 0 ± 4 [1 ± 4] | 0 ± 7 [2 ± 7] |  |
| Number of shared clades with identical support | 114 | 100 | 188 | 189 | 205 | 201 | 166 ± 42 |
| Number of shared clades with increased support when positively selected/gapped sites removed | 29 | 46 | 15 | 16 | 16 | 11 | 22 ± 12 |
| Number of shared clades with decreased support when positively selected/gapped sites removed | 58 | 46 | 24 | 24 | 12 | 26 | 32 ± 15 |

Gapped sites included or excluded

|  | A vs. B | C vs. D | E vs. F | G vs. H | Q vs. R | W vs. X | Y vs. Z | $\bar{x}$ |
|---|---|---|---|---|---|---|---|---|
| Number of shared clades with BP ≥50% | 200 | 197 | 230 | 228 | 211 | 227 | 224 | 215 ± 14 |
| Number (%) of shared clades with BP = 100% | 104 (52) | 95 (48) | 184 (80) | 185 (81) | 158 (75) | 200 (88) | 189 (84) | 159 ± 40 |
| Number of clades with BP ≥50% (≥70%) in one tree and <50% in the other | 14(7) in A, 11 (3) in B | 18(7) in C, 9(2) in D | 9(1) in E, 8(4) in F | 10(3) in G, 8(2) in H | 31(20) in Q, 14(9) in R | 11(3) in W, 15 (8) in X | 22(16) in Y, 22 (11) in Z |  |
| Range of support difference | −16 to 33 | −44 to 33 | −28 to 18 | −27 to 30 | −30 to 43 | −22 to 31 | −23 to 40 |  |
| Mean support difference ± s.d. [absolute mean support difference ± s.d.] | 2 ± 5 [3 ± 5] | 0 ± 9 [4 ± 8] | 0 ± 4 [2 ± 4] | 0 ± 5 [2 ± 5] | 3 ± 8 [3 ± 8] | 1 ± 5 [1 ± 5] | 1 ± 6 [2 ± 6] |  |
| Number of shared clades with identical support | 112 | 103 | 187 | 191 | 159 | 201 | 193 | 164 ± 38 |
| Number of shared clades with increased support when positively selected/gapped sites removed | 20 | 32 | 15 | 19 | 3 | 5 | 9 | 15 ± 9 |
| Number of shared clades with decreased support when positively selected/gapped sites removed | 68 | 62 | 28 | 19 | 49 | 21 | 22 | 38 ± 13 |

**Note:**
For each comparison the table records the number of shared (identical) clades with bootstrap support (BP) ≥50%, maximum BP, identical BP, and BP ≥50% and ≥70% in one tree but <50% in the other; the range of BP difference and mean difference in BP for shared clades when positively selected and gapped sites are excluded; and the numbers of shared clades for which BP increases and decreases when positively selected and gapped sites are excluded.

**Table 4 Pairwise comparison of bootstrap support (BP) for clades identified in three-gene (A–D) and complete plastome coding trees (E–H).**

|  | A vs. E | B vs. F | C vs. G | D vs. H |
|---|---|---|---|---|
| Number of shared clades with maximum BP | 119 | 103 | 116 | 103 |
| Number of shared clades with BP >50% (≥10% higher) in the first of the two compared trees, and the range of the differences in BP for shared clades | 6 (4), 2–29% | 6 (2), 2–28% | 6 (1), 1–14% | 3 (0), 1–7% |
| Number of shared clades with BP >50% and higher (≥10% higher) in the second of the two compared trees, and the range of the differences in BP for shared clades | 71 (26), 1–44% | 91 (50), 1–47% | 77 (38), 1–49% | 86 (44), 1–49% |
| Number of clades with BP ≥70% in the first of the compared trees and <50% in the second of the compared trees | 8 | 2 | 5 | 7 |
| Number of clades with BP support <50% in the first of the compared trees and ≥70% in the second of the compared trees | 34 | 31 | 30 | 31 |

In each comparison of three-gene vs. plastome coding trees, 103–119 shared clades are maximally supported (Table 4). Three to six shared clades are more strongly supported in the three-gene trees than the plastome coding trees; the difference in support is ≥10% in 0–4 of these clades. Reciprocally, 71–91 shared clades are more strongly supported in the plastome coding trees than in the three-gene trees; the difference in support is ≥10% in 26–50 of these clades. Two to eight clades are identified with support ≥70% in the three-gene trees that are unsupported in the plastome coding trees. Reciprocally, 30–34 clades are identified with support ≥70% in the plastome coding trees that are unsupported in the three-gene trees. Twenty-seven clades not supported in any of the three-gene trees are supported (BP ≥ 50%) in one or more of the plastome coding trees, and 25 clades identified with BP ≥ 50% in one or more three-gene trees are not supported in the plastome coding trees (Dataset S16). Nine of the latter have support ≥70% in at least one three-gene tree: PCMAD [aristidoid sister], *Monachather* + Micrairoideae, *Dactyloctenium* + Tripogoninae, Trichoneurinae + Eleusininae, Otachyriinae + Paspalinae, Bambusinae + Greslaninae + Dinochloinae, *Bambusa multiplex* + *Bambusa oldhamii* + *Bambusa emeiensis* + *Bambusa bambos* + *Dendrocalamus*, *Diarrhena* + Bromeae + Poeae + Triticeae, and Agrostidinae + Brizinae + Anthoxanthinae + Aveninae.

We define conflicting clades as those with moderate to strong support (BP ≥ 70%) in two or more trees and different but overlapping sets of species. In 22 instances one or more taxa are part of two conflicting clades, and in five instances one or more taxa are part of three conflicting clades (Table 5). Nineteen conflicting clades are present in a single tree, 20 in two to four trees, and 18 in five or more trees. Conflicts occur in Bambusoideae, Chloridoideae, Oryzoideae, Panicoideae and Pooideae, among subfamilies of the PACMAD clade and among taxa of the Arundinoideae + Micrairoideae clade. There are four instances of conflict among congeneric species in *Aegilops* L., *Bambusa* Schreb., *Oryza* and *Triticum*.

**Table 5 Summary of moderately to strongly supported conflicting clades among trees inferred from different plastome partitions.**

| Conflicting clades | Trees | Support (%) |
|---|---|---|
| Panicoideae + **Chloridoideae + Micrairoideae + Arundinoideae + Danthonioideae** ("aristidoid sister" hypothesis) // Aristidoideae + **Chloridoideae + Micrairoideae + Arundinoideae + Danthonioideae** ("panicoid sister" hypothesis) | A, Q, W, Y // R | 87–100 // 93 |
| Arundinoideae [*Phragmites + Hakonechloa + Elytrophorus + Monachather*] // *Monachather* + Micrairoideae | E–H, Q, R, W–Z // D | 93–100 // 72 |
| Tripogoninae + **Dactylocteniinae + Trichoneurinae + Triodiinae + Eleusininae** // Boutelouinae + Hilariinae + Monanthochloinae + **Dactylocteniinae + Trichoneurinae + Triodiinae + Eleusininae** | A–C, E–H, Q, W–Z // R | 83–100 // 95 |
| Dactylocteniinae + Trichoneurinae + **Triodiinae** // Dactylocteniinae + Trichoneurinae + **Eleusininae** | X // Q, Y | 72 // 77–92 |
| *Leptochloa* + **Chloris + Eustachys + Oxychloris** // *Astrebla* + **Chloris + Eustachys + Oxychloris** | A–H, W, X, Z // Q | 93–100 // 100 |
| Dactylocteniinae + Trichoneurinae // Eleusininae + Trichoneurinae // Dactylocteniinae + Tripogoninae | E–G, Q, R, W, X, Z // D // A | 73–100 // 80 // 72 |
| Centotheceae + Thysanolaeneae + Tristachyideae + **Zeugiteae + Chasmanthieae** // **Chasmanthieae + Zeugiteae** + the rest of Panicoideae except Centotheceae + Thysanolaeneae + Tristachyideae | E, F, G, X // Q, W, Y | 70–87 // 90–100 |
| Paniceae + **Paspaleae** // Lecomtelleae + **Paspaleae** | A, E–H, Q, X–Z // R | 89–100 // 72 |
| *Echinochloa + Amphicarpum + **Alloteropsis*** // *Oplismenus + Amphicarpum + **Alloteropsis*** // *Echinochloa + Oplismenus + **Alloteropsis*** | A, B, F // W // Y | 70–84 // 75 // 97 |
| *Amphicarpum + Alloteropsis* // *Amphicarpum + Thyridolepis* // *Oplismenus + Alloteropsis* | B, E–H, Q, R, X, Z // Y // Y | 71–100 // 97 // 73 |
| *Cenchrus + **Setaria italica + Setaria viridis*** // ***Setaria italica + Setaria viridis*** + *Setaria geminata* | G, H, Q, R, W, X–Z // B, D, E | 100 // 77–100 |
| Arthropogoninae + **Otachyriinae** // **Otachyriinae** + Paspalinae | E–H, Q, R, W–Z // C, D | 98–100 // 74–95 |
| ***Thyridolepis*** + *Dichanthelium* // ***Thyridolepis*** + *Amphicarpum* | E–H, Q, R, W, X, Z // Y | 98–100 // 97 |
| ***Eulalia + Sorghastrum*** + Andropogoninae // ***Eulalia + Sorghastrum*** + *Imperata + Saccharum + Sorghum* | E, F, X, Z // G, Q, W, Y | 70–82 // 75–93 |
| *Potamophila + **Zizania + Rhynchoryza*** // *Chikusichloa + **Zizania + Rhynchoryza*** | E–H, X, Z // Q, W, Y | 100 // 100 |
| ***Oryza sativa*** + *Oryza nivara* // ***Oryza sativa*** + *Oryza rufipogon* | G, Q, R, W–Z // F | 95–100 // 87 |
| ***Bambusa multiplex + Bambusa oldhamii + Bambusa emeiensis*** + *Bambusa bambos* // ***Bambusa multiplex + Bambusa oldhamii + Bambusa emeiensis*** + *Bambusa arnhemica* // ***Bambusa multiplex + Bambusa oldhamii + Bambusa emeiensis*** + *Dendrocalamus* | R, W, X, Z // Q // G | 83–97 // 84 // 75 |
| ***Olmeca*** + *Otatea* // ***Olmeca*** + *Guadua* | A–H, Q, W–Z // R | 97–100 // 87 |
| ***Indosasa*** + *Oligostachyum* // ***Indosasa*** + *Pleioblastus* | Q, W–Z // E, F, H | 70–93 // 70–78 |
| ***Oligostachyum + Indosasa*** + *Pleioblastus* // ***Oligostachyum + Indosasa*** + *Acidosasa* | E–H, Y, Z // R, W | 72–93 // 83–93 |
| Brachypodieae + Poeae + Bromeae + Triticeae // Diarrheneae + Brachypodieae // Diarrheneae + Poeae + Bromeae + Triticeae | A–C, E, G, Q, W, Y // Z // D | 80–100 // 72 // 78 |
| *Triticum macha + Triticum turgidum* // *Triticum turgidum + Triticum aestivum* // *Triticum macha + Triticum aestivum* | A, C, F, H, R, X, Z // E, G, W // Y | 80–100 // 75–90 // 88 |
| *Aegilops kotschyi + Aegilops sharonensis* // *Aegilops longissima + **Aegilops sharonensis*** | Q, W, Y // A | 78 // 74 |
| *Poa + **Alopecurus*** // *Phleum + **Alopecurus*** | D–H, X–Z // Q, W | 85–100 // 100 |

| Conflicting clades | Trees | Support (%) |
|---|---|---|
| *Helictochloa* + *Holcus* // *Helictochloa* + *Deschampsia* + Dactylidinae + Loliinae | W–Y // F | 92–100 // 70 |
| Phalaridinae + Torreyochloinae + **Anthoxanthinae** + **Brizinae** + **Agrostidinae** // Aveninae + **Anthoxanthinae** + **Brizinae** + **Agrostidinae** | Q, R, Z // A–D, H, W, Y | 98–100 // 64–100 |
| Hickeliinae + **Dinochloinae** + **Greslaninae** // Bambusinae + **Dinochloinae** + **Greslaninae** | E–H, Q, W, Y // A, C, D, R, X, Z | 92–100 // 70–100 |

**Note:**
The table identifies clade compositions, the trees in which the clades are present with bootstrap support ≥70%, and bootstrap support for the clades. Shared taxa in conflicting clades are boldfaced.

## Phylogenetic relationships

The following descriptions of phylogenetic relationships refer only to the 10 plastome trees unless otherwise indicated. In all plastome trees, Pharoideae and Puelioideae are successively diverging sisters of a lineage comprising the BOP and PACMAD clades (Figs. 3–8). The BOP clade is maximally supported in all plastome trees except Y (BP = 88%), and the PACMAD clade is maximally supported in all plastome trees.

## BOP clade

In the plastome trees, Bambusoideae (BP = 100%), Oryzoideae (BP = 87–100%) and Pooideae (BP = 100%) are strongly supported, and Bambusoideae and Pooideae form a maximally supported clade (Figs. 3 and 5; Fig. S1).

### Pooideae

*Brachyelytrum aristosum* (Michx.) P. Beauv. ex Trel. (Brachyelytreae), *Lygeum spartum* L. (Lygeae) + *Nardus stricta* L. (Nardeae), *Phaenosperma globosum* Munro ex Benth. (Phaenospermateae), *Melica* L. (two species; Meliceae), and Stipeae–Ampelodesmeae diverge successively with respect to the rest of the Pooideae clade in all trees (BP = 88–100%). Within Stipeae–Ampelodesmeae (BP = 100%), *Eriocoma hymenoides* (Roem. and Schult.) Rydb. + *Nassella hyalina* (Nees) Barkworth (BP = 100%) and *Piptatherum songaricum* (Trin. and Rupr.) Roshev. ex Nikitina form a clade (BP = 100%), and *Oryzopsis asperifolia* Michx. + *Ampelodesmos mauritanicus* (Poir.) T. Durand & Schinz (Ampelodesmeae) (BP = 88–100%) and *Piptochaetium avenaceum* (L.) Parodi form a clade (BP = 88–100%).

Seven trees identify a clade comprising *Brachypodium distachyon* (L.) P. Beauv. (Brachypodieae), *Bromus vulgaris* (Hook.) Shear (Bromeae), Poeae and Triticeae, with varying support. Gapped sites in the plastome noncoding alignments contribute strongly to support for the clade. In tree Q, based on noncoding regions including gapped sites, and in the two complete plastome trees including gapped sites (W, Y), support for the clade is 100%, whereas in tree R, based on noncoding regions excluding gapped sites, support for the clade is less than 50%. Plastome coding regions also identify the clade, but with weaker support, when gapped sites are included (G, E; BP = 80%, 85%) and excluded (F, H; BP = 62%, 55%). When gapped sites are excluded from the complete plastome datasets, however, *Diarrhena obovata* (Gleason) Brandenburg (Diarrheneae) and Brachypodieae are sister taxa with weak (X, BP = 55%) and moderate (Z, BP = 72%)

support. In all trees, Bromeae and Triticeae are maximally supported sister taxa, and Bromeae + Triticeae and Poeae form a maximally supported clade.

Within Triticeae, *Hordeum* L. (two species sampled), *Connorochloa tenuis* (Buchanan) Barkworth, S.W.L. Jacobs & H.Q. Zhang, *Secale cereale* L. and *Taeniatherum caput-medusae* (L.) Nevski diverge successively with respect to the rest of the clade, with all branches strongly supported in most trees. An *Aegilops* L. + *Triticum* clade is variously supported (BP = 60–100%) (Fig. 4) and divided into two subclades. One subclade comprises *Triticum macha* Dekapr. & Menabde + *Triticum turgidum* L. + *Triticum aestivum* L. (BP = 100%) sister to *Aegilops speltoides* Tausch + *Triticum timopheevii* (Zhuk.) Zhuk. (BP = 100%). Relationships among taxa in the former clade conflict among trees (Table 5). The other subclade comprises *Triticum urartu* Thumanjan ex Gandilyan + *Triticum monococcum* L. (BP = 100%) sister to an eight-species *Aegilops* clade (BP = 98–100%) in which relationships are mostly strongly supported; however, relationships among *Aegilops kotschyi* Boiss., *Aegilops sharonensis* Eig and *Aegilops longissima* Schweinf. & Muschl. conflict among trees (Table 5).

Within Poeae, maximally supported clades correspond to Poeae chloroplast groups 1 and 2 (Fig. 4). In group 1, *Phalaris arundinacea* L. (Phalaridinae) and *Torreyochloa pallida* (Torr.) G.L. Church (Torreyochloinae) are sister taxa (BP =100%); Agrostidinae, comprising *Agrostis* L. (two species sampled), *Calamagrostis breviligulata* (Fernald) Saarela [syn. *Ammophila breviligulata* Fernald (*Saarela et al., 2017*)] and *Gastridium ventricosum* (Gouan) Schinz & Thell., is monophyletic; Agrostidinae + *Briza maxima* L. (Brizinae) form a clade (BP = 80–100%); and Agrostidinae + Brizinae + *Anthoxanthum* L. (two species; Anthoxanthinae) form a clade (BP = 100%). There is conflict among trees for the relative branching order of Phalaridinae + Torreyochloinae and Aveninae (*Avena sativa* L. and *Trisetum cernuum* Trin.) with respect to the remainder of the clade (Table 5).

Poeae chloroplast group 2 is divided into two subclades (Fig. 4). One comprises *Puccinellia nuttalliana* (Schult.) Hitchc. + *Zingeria biebersteiniana* (Claus) P.A. Smirn. (Coleanthinae) (BP = 100%) and *Alopecurus arundinaceus* Poir. (Alopecurinae) + *Phleum alpinum* L. (Phleinae) + *Poa palustris* L. (Poinae) (=supersubtribe Poodinae; BP = 100%). Relationships among *Phleum*, *Alopecurus* and *Poa* conflict among trees (Table 5). The other subclade of group 2 is recovered in all trees (BP = 77–100%) but R and comprises *Cynosurus cristatus* L. (Cynosurinae), *Catapodium rigidum* (L.) C.E. Hubb. (Parapholiinae), *Holcus lanatus* L. (Holcinae), *Deschampsia antarctica* E. Desv. (Aristaveninae), *Helictochloa hookeri* (Scribn.) Romero-Zarco (Airinae), *Dactylis glomerata* L. and *Lamarckia aurea* (L.) Moench (Dactylidinae) and *Festuca ovina* L., *Drymochloa sylvatica* (Pollich) Holub and *Lolium* L. (four species) (Loliinae). Holcinae and Airinae are sister taxa in six trees (E, G, Q, W–Y, BP = 50–100%). In all trees, Cynosurinae and Parapholiinae are sister taxa (BP = 100%) and form a clade with Dactylidinae (BP = 83–100%), and in all trees except R these subtribes form a clade with Loliinae (BP = 92–100%).

### Bambusoideae

Within Bambusoideae (Fig. 5), Arundinarieae, Bambuseae and Olyreae are each monophyletic (BP = 100%), and Bambuseae and Olyreae are sister taxa (BP = 100%).

Within Olyreae, *Buergersiochloa bambusoides* Pilg. (Buergersiochloinae) is sister to a clade of *Eremitis* Döll sp. + *Pariana* Nakai (three accessions and at least two species) (Parianinae; BP = 100%) and *Diandrolyra* Stapf sp. + *Raddia brasiliensis* Bertol. + *Olyra latifolia* L. + *Cryptochloa strictiflora* (E. Fourn.) Swallen + *Lithachne pauciflora* (Sw.) P. Beauv. (Olyrinae; BP = 100%). Relationships within Parianinae and Olyrinae are strongly supported.

Within Bambuseae, two major clades are identified (Fig. 5). The paleotropical woody bamboo clade comprises *Hickelia madagascariensis* A. Camus (Hickeliinae), *Nelololeba atra* (Lindl.) Widjaja (Dinochloinae) + *Greslania* Balansa sp. (Greslaniinae) and *Bambusa* spp. + *Dendrocalamus latiflorus* Munro (Bambusinae) (BP = 100%). Dinochloinae and Greslaniinae form a clade (BP = 98–100%), but relationships among Dinochloinae + Greslaniinae, Bambusinae and Hickeliinae conflict among trees (Table 5). The five species of *Bambusa* form a clade in seven trees (E, Q–Z, BP = 50–93%), with a maximally supported subclade comprising *Bambusa emeiensis* L.C. Chia & H.L. Fung and *B. multiplex* (Lour.) Raeusch. ex Schult. & Schult. f. + *B. oldhamii* Munro. Relationships among this latter clade, *B. bambos* (L.) and *B. arnhemica* Voss vary among trees (Table 5). The neotropical woody bamboo clade (BP = 88–100%) comprises *Chusquea* Kunth (four accessions representing at least three species; Chusqueinae; BP = 100%), *Rhipidocladum pittieri* (Hack.) McClure (Arthrostylidiinae), and *Guadua weberbaueri* Pilg., *Olmeca reflexa* Soderstr. and *Otatea acuminata* (Munro) C.E. Calderón & Soderstr. (Guaduinae; BP = 100%). Within Chusqueinae, *Chusquea spectabilis* L.G. Clark is robustly resolved as sister to the remainder of the genus. Arthrostylidiinae and Guaduinae are sister taxa (BP = 100%). Relationships among the three sampled genera of Guaduinae conflict among trees (Table 5).

Within Arundinarieae (Fig. 5), *Ampelocalamus calcareus* C.D. Chu & C.S. Chao (clade XI; see Methods for details of informal clade names in this tribe) is sister to the rest of the subtribe. *Gelidocalamus tessellatus* T.H. Wen & C.C. Chang and *Shibataea kumasaca* (Zoll. ex Steud.) Makino are sister taxa in all trees (BP = 52–100%), and these plus *Ferrocalamus rimosivaginus* T.H. Wen form a clade in all trees (clade IV; BP = 80–100%). *Sasa veitchii* (Carrière) Rehder, *Arundinaria* Michx. (three species; BP = 73–100%), *Pseudosasa japonica* (Siebold & Zucc. ex Steud.) Makino ex Nakai, *Pleioblastus maculatus* (McClure) C.D. Chu & C.S. Chao, *Acidosasa purpurea* (Hsueh & T.P. Yi) Keng f., *Indosasa sinica* C.D. Chu & C.S. Chao and *Oligostachyum shiuyingianum* (L.C. Chia & P. But) G.H. Ye & Z.P. Wang form a clade (clade VI) in all trees (BP = 95–100%). *Indocalamus wilsonii* (Rendle) C.S. Chao & C.D. Chu (clade VIII) forms a clade with clades IV and VI in three trees (Q, W, Y, BP = 67–93%) and is sister to clade IV in two of those trees (W, Y, BP = 98, 75%). Within clade VI, *S. veitchii* and *Arundinaria* form a clade (BP = 63–98%) in all plastome trees except R, in which *S. veitchii* is sister to the rest of the subtribe (BP = 68%). The remaining taxa form a strongly supported clade, in which *Pseudosasa japonica* is sister to the rest of the lineage. However, relationships among *Pleioblastus maculatus*, *A. purpurea*, *I. sinica* and *O. shiuyingianum* are discordant among trees (Table 5).

A deep lineage of Arundinarieae is recovered in five trees (H, W–Z, BP = 50–92%) comprising *Chimonocalamus longiusculus* Hsueh & T.P. Yi (clade III), *Thamnocalamus*

*spathiflorus* (Trin.) Munro (clade VII), *Gaoligongshania megalothyrsa* (Hand.-Mazz.) D.Z. Li, Hsueh & N.H. Xia (clade IX), *Sarocalamus faberi* (Rendle) Stapleton, *Drepanostachyum falcatum* (Nees) Keng f., *Indocalamus longiauritus* Hand.-Mazz., *Yushania* Keng. F. (two species), *Bashania fargesii* (E.G. Camus) Keng f. & T.P. Yi, *Fargesia* Franch. (two species) and *Phyllostachys* Siebold & Zucc. (five species) (clade V). Clades III, IX and V form a clade in all trees except R (BP = 55–85%), and clades III and IX are sisters in all trees except Q and R (BP = 60–93%). Within clade V, *I. longiauritus* and *Yushania* form a clade in all trees except G, H and R. *Fargesia nitida* (Mitford ex Anonymous) Keng f. ex T.P. Yi and *Fargesia murielae* (Gamble) T. P. Yi form a clade (BP = 55–100%) in all trees except R. In all trees, the five species of *Phyllostachys* form a clade (BP = 65–100%), *Phyllostachys aurea* Carrière ex Rivière & C. Rivière and *Phyllostachys propinqua* McClure form a clade (BP = 68–100%), and *Phyllostachys sulphurea* (Carrière) Rivière & C. Rivière and *Phyllostachys nigra* var. *henonis* (Mitford) Stapf ex Rendle form a clade (BP = 87–100%). In subsets of trees, *Phyllostachys edulis* (Carrière) J. Houz. and *P. aurea + P. propinqua* form a clade (Q, W, Y, BP = 82–100%), and *P. aurea, P. propinqua, P. sulphurea* and *P. nigra* var. *henonis* form a clade (E, X, Z, BP = 58–68%). *Phyllostachys* and *Fargesia* are sister taxa in two trees (X, Y, BP = 68, 58%), and a broader clade including these genera and *Bashania* is recovered in all trees (BP = 52–100%).

### *Oryzoideae*

Oryzoideae is monophyletic and moderately to strongly supported in all trees (BP = 87–100%) (Fig. 6). *Streptogyna americana* C.E. Hubb. (Streptogyneae), *Ehrharta bulbosa* Sm. + *Microlaena stipoides* (Labill.) R. Br. (Ehrharteae; BP = 100%) and *Humbertochloa bambusiuscula* A. Camus & Stapf (Phyllorachideae) are successively diverging sisters to Oryzinae + Zizaniinae, with strong support for all branches. Within Oryzinae, *Leersia tisserantii* (A. Chev.) Launert and *Prosphytochloa prehensilis* (Nees) Schweick. are sister taxa (BP = 100%), and the multiple species of *Oryza* form a clade (BP = 100%). Relationships among most species of *Oryza* are strongly supported in most trees; however, relationships among *Oryza sativa* L., *Oryza nivara* Sharma & Shastry and *Oryza rufipogon* Griff. conflict among trees (Table 5). Within Zizaniinae, *Rhynchoryza subulata* (Nees) Baill. and *Zizania aquatica* L. are sister taxa (BP = 100%), and relationships among this clade, *Chikusichloa aquatica* Koidz. and *Potamophila parviflora* R. Br. are discordant among trees (Table 5).

## PACMAD clade

The deepest split in the PACMAD clade varies among complete plastome trees (Figs. 3 and 8, Table 4; Fig. S1). In two trees, Panicoideae are sister to the rest of PACMAD clade, with weak (Z, BP = 58%) and strong (R, BP = 93%) support for the ACMAD subclade. In three trees, Aristidoideae are sister to the rest of PACMAD clade, with moderate (W, BP = 87%) and strong (Q, Y, BP = 93% and 100%) support for the PCMAD subclade. In one tree, Aristidoideae + Panicoideae (F, BP = 52%) are the sister to the rest of the PACMAD clade. In the four other trees (E, G, H, X), no deep topology receives support ≥50%. The four remaining PACMAD subfamilies—Arundinoideae, Micrairoideae, Chloridoideae and

Danthonioideae—form a clade in all trees (BP = 96–100%; Figs. 3 and 8). Chloridoideae and Danthonioideae are sister taxa in all trees (BP = 78–100%). All taxa of Arundinoideae and Micrairoideae form a clade (BP = 100%). Arundinoideae is not consistently resolved as monophyletic, however, because in one three-gene tree *Monachather* forms a clade with Micrairoideae (D, BP = 72%) (Table 5).

### *Panicoideae*

At the base of the Panicoideae subtree, *Zeugites pittieri* Hack. (Zeugiteae) and *Chasmanthium sessiliflorum* (Poir.) H.O. Yates (Chasmantheae) form a clade (BP = 57–100%). *Loudetiopsis kerstingii* (Pilg.) Conert + *Danthoniopsis dinteri* (Pilg.) C.E. Hubb. (Tristachyideae; BP = 100%) and *Thysanolaena latifolia* (Roxb. ex Hornem.) Honda (Thysanolaeneae) + *Centotheca lappacea* (L.) Desv. (Centotheceae) (BP = 100%) are sister groups. Paniceae, Paspaleae, Andropogoneae and Arundinelleae are each monophyletic, and relationships among these four tribes, which compose the core Panicoideae, are maximally supported in all trees: Paniceae is sister to Paspaleae + (Andropogoneae + Arundinelleae) (Fig. 7). However, deep relationships among Zeugiteae + Chasmantheae, Tristachyideae, Centotheceae + Thysanolaeneae and Paniceae + Paspaleae + Andropogoneae + Arundinelleae vary among trees (Table 5).

Within Andropogoneae, most aspects of relationships are strongly supported, including the successive branching of *A. prionodes* (Arthraxoninae) and *Zea mays* L. (Tripsacinae) sister to the rest of the tribe. *Coix lacryma-jobi* L. (Coicinae) and *Rottboellia cochinchinensis* (Lour.) Clayton (Rottboelliinae) are sister taxa (BP = 93–100%), Coicinae + Rottboelliinae and *Ischaemum afrum* (J.F. Gmel.) Dandy (Ischaeminae) form a clade (BP = 87–100%), and this three-subtribe clade is sister to the rest of Andropogoneae. The two species of *Sorghum* Moench are sister taxa (BP = 100%), *Sorghum* and *Saccharum* cv. NCo310 are sister taxa in all trees (BP = 87–100%) except R, and these plus *Imperata cylindrica* (L.) P. Beauv. form a clade (BP = 72–100%). *Eulalia aurea* (Bory) Ku and *Sorghastrum nutans* (L.) Nash are sister taxa in all trees (BP = 62–100%) except F, but relationships among these species, the rest of Saccharinae, and Andropogoninae are discordant among trees (Table 5). Saccharinae, comprising *Sorghum*, *Saccharum*, *Imperata*, *Eulalia* and *Sorghastrum*, is monophyletic only in trees G, Q, W and Z (BP = 75–93%). Within Andropogoninae, the following lineages diverge successively with strong support: *Diheteropogon amplectens* var. *catangensis* (Chiov.) Clayton + *Hyparrhenia subplumosa* Stapf, *Themeda* Forssk. sp., *Iseilema macratherum* Domin and *Capillipedium venustum* (Thwaites) Bor + *Bothriochloa alta* (Hitchc.) Henrard.

Within Paniceae, *Digitaria exilis* (Kippist) Stapf (Anthephorinae) is sister to the rest of the tribe (BP = 88–100%) (Fig. 7). *Alloteropsis* J. Presl (three species), *Amphicarpum muhlenbergianum* (Schult.) Hitchc., *Echinochloa* P. Beauv. (two species) and *Oplismenus hirtellus* (L.) P. Beauv. form a clade (Boivinellinae; BP = 100%) in all trees except Y, in which *Amphicarpum muhlenbergianum* and *Thyridolepis xerophila* (Domin) S.T. Blake (Neurachninae) form a clade (BP = 97%). Within Boivinellinae, relationships among the four genera are discordant among trees (Table 5). *Alloteropsis angusta* Stapf and *Alloteropsis semialata* (R. Br.) Hitchc. are sister taxa (BP = 100%). The remaining taxa of

Paniceae form a clade in all trees (BP = 80–100%; Fig. 7). *Dichanthelium acuminatum* (Sw.) Gould & C.A. Clark (Dichantheliinae) and *T. xerophila* (Neurachninae) are sister taxa (BP = 98–100%) in all trees except Y (see above). The two species of *Panicum* L. and *Whiteochloa capillipes* (Benth.) Lazarides form a clade (Panicinae; BP = 100%), but *Panicum* is not monophyletic because *Panicum capillare* L. and *W. capillipes* are sister taxa (BP = 97–100%). *Eriochloa meyeriana* (Nees) Pilg. and *Urochloa reptans* (L.) Stapf are sister taxa (BP = 100%), and these species plus *Megathyrsus maximus* (Jacq.) B.K. Simon & S.W.L. Jacobs form a clade (Melinidinae; BP = 100%). *Cenchrus americanus* (L.) Morrone and the three species of *Setaria* P. Beauv. form a clade (Cenchrinae; BP = 100%) in all trees, but *Setaria* is monophyletic only in trees E and F (BP = 98–100%). In all other trees, *Setaria geminata* (Forskk.) Veldkamp is sister to *Cenchrus americanus* + (*Setaria italica* (L.) P. Beauv. + *Setaria viridis* (L.) P. Beauv.). Cenchrinae and Melinidinae are sister taxa (BP = 100%), and Panicinae and Cenchrinae + Melinidinae form a broader clade (BP = 98–100%).

Within Paspaleae, *Axonopus fissifolius* (Raddi) Kuhlm. and *Paspalum* L. (three species) form a clade (Paspalinae; BP = 100%). *Otachyrium versicolor* (Döll) Henrard and *Steinchisma laxum* (Sw.) Zuloaga are sister taxa (BP = 100%), and these plus *Plagiantha tenella* Renvoize form a clade (Otachyriinae; BP = 100%). *Coleataenia prionitis* (Nees) Soreng and *Oncorachis ramosa* (Zuloaga & Soderstr.) Morrone & Zuloaga form a clade (Arthropogoninae; BP = 98–100%; Fig. 7). Arthropogoninae and Otachyriinae are sister groups (BP = 98–100%).

### Chloridoideae

Within Chloridoideae, *Centropodia glauca* (Nees) Cope (Centropodieae), *Neyraudia reynaudiana* (Kunth) Keng ex Hitchc. (Triraphideae), Eragrostideae, Zoysieae and Cynodonteae diverge successively, with maximum support for all branches (Fig. 8). Eragrostideae (BP = 100%) comprises *Eragrostis* Wolf (two species; Eragrostidinae) and *Uniola paniculata* L. (Unioliinae). Zoysieae (BP = 100%) comprises *Zoysia macrantha* Desv. (Zoysiinae) and three species of *Sporobolus* R. Br. *nom. cons.* (Sporobolinae; BP = 100%). *Sporobolus michauxianus* (Hitchc.) P.M. Peterson & Saarela and *Sporobolus maritimus* (Curtis) P.M. Peterson & Saarela are sister taxa (BP = 100%). The eight sampled subtribes of Cynodonteae form a clade in all trees (BP = 100%). *Bouteloua* Lag. (two species; Boutelouinae; BP = 100%) + *Distichlis* Raf. (two species; Monanthochloinae; BP = 100%) and *Hilaria* Kunth (two species; Hilariinae; BP = 100%) form a clade corresponding to supersubtribe Boutelouodinae. *Melanocenchris abyssinica* (R. Br. ex Fresen.) Hochst. and *Halopyrum mucronatum* (L.) Stapf form a clade (Tripogoninae; BP = 100%), as do the two species of *Triodia* R. Br. (Triodiinae; BP = 100%). *Trichoneura grandiglumis* (Nees) Ekman (Trichoneurinae) and *Dactyloctenium aegyptium* (L.) Willd. (Dactylocteniinae) are sister taxa (BP = 68–100%). Within Eleusininae (BP = 100%), *Oxychloris scariosa* (F. Muell.) Lazarides + *Eustachys glauca* Chapm. (BP = 100%) and *Chloris barbata* Sw. form a clade (BP = 100%), and relationships among this clade, *Astrebla pectinata* (Lindl.) F. Muell. ex Benth and *Leptochloa pluriflora* (E. Fourn.) P.M. Peterson & N. Snow are discordant among trees (Table 5). Eleusininae, Triodiinae,

Dactylocteniinae and Trichoneurinae form a clade in all trees (BP = 65–97%), but relationships vary both within the clade and among the four-subtribe clade, Tripogoninae and Boutelouodinae (Table 5).

## DISCUSSION

We conducted plastome phylogenomic analyses of 250 species of grasses, many of which have not previously been combined in a single study, including 15 newly generated plastomes from six subfamilies. Deep relationships among grass subfamilies are fully congruent with most previous few-gene plastid and plastome studies that identified Anomochlooideae, Pharoideae and Puelioideae as successive sisters to a clade comprising the BOP and PACMAD clades (*Clark, Zhang & Wendel, 1995*; *Grass Phylogeny Working Group, 2001*; *Duvall et al., 2007*; *Bouchenak-Khelladi et al., 2008*; *Saarela & Graham, 2010*; *Grass Phylogeny Working Group II, 2012*; *Jones, Burke & Duvall, 2014*; *Burke et al., 2016a*, *2016b*). The plastome phylogeny represents 85% of the lineages of grasses currently recognized as tribes, 67% as subtribes, 23% as genera and ca. 2% as species (*Peterson, Romaschenko & Herrera Arrieta, 2017a*; *Soreng et al., 2017*), and our results corroborate many aspects of relationships among tribes, subtribes, genera and species identified in previous plastid studies, in most cases with increased support here. Indeed, over 230 clades are identified with maximum support in at least one plastome tree, 85 clades are maximally supported in all 14 trees, and 144 clades are identified with maximum support in all plastome trees. Clades that are maximally supported in all trees are distributed from the deepest to the shallowest levels of the grass tree of life, including the branches defining the spikelet clade (all Poaceae excluding Anomochlooideae), the BOP + PACMAD clade, the PACMAD clade, as well as clades comprising subfamilies, tribes, subtribes, genera and even multiple congeneric species.

### Comparison of three-gene vs. complete plastome coding trees

We compared trees inferred from three plastome coding regions and all plastome coding regions, the former being more representative of, and comparable to, the numerous few-gene/region phylogenetic studies of grasses conducted previously. If phylogenetic signal among plastome CDSs is congruent, support for a topology, when less than maximal in a few-gene tree, would be expected to increase as the number of CDSs and phylogenetically informative characters in an analysis increase. More than 100 shared clades are maximally supported in each of the compared three-gene and plastome coding trees. For these clades, there is sufficient phylogenetic signal in the three-gene datasets to robustly resolve relationships, and recovery of the same maximally supported clades in the plastome coding trees indicates there is either no conflict among plastome coding regions, or minimal conflict that does not affect support levels; the current data do not distinguish between these two possibilities. On the other hand, there is increased support for many clades in plastome coding trees compared to three-gene trees, consistent with our expectations and with results of earlier phylogenomic studies of grasses (*Jones, Burke & Duvall, 2014*; *Cotton et al., 2015*; *Saarela et al., 2015*; *Burke et al., 2016a*, *2016b*; *Duvall et al., 2016*, *2017*; *Orton et al., 2017*), confirming the utility of plastome phylogenomic

studies for clarifying phylogenetic relationships at multiple hierarchical levels of the grass family.

However, we also found some differences in resolution and support among three-gene and plastome coding trees. Twenty-three clades identified in one or more of the three-gene trees are not present in the plastome coding trees (Dataset S16). Thirteen of these are only weakly supported in one to three of the three-gene trees, but the remaining ten are moderately to strongly supported (BP $\geq$ 70%) in one or more of the trees; these clades represent relationships among subfamilies, tribes, subtribes and species of a bamboo genus. In most cases, the plastome coding trees identify alternative moderately to strongly supported topologies, indicative of character conflict among the three-gene data and the complete plastome coding data. In the plastome coding trees, the differing signal in the three-gene data, whatever its origin, may be "swamped" by the stronger signal in the much larger plastome coding dataset. Similar conflict was identified among few-gene vs. plastome coding partitions in a study of monocot phylogeny (Davis et al., 2013). Overall, these results indicate that supported clades in few-gene plastid trees may sometimes be misleading, such that well-supported and few-gene phylogenies should not necessarily be accepted as the "final word" on plastome phylogenetic relationships, until compared with whole plastome phylogenies that maximize available phylogenetic information in the plastome.

## Comparison of coding, noncoding and complete plastome partitions

We also identified some strongly conflicting topologies among our analyses of coding, noncoding and complete plastome partitions variously including and excluding coding gapped sites and positively selected sites (Table 4). Of these conflicting relationships, only one, or possibly none, is likely to be an accurate representation of the evolutionary history of the plastome, which is uni-parentally inherited. Although at least one instance of conflict was identified among each of the 14 trees, conflicting clades (relative to the majority topology) were more common in trees derived from partitions including gapped sites, noncoding data or both. For example, three of the four conflicting clades in tree R (plastome noncoding partition excluding gapped sites) are not present in any other trees, indicating the conflicting signal is restricted in these alignments to noncoding regions, whereas most other conflicting clades are present in two or more trees inferred from datasets including gapped sites.

## Effects of gapped sites on tree topology and support

The complete 250 plastome alignment includes many gapped sites, given that it is approximately 45% longer than the length of an average unaligned grass plastome (ca. 136,000 bp; Saarela et al., 2015). Gapped sites in an alignment of plastome sequences often reflect evolutionary history and may result from microstructural changes (indels, inversions) in specific lineages and from gene transfers. Such rare genomic changes are generally straightforward to align, at least among close relatives, and may be phylogenetically informative, as demonstrated for several grass lineages (Jones, Burke & Duvall, 2014; Burke et al., 2016a; Orton et al., 2017). However, gapped sites may also be

introduced in an alignment when portions of the plastome are difficult to align across divergent taxa, and poorly aligned regions may represent noise in an analysis. As such, unique clades in trees inferred from datasets including ambiguously aligned gapped sites may reflect systematic error. A particular challenge in phylogenomics is differentiating data signal reflecting evolutionary history from nonphylogenetic signal reflecting systematic error (*Rodríguez-Ezpeleta et al., 2007*).

Options for dealing with gapped sites in a phylogenetic analysis include removing them, assigning an additional state for each gap, coding gaps and treating them as binary characters, and treating gaps as missing data; the latter option is the most common approach (*Warnow, 2012*), and is what we did in a subset of analyses. *Warnow (2012)* demonstrated, however, that ML analyses may be statistically inconsistent when gaps are treated as missing data (but see *Truszkowski & Goldman, 2016*), and other studies have similarly shown that treating gaps as missing data can result in incorrect tree topologies in varying phylogenetic contexts (*Roure, Baurain & Philippe, 2013*; *Shavit Grievink, Penny & Holland, 2013*; *McTavish, Steel & Holder, 2015*). Therefore, as an alternative treatment for another subset of analyses we removed possibly–ambiguously aligned nucleotides by excluding all sites with a gap in at least one taxon (*Jones, Burke & Duvall, 2014*; *Cotton et al., 2015*; *Saarela et al., 2015*; *Attigala et al., 2016*; *Burke et al., 2016a*, *2016b*; *Duvall et al., 2016*; *Orton et al., 2017*). This allowed us to compare the effects on topology of including vs. excluding gapped sites. A limitation of this approach, however, is that potentially phylogenetically informative gapped sites or characters within alignment portions including gapped sites are also excluded from consideration. Differentiating between phylogenetically informative gaps and noninformative gaps in an alignment would require manual characterization of all alignment gaps, which we did not explore.

Another strategy for minimizing potential systematic error in plastome phylogenomic analyses caused by gapped sites is to exclude all noncoding data from consideration because the majority of gapped sites in plastome alignments are present in the noncoding partition. This would also exclude potential conflicting signal in the unambiguously aligned subset of noncoding data, like we found in a few instances in tree R. Researchers routinely exclude noncoding data from phylogenomic analyses, especially when generating phylogenies spanning multiple families and orders, where it is often difficult or impossible to align the more rapidly evolving noncoding regions among distantly related taxa. On the other hand, plastome noncoding regions are usually straightforward to align among closely-related species and genera with little overall plastome divergence, and branch support from noncoding data alone or when combined with coding data is sometimes stronger than from coding data for relationships among closely related taxa (*Ma et al., 2014*; *Saarela et al., 2015*). Examples in the current study of shared clades with higher support in noncoding compared to coding trees include Trichoneurinae + Dactylocteniinae + Triodiinae + Eleusininae, Trichoneurinae + Dactylocteniinae, *Oryza sativa* + *O. nivara* + *O. rufipogon*, the *Bambusa* clade, the *Phyllostachys* clade, and *Gelidocalamus* + *Shibataea* (Dataset S16).

## Effect of positively selected sites on tree topology and support

Positively selected codons have been shown to impact phylogenies inferred from single loci, and widely used phylogenetic methods do not automatically identify or correct for such bias. For example, in grasses in which photosynthetic genes, such as *rbcL* or *PEPC*, converge under selection for C$_4$ photosynthesis, misleading phylogenies can result (*Christin et al., 2008a, 2008b*). Multi-gene analyses should be somewhat less susceptible to selection bias since loci under different selective regimes would not be expected to reinforce an erroneous phylogenetic signal. Four protein coding loci commonly used for phylogenetic inference in grasses are *rbcL, matK, ndhF* and *rpoC2* (*Clark, Zhang & Wendel, 1995; Duvall et al., 2003, 2007; Grass Phylogeny Working Group II, 2012*). We find the highest numbers of selected codons in these four loci among all of the protein coding loci in the grass plastome. *Piot et al. (2018)* identified these same four genes as having the greatest signature of positive selection in plastomes of 113 PACMAD species. The considerable range in support for clades, differing by up to 41%, among three-gene trees that include and exclude positively selected sites indicates that the possibility of selection-induced bias in multi-gene analyses of these loci cannot be discounted.

Including positively selected sites in complete plastome analyses did not considerably affect topology and support for the majority of clades in our trees. However, like in the three-gene trees, we found a considerable range of BP for some clades in analyses including or excluding positively selected sites, indicating these sites influence phylogenetic reconstruction. In several instances clades identified when positively selected sites were included were not identified when those sites were excluded. This is most evident in the complete plastome trees: in tree W (including positively selected sites) there are 10 clades with support ≥70% that in tree Y (excluding positively selected sites) are unsupported. Reciprocally, in tree Y there are 11 clades with support ≥70% that in tree W are unsupported. Furthermore, two cases of strong conflict between trees W and Y can be attributed specifically to inclusion or exclusion of positively selected sites: relationships among *Oplismenus* P. Beauv., *Amphicarpum* Kunth, *Alloteropsis* and *Echinochloa*, and among *Triticum turgidum, T. aestivum* and *T. macha*.

Among our analyses, we also compared phylogenies inferred from partitions with all plastome coding loci against exclusively noncoding partitions. This diversity of loci across the functional groups of the plastome would again be expected to reduce any particular selection bias, but possibly at the expense of increasing the noise to signal ratio. Consistent with this observation is that the removal of selected sites in our analyses did not introduce extensive topological incongruities and that a greater range of support values for the included clades was observed.

## Comparison of plastome trees with previous phylogenetic studies of plastomes, subsets of plastid regions, and nuclear genes
### BOP Clade

The robust sister group relationship between Pooideae and Bambusoideae is congruent with most previous studies of multiple plastid genes and plastomes (*Bouchenak-Khelladi et al., 2008; Saarela & Graham, 2010; Zhang, Ma & Li, 2011; Grass Phylogeny Working*

Group II, 2012; *Wu & Ge, 2012*) as well as nuclear genes (*Zhao et al., 2013*; *Wysocki et al., 2016b*). However, in a recent plastid study Oryzoideae and Pooideae were recovered as sister taxa, although with uneven sampling throughout the family (*Pimentel et al., 2017*).

### Bambusoideae

Bambusoideae is divided into two tribes of woody bamboos (the tropical Bambuseae with eleven subtribes and two genera *incertae sedis*, and the temperate Arundinarieae with a single subtribe) and one of herbaceous bamboos (Olyreae, with three subtribes) (*Bamboo Phylogeny Group, 2012*; *Soreng et al., 2017*). Bamboo taxonomy is complicated by the fact that woody bamboos are polyploid. Many genera are paraphyletic or polyphyletic, and in many cases revised generic classifications have not yet been proposed. The plastome sampling includes six subtribes of Bambuseae, subtribe Arundinariinae and each subtribe of Olyreae (Buergersiochloinae, Olyrinae, Parianinae). *Wysocki et al. (2015)* identified an insertion of approximately 500 bp in the *rps16–trnQ* intergenic spacer in the 10 members of Arundinarieae they sampled that was not present in taxa of the other bamboo tribes, and we confirm this insertion is present in all members of Arundinarieae sampled here, with the single exception of *P. japonica*. A 150 bp inversion in the *trnD–psbM* intergenic spacer defines the Olyrinae clade (*Wysocki et al., 2015*). In the plastome trees, Arundinarieae is sister to Bambuseae + Olyreae, congruent with other studies of plastid data that identified paraphyly of woody bamboos (*Clark et al., 2007*; *Bouchenak-Khelladi et al., 2008*; *Sungkaew et al., 2008*; *Bamboo Phylogeny Group, 2012*; *Kelchner & Bamboo Phylogeny Group, 2013*). Phylogenetic studies of nuclear genes, however, identify Bambuseae and Arundinarieae as sister taxa, supporting monophyly of woody bamboos (*Triplett et al., 2014*; *Wysocki et al., 2016b*).

### Olyreae

Species of Olyreae fall on long branches relative to species of Arundinarieae and Bambuseae, indicating a faster mutation rate in plastomes of Olyreae than in woody bamboos. Within Olyreae, the robustly resolved relationships among Buergersiochloinae, Parianinae and Olyrinae in the plastome trees are congruent with previous plastid studies (*Oliveira et al., 2014*; *Wysocki et al., 2015*). Although the three sampled species of *Pariana* form a maximally supported clade in our trees, the genus is not monophyletic in plastid and ITS trees in *Oliveira et al. (2014)* because two species of *Eremitis*, including one now recognized in *Parianella* Hollowell, F.M. Ferreira & R.P. Oliveira (*Ferreira et al., 2013*; *Soreng et al., 2017*), are nested within it. A molecular phylogenetic analysis of these three genera is in progress (L.G. Clark, 2017, unpublished data). The robustly supported topology within Olyrinae is congruent with trees based on six plastid regions (*Zhang et al., 2016c*) and plastome-scale data (*Wysocki et al., 2015*), and better resolved and supported than in a tree based on the *trnD–trnT* intergenic spacer (*Oliveira et al., 2014*).

### Arundinarieae

Seven of the twelve major lineages (clades I–XII) of Arundinarieae are represented in the plastome trees, which in many cases are better resolved and supported than trees based on a few plastid regions. Within the tribe, only 12 clades of two or more taxa are weakly to

strongly supported in all plastome trees, indicating weak or conflicting signal in some of the plastome partitions analyzed. Placement of *A. calcareus* (clade XI) as sister to the rest the tribe is congruent with other studies (*Ma et al., 2014*; *Attigala et al., 2016*; *Zhang & Chen, 2016*; *Zhang et al., 2016c*), but some deep relationships among the other lineages of Arundinarieae are variously unsupported or discordant. In previous phylogenies based on complete plastomes, a lineage comprising clades IV, VI and VIII was identified, with strong support only in Bayesian trees (*Ma et al., 2014*; *Attigala et al., 2016*). This same clade is identified in two ML trees here (plastome noncoding including gapped sites and complete plastome including gapped sites) with weak to moderate support. Within the clade, the Chinese species *I. wilsonii* (clade VIII) is moderately to strongly supported as sister to clade IV in the two complete plastome trees including gapped sites. In the earlier plastome trees, *I. wilsonii* is also sister to clade IV (*Ma et al., 2014*; *Attigala et al., 2016*), but again with support only in the Bayesian trees. In few-gene plastid trees, placement of *I. wilsonii* is unresolved within Arundinarieae (*Zeng et al., 2010*; *Zhang et al., 2016c*). In the plastome noncoding tree, however, clades IV and VI are strongly supported as sister taxa, a topology that conflicts with the trees that identify clades VIII and IV as sister taxa. All known taxa that are part of the clade comprising clades VI, VIII and IV have leptomorph rhizomes (*Attigala et al., 2016*), but we are not aware of morphological characters that would favor one or the other of the topologies among the three clades. Resolution of relationships among these clades is likely complicated by the short branches that define each of them (Fig. S1).

The robustly supported clade IV has been recovered in other studies, many of which are more broadly sampled than our analyses (*Triplett & Clark, 2010*; *Zeng et al., 2010*; *Attigala et al., 2014*; *Zhang et al., 2016c*). The plastome tree here includes three of the five genera recognized in the clade compared to two genera included in earlier plastome studies (*Ma et al., 2014*; *Attigala et al., 2016*). The sister relationship between *Gelidocalamus tessellatus* and *Shibataea kumasaca* is congruent with the plastid tree in *Zeng et al. (2010)*, but contrasts with a plastid tree in which species of *Ferrocalamus* Hsueh & Keng f., *Shibataea* Makino ex Nakai and *Sasa* Makino & Shibata form a clade that excludes *G. tessellatus* (*Zhang et al., 2016c*). In other few-gene plastid studies, relationships among taxa of clade IV are mostly unresolved (*Triplett & Clark, 2010*; *Attigala et al., 2014*).

The strongly supported clade VI (also called the *Arundinaria* clade) was similarly resolved in earlier plastome trees (*Ma et al., 2014*; *Attigala et al., 2016*). It includes subclades referred to as the Japan-North American clade (here including *Arundinaria* spp. and *Sasa veitchii*) and Sino-Japanese clade (*Pseudosasa japonica*, *Pleioblastus maculatus*, *Acidosasa purpurea*, *Indosasa sinica* and *Oligostachyum shiuyingianum*) (*Zhang et al., 2016c*). Support for the Japan-North American clade varies from weak to strong in all trees except R, in which the clade is not resolved because *S. veitchii* is sister to the rest of the subtribe. In other studies, support for the clade is moderate in maximum parsimony (MP) and ML trees based on plastomes and taxon sampling comparable to the current study (*Attigala et al., 2016*), and weak in MP and ML trees based on four plastid regions and denser taxon sampling (*Triplett & Clark, 2010*). The clade was not, however, identified in the eight-region plastid tree in *Zeng et al. (2010)*. The generally strong support for the

three species New World *Arundinaria* clade in the current and earlier plastome trees (*Burke et al., 2014*; *Attigala et al., 2016*), in which *Arundinaria tecta* and *Arundinaria appalachiana* are sister taxa, is an improvement on few-gene plastid trees, in which the three species do not form a clade (*Triplett & Clark, 2010*) or form a clade with support only in BI trees (*Zeng et al., 2010*) and in BI and MP trees (*Zhang, Zeng & Li, 2012*). The three species formed an unsupported clade in a nuclear GBSSI phylogeny, with a differing but weakly supported internal topology (*Zhang, Zeng & Li, 2012*). Recovery of the Sino-Japanese clade in all plastome trees, and placement of *P. japonica* as sister to the rest of the lineage (a four-taxon strongly supported clade), is congruent with other plastid trees (*Triplett & Clark, 2010*; *Zhang, Zeng & Li, 2012*; *Attigala et al., 2016*; *Zhou et al., 2016*) but not with a nuclear phylogeny (*Zhang, Zeng & Li, 2012*). In plastid trees with better taxon sampling, *Pseudosasa japonica* is part of a deep lineage referred to as the "Medake subclade" (*Triplett & Clark, 2010*; *Zeng et al., 2010*). The *Acidosasa purpurea* + *Indosasa sinica* + *Oligostachyum shiuyingianum* clade, strongly supported in two trees (R, W) and with *Pleioblastus maculatus* resolved as its sister group, is also identified in the plastome tree in *Zhang & Chen (2016)*, based on "complete cp genomes" (they did not indicate how they dealt with gapped sites, although there were likely fewer gapped sites in their alignments than ours because they analyzed only plastomes of bamboos), and in the complete plastome tree in *Ma et al. (2014)*, and congruent with the tree in *Attigala et al. (2016)*. In six trees, however, *P. maculatus* + *I. sinica* + *O. shiuyingianum* form a moderately to strongly supported clade (E–H, Y, Z). Neither of these conflicting clades is identified in other plastid trees (*Yang et al., 2013*; *Zhang et al., 2016c*). We are not aware of morphological characters that would support one of these competing topologies, as morphological variation of the genera of clade VI, none of which is monophyletic, is insufficiently known.

The deep lineage of Arundinarieae comprising clades III, V, VII and IX recovered in three complete plastome trees with weak to moderate support is weakly supported or unsupported in other multi-region plastid trees (*Triplett & Clark, 2010*; *Zeng et al., 2010*; *Yang et al., 2013*; *Zhang et al., 2016c*), and variously supported (depending on method of phylogenetic inference) to strongly supported in earlier plastome trees (*Ma et al., 2014*; *Attigala et al., 2016*). Placement of *T. spathiflorus* (clade VII) sister to the rest of the clade in nine plastome trees here is congruent with one earlier plastome tree (*Attigala et al., 2016*) but not the other, in which clades III + IX are sister to the rest of the clade (*Ma et al., 2014*). The deep placement of *T. spathiflorus* in the plastome trees here was not recovered in broadly sampled few-gene plastid trees (*Triplett & Clark, 2010*; *Zeng et al., 2010*; *Yang et al., 2013*; *Zhang et al., 2016c*). The sister relationship between *G. megalothyrsa* (clade IX) and *C. longiusculus* (clade III) in all plastome trees except the two based on noncoding data, one of which weakly supports clades III and VII as sister taxa, is congruent with other studies (*Zeng et al., 2010*; *Yang et al., 2013*; *Ma et al., 2014*; *Attigala et al., 2016*; *Zhang et al., 2016c*). In a better sampled plastome study also including clades II and XII, however, clades II, III, IX and XII form a clade and clades III and XII are sister taxa (*Attigala et al., 2016*).

The strongly supported clade V, the *Phyllostachys* clade (*Zeng et al., 2010*; *Kellogg, 2015*), has been recovered in other few-gene plastid and plastome trees with varying support

(*Triplett & Clark, 2010*; *Zeng et al., 2010*; *Yang et al., 2013*; *Ma et al., 2014*; *Attigala et al., 2016*; *Zhang et al., 2016c*), but not in a nuclear GBSSI phylogeny in which most deep branches of Arundinarieae are unresolved or poorly supported (*Zhang, Zeng & Li, 2012*). Most genera currently recognized in clade V are not monophyletic (*Kellogg, 2015*). Overall, resolution and support for relationships in clade V are better and stronger in the plastome trees here compared to few-gene plastid trees (*Triplett & Clark, 2010*; *Zeng et al., 2010*; *Zhang, Zeng & Li, 2012*; *Zhang et al., 2016c*). The varying support we find for *S. faberi* being sister to the rest of clade V is congruent with the plastome tree in *Ma et al. (2014)*, whereas its affinities in clade V are unresolved in the tree in *Triplett & Clark (2010)*. Affinities of *D. falcatum* (type species of the genus) within clade V are poorly supported in the plastome trees, as in previous studies of plastid and GBSSI sequences in which species of *Drepanostachyum* Keng f. and *Himalayacalamus* Keng. f., neither of which is monophyletic, form a clade of unresolved affinity within clade V (*Triplett & Clark, 2010*; *Zeng et al., 2010*; *Zhang, Zeng & Li, 2012*, 2016). The *D. falcatum* plastome is the first one sequenced for the genus, which comprises ten species from the Himalayan regions of Bhutan, China, India and Nepal (*Kellogg, 2015*). No plastomes have been published from *Himalayacalamus*, comprising eight species also from Bhutan, China, India and Nepal (*Kellogg, 2015*). Relationships among the remaining taxa of clade V are mostly congruent with those found by *Ma et al. (2014)*. The five species of *Phyllostachys* included here have not previously been combined in a phylogenetic analysis. They form a clade in all plastome trees, an improvement compared to few-gene trees here and elsewhere (*Triplett & Clark, 2010*; *Zeng et al., 2010*; *Zhang et al., 2016c*). Within *Phyllostachys*, the affinities of *P. edulis*, a species that grows rapidly and is of critical ecological, economic and cultural value in Asia (*Peng et al., 2013*), vary among analyses.

*Bambuseae*

Two major clades have been identified in Bambuseae: the paleotropical woody bamboo clade and the neotropical woody bamboo clade (*Kelchner & Bamboo Phylogeny Group, 2013*; *Zhang et al., 2016c*). Our plastome sampling in the paleotropical woody bamboo clade represents four of the eight subtribes that are part of the lineage (*Sungkaew et al., 2009*; *Goh et al., 2010*; *Kelchner & Bamboo Phylogeny Group, 2013*; *Wong et al., 2016*; *Zhang et al., 2016c*). Plastomes representing subtribes Racemobambosinae, Holttumochloinae and Temburongiinae have not yet been published. The sister group relationship between Dinochloinae and Greslaninae and the monophyly of Bambusinae are congruent with earlier plastome trees (*Wysocki et al., 2015*) and with better sampled few-gene plastid trees (*Yang et al., 2008*; *Sungkaew et al., 2009*; *Chokthaweepanich, 2014*; *Zhou et al., 2017*). However, relationships among Hickeliinae, Bambusinae, Dinochloinae and Greslaninae vary among our plastome trees, as they do among other studies. Relationships among these lineages in an 18-region plastid tree are unresolved (*Zhou et al., 2017*). Hickeliinae forms a strongly supported clade with Dinochloinae + Greslaninae in seven trees (E–H, Q, W, Y), a topology found in an earlier plastome study (*Wu et al., 2015*), and congruent with a six-gene plastid tree in which Dinochloinae is not sampled (*Zhang et al., 2016c*). On the other hand, Hickeliinae is sister to a strongly supported Bambusinae + Dinochloinae +

Greslaninae clade in three trees (R, X, Z), a topology recovered with moderate support in analyses of a plastome matrix equivalent to X here (*Wysocki et al., 2015*) and of plastomes excluding gapped sites (*Vieira et al., 2016*). Plastome sampling from the eight additional genera included in Hickeliinae may help resolve ambiguity in affinities of the subtribe within Bambuseae, as well as from Melocanninae, Racemobambosinae, Temburongiinae and Holttumochloinae, the subtribes of the paleotropical woody bamboo clade not represented in our plastome trees.

The neotropical woody bamboo clade includes subtribes Chusqueinae, Arthrostylidiinae and Guaduinae. Placement of Chusqueinae (*Chusquea*) sister to a maximally supported clade comprising Arthrostylidiinae and Guaduinae is congruent with other studies (*Sungkaew et al., 2009*; *Kelchner & Bamboo Phylogeny Group, 2013*; *Chokthaweepanich, 2014*; *Vieira et al., 2016*; *Zhang et al., 2016c*). Within Guaduinae, *Olmeca* Soderstr. and *Otatea* (McClure & E.W. Sm.) C.E. Calderón & Soderstr. are strongly supported sister taxa in most trees, but in tree R, *Olmeca* and *Guadua* Kunth form a moderately supported clade. The former topology was previously identified in a plastome study (*Wu et al., 2015*) and is congruent with a few-gene plastid study (*Ruiz-Sanchez, Sosa & Mejia-Saules, 2011*). Relationships among the four samples of *Chusquea* are congruent with a more detailed study of *Chusquea* phylogeny (*Fisher, Clark & Kelchner, 2014*). *Chusquea spectabilis* used to be included in *Neurolepis* Meisn., a genus that lacked elongated woody culms of *Chusquea* as historically applied (*Fisher et al., 2009*).

### Pooideae

#### Relationships among subtribes

The successive divergences of Brachyelytreae, Lygeae + Nardeae, Phaenospermateae, Meliceae and Stipeae (including Ampelodesmeae) with respect to the rest of the subfamily in the plastome trees are congruent with previous few-gene plastid studies, and the robust support in the plastome trees for the respective branches is in many instances stronger than in few-gene trees (*Catalán, Kellogg & Olmstead, 1997*; *Soreng & Davis, 1998*, *2000*; *Mathews, Tsai & Kellogg, 2000*; *Grass Phylogeny Working Group, 2001*; *Davis & Soreng, 2007*, *2010*; *Döring et al., 2007*; *Duvall et al., 2007*; *Bouchenak-Khelladi et al., 2008*; *Schneider et al., 2011*; *Grass Phylogeny Working Group II, 2012*; *Blaner, Schneider & Röser, 2014*; *Hochbach, Schneider & Röser, 2015*; *Pimentel et al., 2017*). Morphological synapomorphies supporting most of these deep splits in Pooideae have been identified (*Kellogg et al., 2013*; *Kellogg, 2015*). Of the early diverging lineages of Pooideae, the only tribes from which plastomes have not been sampled are Brylkinieae and Duthieae (*Soreng et al., 2017*). The three plastomes of Stipeae newly sampled here (*Eriocoma hymenoides*, *Nassella hyalina*, *Piptatherum songaricum*) form a clade sister to a lineage of *Oryzopsis asperifolia*, *Ampelodesmos mauritanicus* (Ampelodesmeae), which is a polyploid reticulate species with Duthieeae and Stipeae that obtained its plastome from a stipoid grass (*Romaschenko et al., 2014*), and *Piptochaetium avenaceum*. The strongly supported relationships within both clades are congruent with a tree based on fewer plastid regions but denser taxon sampling (*Romaschenko et al., 2012*).

Tribes Brachypodieae, Diarrheneae, Bromeae, Poeae and Triticeae form a maximally supported clade in all trees here, as in numerous other studies of plastid and nuclear ribosomal DNA (*Catalán, Kellogg & Olmstead, 1997*; *Davis & Soreng, 2007*; *Bouchenak-Khelladi et al., 2008*; *Schneider et al., 2011*; *Pimentel et al., 2017*; *Sancho et al., 2017*). Diarrheneae is unique in the clade in having nondistichous two-ranked inflorescence phyllotaxy, a character-state reversion in this taxon following the origin of distichous phyllotaxis in the ancestor of the clade including Phaenospermateae and the rest of the subfamily (*Kellogg et al., 2013*). The maximally supported relationships among Bromeae, Poeae and Triticeae are congruent with a recent plastome study (*Saarela et al., 2015*) and numerous few-gene plastid studies. Monophyly of Triticeae is maximally supported in all plastome trees here. However, when plastid data for *Psathyrostachys* Nevski are included in analysis, Triticeae is paraphyletic because Bromeae is included within it (*Bernhardt et al., 2017*). We have not sampled the monogeneric tribe Littledaleae (*Soreng et al., 2017*), which is sister to Bromeae + Triticeae in plastid trees (*Soreng, Davis & Voionmaa, 2007*; *Schneider, Winterfeld & Röser, 2012*) and sister to Triticeae in nuclear trees (*Hochbach, Schneider & Röser, 2015*). A plastome from a species of *Littledalea* Hemsl. was recently published (*Liu et al., 2017*).

Clarification of the evolutionary placement of *Brachypodium* P. Beauv. within pooids is important because the annual species *B. distachyon* is a model system for grasses (*International Brachypodium Initiative, 2010*). Relationships and support levels among Diarrheneae, Brachypodieae and Bromeae + Poeae + Triticeae vary among plastome trees and are affected particularly by inclusion or exclusion of gapped sites in the noncoding data partition. Relationships among these taxa inferred from different plastome partitions were similarly variable in an earlier plastome study, which also found differences among ML, BI and MP trees (*Saarela et al., 2015*). Presence of parallel-sided subsidiary cells is a putative synapomorphy for a Brachypodieae + Bromeae + Poeae + Triticeae clade (*Kellogg, 2015*), found in a subset of our trees. Some analyses of low copy nuclear genes also identify a Brachypodieae + Bromeae + Poeae + Triticeae clade (*Hochbach, Schneider & Röser, 2015*), but in others relationships among these lineages are either unresolved or *Diarrhena* and *Brachypodium* are sister taxa (*Hochbach, Schneider & Röser, 2015*; *Minaya et al., 2015*), like in the trees here based on complete plastomes excluding gapped sites. In a recent study including plastomes from three *Brachypodium* species, *Diarrhena*, *Brachypodium*, and Bromeae + Poeae + Triticeae diverged successively, with strong support for the topology (*Sancho et al., 2017*). Those trees were based on a dataset that excluded poorly aligned regions but included "robust gaps."

Variation in topology and support among Diarrheneae, Brachypodieae and Bromeae + Poeae + Triticeae in the plastome trees might be related to the long branch subtending *B. distachyon*—the longest one in Pooideae in our trees—relative to the lengths of nearby branches (Fig. S1). This long branch might be attributable to one or a combination of an accelerated plastome substitution rate in the genus or the one annual species of the genus we sampled, long persistence of the lineage since its divergence from the common ancestor it shares with its sister group, or extinction(s) of closely related taxa (none of which is known). Substitution rates in plastid coding regions are significantly lower in

lineages of *Triticum* and *Aegilops* compared to *Brachypodium* (*Gornicki et al., 2014*), supporting an accelerated rate of evolution along the *Brachypodium* branch. Combined plastid and nuclear ribosomal data do not support an older age for the *Brachypodium* crown clade than for the Poeae + Triticeae crown clade (*Catalán et al., 2012*), whereas in a plastome-based nested dating analysis of the grass family and *Brachypodium*, the ages of the *Brachypodium* and Poeae + Bromeae–Triticeae crown clades were estimated at 10.1 Ma and 27.8 Ma, respectively (*Sancho et al., 2017*). Plastomes from the other two annual species of *Brachypodium*, when analyzed phylogenetically with *B. distachyon* in analyses including some gapped sites in the alignment, resulted in a slightly shortened stem branch of the *Brachypodium* clade, which may have contributed to the strongly resolved relationships among Brachypodieae, Diarrheneae and Bromeae–Poeae–Triticeae in those analyses (*Sancho et al., 2017*). Plastomes from the perennial species *Brachypodium mexicanum* (Roem. & Schult.) Link and *Brachypodium boissieri* Nyman, based on their affinities to the annual species and the core perennial clade in a two-gene plastid tree (*Catalán et al., 2012*, *2016*), might further break up the long stem branch, which may help clarify relationships among Brachypodieae, Diarrheneae and Bromeae–Poeae–Triticeae from different plastome partitions. Plastome sampling of *Neomolinia* Honda, the other genus of Diarrheneae, might also help clarify these relationships. *Neomolinia*, with five species and sometimes treated as a synonym of *Diarrhena* (*Kellogg, 2015*), has been sampled in only three studies (*Schneider et al., 2011*; *Romaschenko et al., 2012*; *Hochbach, Schneider & Röser, 2015*).

*Triticeae*

Within Triticeae, the strongly supported successive divergences of *Hordeum, Connorochloa tenuis, Secale cereale, Taeniatherum caput-medusae* and *Aegilops/Triticum* in the plastome trees are congruent with (or at least not in conflict with) and better supported than few-gene plastid trees (*Petersen & Seberg, 1997*; *Mason-Gamer, Orme & Anderson, 2002*; *Petersen et al., 2006*; *Seberg & Petersen, 2007*) and some nuclear trees (*Mason-Gamer, 2001*; *Petersen et al., 2006*; *Escobar et al., 2011*). However, conflict is well known among plastid and nuclear trees, reflecting hybridization in the origins of many genera and species in the tribe (*Petersen & Seberg, 2002*; *Mason-Gamer, 2005*; *Petersen et al., 2006*; *Seberg & Petersen, 2007*; *Escobar et al., 2011*). The plastome topology here is congruent with the plastome tree in *Bernhardt et al. (2017)*. *C. tenuis*, an octoploid endemic to New Zealand and the only species in its genus (*Barkworth, Jacobs & Zhang, 2009*), has apparently not been included in any previous phylogenetic study.

   Our sampling includes plastomes from additional species of *Aegilops* and *Triticum* (*Gornicki et al., 2014*; *Gogniashvili et al., 2015*) compared to an earlier study (*Saarela et al., 2015*). Although all species of *Aegilops* and *Triticum* form a clade in the plastome trees, other studies have demonstrated that genera not sampled here (e.g., *Amblyopyrum* (Jaub. & Spach) Eig, *Thinopyrum* Á. Löve, *Lophopyrum* Á. Löve, *Crithopsis* Jaub. & Spach.) are part of the lineage (*Petersen et al., 2006*). Despite our incomplete genus-level sampling of the *Aegilops* and *Triticum* lineage, recovery of major subclades in the lineage is congruent with other studies (*Petersen et al., 2006*; *Gornicki et al., 2014*; *Bernhardt et al., 2017*).

The *A. speltoides* + *T. timopheevii* sublineage has been found in other studies (*Golovnina et al., 2007*; *Gornicki et al., 2014*; *Gogniashvili et al., 2015*), as has the lineage comprising *T. macha* (=*T. aestivum* subsp. *macha* (Dekapr. & Menabde) MacKey), *T. turgidum* and *T. aestivum* "Chinese Spring," whose relationships conflict strongly among the plastome trees. Close relationships among multiple subspecies of *T. turgidum* (sometimes recognized at species level) and *T. aestivum* "Chinese Spring" were found in earlier plastome (*Gornicki et al., 2014*) and nuclear (*Petersen et al., 2006*; *Nasernakhaei et al., 2015*) trees, but none of those studies sampled the Georgian endemic *T. macha*. *Golovnina et al. (2007)* sampled one individual of *T. macha* in their *matK* tree, which was identical to multiple other taxa, including *Triticum durum* Desf. and *T. aestivum*, congruent with our results. The subclade comprising *T. urartu* + *T. monococcum* and the eight-species *Aegilops* clade is congruent with other plastid trees with similar sampling (*Petersen et al., 2006*; *Golovnina et al., 2007*; *Gornicki et al., 2014*; *Middleton et al., 2014*; *Gogniashvili et al., 2015*). The sister relationship between *Aegilops cylindrica* Host and *Aegilops tauschii* Coss. has been found in other studies (*Middleton et al., 2014*; *Gogniashvili et al., 2015*), as has the five-species clade comprising *Aegilops bicornis* (Forssk.) Jaub. & Spach, *A. sharonensis*, *A. longissima*, *A. kotschyi* and *Aegilops searsii* Feldman & Kislev (*Gornicki et al., 2014*).

*Poeae*

The maximally supported clades in the plastome trees recognized as Poeae chloroplast groups 1 and 2 have been recovered in other plastid-based studies (*Quintanar, Castroviejo & Catalán, 2007*; *Saarela et al., 2010*, *2015*, *2017*; *Pimentel et al., 2017*; *Sancho et al., 2017*), but not in studies based on nuclear ribosomal DNA, in which Scolochloinae and Sesleriinae, both part of Poeae chloroplast group 2 and not sampled here, are closely related to taxa of Poeae chloroplast group 1 (*Quintanar, Castroviejo & Catalán, 2007*; *Saarela et al., 2010*, *2015*, *2017*). Six of the eight subtribes of Poeae chloroplast group 1 (*Soreng et al., 2017*) are represented in our trees. Relationships among the four taxa of Agrostidinae and the sister-group relationship between Agrostidinae and Brizinae in the plastome trees are congruent with other plastid studies, with the caveat that studies with broader sampling of these subtribes and related taxa have identified problems with generic circumscriptions and conflicts between plastid and nuclear data (*Quintanar, Castroviejo & Catalán, 2007*; *Soreng, Davis & Voionmaa, 2007*; *Saarela et al., 2017*). The maximally supported sister group relationship between Anthoxanthinae and Agrostidinae + Brizinae is also congruent with other plastome and few-gene plastid studies (*Saarela et al., 2015*, *2017*). In a recent five region plastid study, however, Anthoxanthinae is strongly supported as sister to Aveninae/Koeleriinae and *Lagurus* L. (*Pimentel et al., 2017*), a topology conflicting with our results.

The sister-group relationship between Phalaridinae and Torreyochloinae was first identified in a previous plastome study (*Saarela et al., 2015*); however, these subtribes are not sister taxa in combined ITS + ETS trees, possibly indicative of ancient hybridization (*Saarela et al., 2017*). The major conflict in the relative branching order of Phalaridinae + Torreyochloinae and Aveninae at the base of Poeae chloroplast group 1 in the plastome trees was also found in a previous plastome study (*Saarela et al., 2015*), but

in that study the different topologies were inferred in ML and BI vs. MP analyses rather than among plastome partitions, as is the case here. The phylogenetic signal for Phalaridinae + Torreyochloinae being sister to the rest of the clade is present in plastome noncoding data, regardless of whether gapped sites are included or excluded, whereas phylogenetic signal for Aveninae being sister to the rest of the clade is present primarily in plastome coding data including and excluding gapped sites. The latter topology is also identified in trees based on complete plastomes, both including and excluding gapped sites. In complete plastome trees, when gapped sites are excluded and positively selected sites are included in the analysis, the branching order at the base of the subtree is ambiguous, whereas when both gapped and positively selected sites are excluded in complete plastome trees, Phalaridinae + Torreyochloinae are strongly supported as sister to the rest of the clade. These differences indicate the presence of some conflicting signal in positively selected sites of plastome coding regions that affect support levels when gapped sites are excluded. The latter is confirmed by the decrease in support for Aveninae being sister to the rest of the clade in analyses of plastome coding regions including and excluding positively selected sites (E vs. G, BP = 98% vs. 80%).

Poeae chloroplast group 2 comprises 18 subtribes and numerous genera unplaced to subtribe (*Soreng et al., 2017*), and twelve subtribes are represented in the current plastome sampling. The major clade comprising *Puccinellia nuttalliana* + *Zingeria biebersteiniana* (Coleanthinae), and *Alopecurus arundinaceus* (Alopecurinae) + *Phleum alpinum* (Phleinae) + *Poa palustris* (Poinae) has been identified in other plastid studies (*Gillespie, Archambault & Soreng, 2007*; *Gillespie et al., 2008*; *Soreng, Davis & Voionmaa, 2007*; *Schneider, Winterfeld & Röser, 2012*; *Hochbach, Schneider & Röser, 2015*), but relationships among Alopecurinae, Poinae and Phleinae are discordant among the plastome trees. The conflict is primarily between the noncoding partition, which identifies *Phleum alpinum* and *A. arundinaceus* as sister taxa (also in one complete plastome analysis including gapped sites) and all other partitions, which identify *Poa palustris* and *A. arundinaceus* as strongly supported sister taxa. Although it is unclear which of the two highly supported topologies is accurate, there is sufficient variation in complete plastomes to robustly resolve relationships among these closely related genera compared to earlier plastid studies in which relationships among clades including these three genera were unresolved and/or poorly supported (*Gillespie, Archambault & Soreng, 2007*; *Gillespie et al., 2008*; *Soreng, Davis & Voionmaa, 2007*). *Poa* L. and *Phleum* L. were more closely related to each other than to *Alopecurus* L. in analyses of combined plastid and nuclear ribosomal data (*Gillespie et al., 2010*; *Soreng et al., 2015a*), a topology probably influenced by the nuclear ribosomal signal in that dataset. None of the plastome trees identify a *Poa* + *Phleum* clade.

Holcinae (*Holcus lanatus*) and Airinae (*Helictochloa hookeri*) are strongly supported sister taxa in three plastome trees. This topology conflicts with the combined ITS and plastid tree in *Minaya et al. (2015)*, in which *Helictochloa bromoides* (Gouan) Romero-Zarco (as *Avenula bromoides* (Gouan) H. Scholz) is sister to a Dacytilidinae + Cynosurinae clade, and *Holcus* L. + *Echinaria* Desf. (Sesleriinae) are sister to a clade including taxa of Airinae and *Deschampsia* P. Beauv. The plastome topology also conflicts with the β-amylase tree in *Minaya et al. (2015)*, indicative of reticulation. Although

*Helictochloa* Romero-Zarco is currently classified in subtribe Airinae, few-gene plastid and nuclear analyses indicate the genus is not allied with other taxa of the subtribe (*Quintanar, Castroviejo & Catalán, 2007*; *Saarela et al., 2017*). Plastome sampling is needed of the other genera included in Airinae (*Aira* L., *Antinoria* Parl., *Avenella* Drejer, *Corynephorus* P. Beauv., *Molineriella* Rouy, *Periballia* Trin.) to clarify circumscription of the subtribe.

In a previous classification (*Soreng et al., 2015b*), Holcinae comprised *Deschampsia*, *Holcus* and *Vahlodea* Fr., but in the plastome trees here and in other plastid and nuclear trees, *Deschampsia* and *Holcus* + *Vahlodea* (not sampled here) do not form a clade (*Quintanar, Castroviejo & Catalán, 2007*; *Saarela et al., 2010*, *2017*; *Grass Phylogeny Working Group II, 2012*; *Minaya et al., 2015*; *Persson & Rydin, 2016*). In the plastome trees, *Deschampsia* is sister to a clade comprising taxa of Cynosurinae, Dactylidinae, Parapholiinae and Loliinae. Accordingly, *Deschampsia* is now recognized in its own subtribe, Aristaveninae, and Holcinae is circumscribed more narrowly comprising only *Holcus* and *Vahlodea* (*Soreng et al., 2017*).

Relationships among the remaining six sampled subtribes of Poeae chloroplast group 2 are robustly resolved here. The close relationship between *D. glomerata* and *L. aurea*, both included in Dacytilidinae, is congruent with other plastid and nuclear analyses (*Inda et al., 2008*; *Birch et al., 2014*; *Minaya et al., 2015*). The sister group relationship between Cynosurinae (*Cynosurus* L., monotypic) and Parapholiinae (eight genera, represented by *Catapodium rigidum*) is congruent with earlier plastid and nuclear ribosomal analyses (*Inda et al., 2008*; *Schaefer et al., 2011*; *Schneider, Winterfeld & Röser, 2012*; *Pimentel et al., 2017*) with denser sampling of Parapholiinae. The sister-group relationship between Dactylidinae and Cynosurinae + Parapholiinae corroborates the findings of earlier studies (*Inda et al., 2008*; *Birch et al., 2014*). The strongly supported Cynosurinae + Dactylidinae + Parapholiinae + Loliinae clade in the plastome trees is congruent with a *matK* tree (*Schneider, Winterfeld & Röser, 2012*) and is an improvement on the mostly unresolved and poorly supported relationships among these taxa in other plastid trees (*Quintanar, Castroviejo & Catalán, 2007*; *Soreng, Davis & Voionmaa, 2007*; *Pimentel et al., 2017*). In an earlier plastome study, *Dactylis* L. was weakly supported as sister to Loliinae in a ML tree based on plastome coding regions (*Saarela et al., 2015*), whereas in the tree based on the parallel plastome coding dataset here (F), the same branch is strongly supported. This increased support may be a function of the improved taxon sampling in Dactylidinae, Cynosurinae and Parapholiinae here. Sampling and relationships within Loliinae here and in an earlier plastome study (*Saarela et al., 2015*) are identical, although here we have updated names of some species to reflect their current classification.

### Oryzoideae

Subfamily Oryzoideae is divided into tribes Streptogyneae, Ehrharteae, Phyllorachideae and Oryzeae (*Soreng et al., 2017*), which are each represented in our analyses.

#### Streptogyneae

Clarification of the evolutionary affinities of the amphi-Atlantic genus *Streptogyna* P. Beauv. has been problematic. *Streptogyna* was traditionally classified as a herbaceous

bamboo in its own tribe, Streptogyneae (*Calderón & Soderstrom, 1980*; *Soderstrom & Judziewicz, 1987*). It shares several morphological characters with various distantly related lineages (*Kellogg, 2015*), interpretation of which has complicated classification. Molecular studies have helped clarify its affinities. In studies based primarily on one or a few plastid regions, *Streptogyna* was sister to Oryzoideae with varying levels of support (*Clark, Zhang & Wendel, 1995*; *Zhang, 2000*; *Duvall et al., 2007*; *Davis & Soreng, 2010*; *Triplett & Clark, 2010*; *Kelchner & Bamboo Phylogeny Group, 2013*). *Kelchner & Bamboo Phylogeny Group (2013)* identified a high level of character conflict in *Streptogyna* at "key nodes" in a neighbour net analysis of a five-region plastid data set, despite strong support for its placement in their tree. In recent classifications, *Streptogyna* has been treated as *incertae sedis* among grasses (*Grass Phylogeny Working Group, 2001*), as *incertae sedis* within the BOP clade (*Kellogg, 2015*) and as a tribe of Oryzoideae (*Soreng et al., 2015b*, *2017*). *Streptogyna* differs from other oryzoids by having multi-flowered (vs. one-flowered) spikelets. Our results corroborate previous support for the monophyly of Oryzoideae including *Streptogyna*, as the subfamily is maximally supported in all plastome trees except the two based on noncoding regions, in which support for the same topology is lower (BP = 87–88%), indicative of some conflict in the noncoding partition relative to the rest of the plastome. Nevertheless, there is robust support in the plastome trees for *Streptogyna* being sister to the rest of the subfamily.

The affinities of *Streptogyna* are different, however, in nuclear trees. In a phylogeny based on phytochrome B, *Streptogyna* is sister to the BOP clade (*Mathews, Tsai & Kellogg, 2000*), and in a phylogeny based on FLOWERING LOCUS T (FT) in which both species of *Streptogyna* were sampled, the genus is monophyletic and sister to Bambusoideae with moderately strong support, and an Oryzoideae + (*Streptogyna* + Bambusoideae) clade is weakly supported (*Hisamoto, Kashiwagi & Kobayashi, 2008*); no pooid taxa were included in that study. Given the discordances among nuclear and plastome phylogenies, it is possible *Streptogyna* might have arisen as part of an ancient hybridization event involving a maternal parent ancestral to crown Oryzoideae and a paternal parent ancestral to crown Bambusoideae. The evolutionary patterns identified in the nuclear trees might alternatively reflect incomplete lineage sorting. Further sampling of the nuclear genome of *Streptogyna*, bamboos and other oryzoids will be required to further characterize the history of this lineage of grasses.

*Ehrharteae and Phyllorachideae*
Ehrharteae includes four genera: *Ehrharta* Thunb., *Microlaena* R. Br., *Tetrarrhena* R. Br. and *Zotovia* Edgar & Connor (*Soreng et al., 2017*), all of which are sometimes included in a single genus, *Ehrharta* (*Kellogg, 2015*), a classification congruent with phylogenetic data (*Verboom et al., 2003*). As expected, the two species we sampled, *E. bulbosa* and *M. stipoides* (= *Ehrharta stipoides* Labill.), form a clade, and this clade is robustly placed as sister to the rest of the subfamily except *Streptogyna*. This topology is congruent with other plastid trees, although not all included *Streptogyna* (*Grass Phylogeny Working Group, 2001*; *Bouchenak-Khelladi et al., 2008*; *Grass Phylogeny Working Group II, 2012*).

The poorly known tribe Phyllorachideae comprises two genera: *Humbertochloa* A. Camus & Stapf, with two species from Madagascar and Tanzania, and *Phyllorachis* Trimen, with one species from equatorial Africa (*Kellogg, 2015*; *Soreng et al., 2017*). *Humbertochloa* has been included in only two molecular studies (*Zhang, 2000*; *Vorontsova et al., 2016*), neither of which have sufficient taxon sampling from which to draw conclusions about its affinities to other rice grasses, and *Phyllorachis* has not been sampled in any molecular studies. In our trees, *Humbertochloa* is maximally supported as sister to the Oryzeae clade. This topology is congruent with recognition of the lineage at either tribal rank within Oryzoideae or subtribal rank within Oryzeae, with the caveat that the affinities of *Phyllorachis sagittata* Trimen are unknown.

*Oryzeae*

Clades corresponding to Oryzeae and the subtribes Oryzinae and Zizaniinae (*Ge et al., 2002*; *Guo & Ge, 2005*; *Tang et al., 2010*; *Soreng et al., 2017*) are maximally supported in all plastome trees. Oryzinae includes four genera: *Oryza*, *Leersia* Sw., *Maltebrunia* Kunth and *Prosphytochloa* Schweick. (*Tang et al., 2010*; *Kellogg, 2015*; *Soreng et al., 2017*). An earlier classification (*Soreng et al., 2015b*) placed *Maltebrunia* and *Prosphytochloa* in Zizaniinae, based on phylogenies in which a sample identified as *Prosphytochloa prehensilis* was resolved as part of the Zizaniinae clade (*Ge et al., 2002*; *Guo & Ge, 2005*). However, that *Prosphytochloa* sample was later re-determined as a species of *Potamophila* R. Br. (*Tang et al., 2010*). *Prosphytochloa* and *Leersia* are maximally supported sister taxa in the plastome trees, and this clade is sister to *Oryza*, congruent with a plastid tree in which *Maltebrunia* and *Prosphytochloa* are sister taxa and a *Maltebrunia* + *Prosphytochloa* + *Leersia* clade is sister to *Oryza* (*Tang et al., 2010*).

Since rice (*Oryza*) is the most important food crop worldwide, there has been extensive phylogenetic research on the ca. 22 wild and two cultivated species of the genus. Although the 11 plastomes included here have all been published elsewhere, they have not all been combined in a single phylogenomic analysis. The relationships among the species of *Oryza* in our plastome trees are mostly congruent with similar plastome trees (*Kim et al., 2015*; *Liu et al., 2016*), although when multiple individuals of *O. nivara*, *O. sativa*, *O. rufipogon*, *Oryza barthii* A. Chev. and *Oryza glaberrima* Steud. were sampled none of the species was monophyletic (*Kim et al., 2015*). Placement of *Oryza australiensis* Domin sister to the rest of the genus is congruent with the neighbor joining tree in *Liu et al. (2016)*, in which *Oryza brachyantha* A. Chev. & Roehr.—the most distant congeneric relative of cultivated rice—and *O. australiensis* are successive sisters to the rest of the genus, a topology congruent with studies based on other types of data (*Ge et al., 1999*). Plastome data are useful for robustly resolving relationships among closely related species of *Oryza* with limited conflict among partitions.

Zizaniinae includes seven genera (*Soreng et al., 2017*) and our sampling includes four of these. The sister group relationship between *Rhynchoryza subulata* and *Zizania aquatica* in the plastome trees is congruent with results of other studies (*Tang et al., 2010*, *2015*), but the varying branching order of *Chikusichloa aquatica* and *Potamophila parviflora* at the base of the clade is a novel result. Successive branching of *Chikusichloa* Koidz. and

*Potamophila*, found in six trees, is congruent with an ML tree based on 20 plastid regions (*Tang et al., 2010, 2015*). The plastome partitions that identify *Potamophila* and *Chikusichloa* as successively diverging lineages comprise noncoding regions including gapped sites, either alone or in combination with coding regions. The dominant signal for this topology (whether accurate or not) is present in the gapped sites of the noncoding alignment, which when combined with coding region data seemingly override the conflicting signal in the latter partition. Recent data from the nuclear genome provides further insight into the Zizaniinae evolutionary tree, even though the nuclear trees are discordant, in part, with plastome trees. In trees based on 15 individual nuclear genes, relationships among taxa of Zizaniinae varied considerably and were strongly discordant with each other and with the plastid topologies here and elsewhere, whereas when the same 15 genes were analyzed together, *Chikusichloa* and *Potamophila* formed a clade sister to the rest of the subtribe (*Tang et al., 2015*). No plastome trees here identify a *Chikusichloa* + *Potamophila* clade. This discordance between plastid and nuclear trees might be due to incomplete lineage sorting, introgression, or both (*Tang et al., 2015*).

### PACMAD clade

Although the PACMAD clade has been consistently identified in molecular studies, relationships among the subfamilies have been generally poorly resolved and weakly supported, and identifying the root of the PACMAD clade—placement of the branch defining the first or deepest split in the lineage—has proven particularly challenging (*Clark, Zhang & Wendel, 1995*; *Mathews, Tsai & Kellogg, 2000*; *Grass Phylogeny Working Group, 2001*; *Duvall et al., 2007*; *Bouchenak-Khelladi et al., 2008*; *Davis & Soreng, 2010*; *Saarela & Graham, 2010*; *Grass Phylogeny Working Group II, 2012*). Numerous studies, mostly of plastid data, have identified Aristidoideae as the sister group of the rest of the PACMAD clade, but support for this topology (i.e., for the subclade including all PACMAD subfamilies except Aristidoideae) has mostly been weak (*Clark, Zhang & Wendel, 1995*; *Hilu, Alice & Liang, 1999*; *Grass Phylogeny Working Group, 2001*; *Duvall et al., 2007*; *Sánchez-Ken & Clark, 2007*; *Christin et al., 2008a*; *Grass Phylogeny Working Group II, 2012*); for an exception see *Vicentini et al. (2008)*. *Cotton et al. (2015)* explored relationships among the PACMAD subfamilies based on complete plastomes and identified two strongly conflicting topologies at the base of the clade. In their ML and BI trees, Panicoideae were moderately to strongly supported as the sister group of the rest of the PACMAD clade (the "panicoid-sister" hypothesis)—this was an unexpected topology not recovered in previous studies. In their MP tree, however, Aristidoideae were strongly supported as the sister group to the rest of the PACMAD clade (the "aristidoid-sister" hypothesis), congruent with most earlier studies of grasses. In a subsequent plastome study with increased Panicoideae taxon sampling but fewer representatives of most other PACMAD subfamilies, *Burke et al. (2016b)* also identified the panicoid-sister topology. By contrast, in a plastome study focused on Arundinoideae and with broad sampling across Poaceae, *Teisher et al. (2017)* recovered all three possible topologies for the base of PACMAD, of which none was particularly well supported. For example, they recovered the aristidoid-sister topology when gapped sites were included, with weak support in ML and

BI trees. *Teisher et al. (2017)* concluded that plastome data may be insufficient to resolve this particular set of relationships. The plastome phylogeny of the PACMAD clade generated by *Piot et al. (2018)* cannot be used to address deep relationships in the clade because they rooted their tree with Aristidoideae.

Relationships at the base of the PACMAD clade in our 14 trees similarly vary in topology and support. The same two conflicting topologies are identified in seven of our trees, each with moderate to strong support in at least one of these trees. Three trees excluding gapped sites identify the panicoid-sister topology (D, R, Z), similar to gap-stripped results in *Teisher et al. (2017)*, three trees including gapped sites identify the aristidoid-sister topology (Q, W, Y), and in one tree (F), Aristidoideae + Panicoideae are weakly supported as sister to the rest of the PACMAD clade, a topology rarely inferred elsewhere. Our three-gene trees parallel the plastid sampling of *Grass Phylogeny Working Group II (2012)* (partition A is most similar to their dataset), who identified the aristidoid-sister topology with weak support in ML trees and maximum support in BI trees, a topology congruent with the two three-gene trees including gapped sites reported here. Overall, the aristidoid-sister topology is solely recovered in matrices in which gapped sites were not stripped, suggesting that the signal for this topology is largely in the gapped regions.

It is surprising that the two trees of complete plastomes excluding gapped sites provide no (BP < 50%; X) or only weak (BP = 58%; Z) support for the panicoid-sister topology because (1) this partition includes noncoding regions excluding gapped sites (dataset R), which, when analyzed separately, strongly support the panicoid-sister topology; and (2) this partition includes coding regions that, when analyzed separately, do not provide support greater than 50% for any particular topology. Nevertheless, it is possible there is discordant signal in coding regions that might be contributing to reduced support for the panicoid-sister topology when combined with noncoding regions excluding gapped sites. Indeed, the alternative topology in tree F, even though only weakly supported, supports the idea of discordant signal in coding regions. Furthermore, the lack of support greater than 50% for any topology in tree X differs from the ML tree in *Cotton et al. (2015)* based on an equivalent dataset, in which the panicoid-sister topology receives moderate support (BP = 77%). These differences might be related to the denser taxon sampling here in Panicoideae and particularly in Aristidoideae, in which we sampled two species each of *Aristida* L., *Sartidia* De Winter and *Stipagrostis* Nees, compared to *Cotton et al. (2015)*, as well as possible alignment differences.

Choice of outgroup in a phylogenetic analysis can affect inferences of the location of the root of a clade (*Graham, Olmstead & Barrett, 2002*; *de la Torre-Bárcena et al., 2009*), especially in clades such as the PACMAD clade with short deep internodes that are difficult to resolve. *Cotton et al. (2015)* tested the effects of including different and varying numbers of non-PACMAD grass outgroups on the basal topology of the PACMAD clade in ML analyses. Although the panicoid-sister topology was supported in all but one of their experiments, BP for this topology ranged considerably (from 60% to 91%) indicating some effect of outgroup on ingroup branch support. We do not attribute our

conflicting topologies to outgroup-effect, since all analyses include the same broad sampling of taxa of Poaceae, which exceeds the diversity in any other plastome analysis.

In spite of the uncertain branching order of Aristidoideae and Panicoideae with respect to the rest of the PACMAD clade, the four remaining PACMAD subfamilies—Arundinoideae, Micrairoideae, Chloridoideae and Danthonioideae—form a strongly to maximally supported clade in the plastome trees. This clade has been identified in previous studies (*Duvall et al., 2007*, *2010*; *Grass Phylogeny Working Group II, 2012*; *Cotton et al., 2015*; *Burke et al., 2016b*; *Piot et al., 2018*; *Teisher et al., 2017*). The strongly supported sister group relationship between Chloridoideae and Danthonioideae has similarly been found in other plastid studies, with weak (*Hilu & Alice, 1999*; *Duvall et al., 2007*; *Sánchez-Ken & Clark, 2007*; *Christin et al., 2008a*) or moderate to strong support (*Grass Phylogeny Working Group, 2001*; *Bouchenak-Khelladi et al., 2008*; *Peterson, Romaschenko & Johnson, 2010*; *Piot et al., 2018*; *Teisher et al., 2017*). These relationships were also found in a combined ITS and plastid tree with strong support (*Minaya et al., 2015*), and in a BI tree based on 122 nuclear loci, but not in MP-EST (maximum pseudo-likelihood for estimating species analyses) analyses of the same nuclear data (*Liu, Yu & Edwards, 2010*). By contrast, a strongly supported conflicting topology, in which Danthonioideae is sister to Arundinoideae + Chloridoideae + Panicoideae (Micrairoideae not sampled), was identified in a Bayesian analysis of combined *ndhF* and *phyB* data (*Vicentini et al., 2008*), a topology likely influenced by the nuclear gene included there. Another strongly conflicting topology, in which the remainder of the PACMAD subfamilies did not form a clade, was found in a nuclear *β*-amylase phylogeny: Chloridoideae were placed sister to the BOP clade, Danthonioideae were sister to Chloridoideae + the BOP clade, and Panicoideae was not resolved as monophyletic (*Minaya et al., 2015*). The maximally supported clade comprising Arundinoideae and Micrairoideae in the plastome trees has been recovered in other plastome studies (*Duvall et al., 2010*; *Cotton et al., 2015*; *Teisher et al., 2017*), whereas in few-gene studies the clade has been recovered with poor (*Duvall et al., 2007*; *Sánchez-Ken et al., 2007*; *Christin et al., 2008a*) or strong support (*Grass Phylogeny Working Group II, 2012*). Given the consistent, non-conflicting support in the plastome trees we are confident in the accuracy of the relationships among these four subfamilies inferred from plastome data.

### *Danthonioideae*

Danthonioideae includes a single tribe comprising 18 genera, and one genus is *incertae sedis* in the subfamily (*Linder et al., 2010*; *Soreng et al., 2017*). The current analyses include plastomes from seven species and six genera. We find strong support from all plastome partitions for successive divergences of *Chionochloa macra* Zotov, *Chaetobromus involucratus* subsp. *dregeanus* (Nees) Verboom, *Danthonia californica* Bol., *Tribolium hispidum* (Thunb.) Desv., *Tenaxia guillarmodiae* (Conert) N.P. Barker & H.P. Linder and *Rytidosperma* Steud. (two species). This topology is congruent with and better supported than the few-gene phylogenetic tree on which the current classification of the subfamily is based (*Linder et al., 2010*).

### Arundinoideae and Micrairoideae

Arundinoideae includes Arundineae, with four genera and represented here by *Monachather paradoxa* Steud., and Molinieae, with two subtribes (Crinipinae and Molininae) and three genera *incertae sedis* (*Soreng et al., 2017*). Crinipinae includes four genera and is represented here by *Elytrophorus spicatus* (Willd.) A. Camus, and subtribe Molininae includes four genera and is represented here by *Hakonechloa macra* (Munro) Honda and *Phragmites australis* (Cav.) Trin. ex Steud., as in *Cotton et al. (2015)*. Relationships among these taxa are congruent with the plastome trees in *Piot et al. (2018)* and *Teisher et al. (2017)*, the latter one better sampled.

Micrairoideae includes three tribes, each represented here: Micraireae (monogeneric), represented here by *Micraira* F. Muell., Eriachneae (monogeneric) by *Eriachne* Eck- Boorsb. and Isachneae (six genera) by *Isachne* R. Br. As in other studies, Micraireae is sister to Eriachneae + Isachneae (*Sánchez-Ken et al., 2007*; *Cotton et al., 2015*; *Piot et al., 2018*; *Teisher et al., 2017*). Of the eight genera of Micrairoideae (*Soreng et al., 2017*), plastomes have not yet been published from species of *Coelachne* R. Br., *Heteranthoecia* Stapf. and *Sphaerocaryum* Nees ex Hook. f.

### Panicoideae

Panicoideae includes 13 tribes and three genera *incertae sedis* (*Soreng et al., 2017*). Overall support and topology among panicoid lineages here is nearly identical to the earlier plastome study of *Burke et al. (2016b)*, and the plastome trees are generally better resolved and supported than in earlier few-gene plastid trees (*Sánchez-Ken & Clark, 2010*; *Morrone et al., 2012*). However, deep relationships among Zeugiteae, Chasmanthieae, Tristachyideae, Centotheceae and Thysanolaeneae conflict among plastome partitions: the five taxa form a weakly to moderately supported clade in some trees (E–H, X, Z, BP = 60– 87%), as in *Burke et al. (2016b)*. In three trees, however, Tristachyideae + Centotheceae + Thysanolaeneae and Zeugiteae + Chasmanthieae are successively diverging sisters to the rest of the subfamily, with the large clade including Zeugiteae + Chasmanthieae strongly supported as the sister group of the rest of Panicoideae excluding Tristachyideae + Centotheceae (Q, W, Y, BP = 90–100%). These differing topologies indicate some strong discordance in the plastome datasets. Plastomes representing Cyperochloeae and Steyermarkochloeae have not yet been published. *Lecomtella madagascariensis* A. Camus is the next to diverge, consistent with recognition of this taxon in its own tribe, Lecomtelleae (*Besnard et al., 2013*; *Soreng et al., 2017*).

The strongly supported relationships among Paniceae, Paspaleae, Andropogoneae and Arundinelleae in the plastome trees are congruent with other studies: Paniceae is sister to a clade comprising Paspaleae + (Andropogoneae + Arundinelleae) (*Grass Phylogeny Working Group II, 2012*; reviewed in *Kellogg (2012)*). This latter clade was recently recognized as supertribe Andropogonodae (*Soreng et al., 2017*), a clade mainly of species with $x = 10$ that is robustly supported in the plastome trees. Paspaleae genera, recently reconstituted a tribe (*Morrone et al., 2012*), were historically included in supertribe Panicodae (*Soreng et al., 2015b*).

*Andropogoneae*

Nine subtribes and six genera *incertae sedis* are recognized in tribe Andropogoneae (*Soreng et al., 2017*), a large lineage in which multiple allopolyploidization events have been documented (*Estep et al., 2014*). The two subtribes of Andropogoneae for which plastomes are not sampled here are Chionachninae (five genera) and Germainiinae (four genera) (*Soreng et al., 2017*). Relationships among the seven subtribes sampled in the plastome trees here and in *Burke et al. (2016b)* are much better resolved and supported than in studies based on a few plastid genes and ITS (*Mathews et al., 2002*; *Skendzic, Columbus & Cerros-Tlatilpa, 2007*; *Teerawatananon, Jacobs & Hodkinson, 2011*), and are mostly congruent with recent studies of low-copy nuclear loci in the tribe (*Estep et al., 2014*; *Hawkins et al., 2015*) and with another plastome tree with somewhat different sampling (*Piot et al., 2018*). Andropogoninae and Saccharinae are sister taxa, and within Andropogoninae, the sister group relationship between *Diheteropogon amplectens* var. *catangensis* and *Hyparrhenia subplumosa*, and the successive branching of *Themeda* sp., *Iseleima macratherum* and *Bothriochloa alta* + *Capillipedium venustum*, are congruent with other plastid and nuclear trees (*Kellogg, 2012*; *Estep et al., 2014*; *Hawkins et al., 2015*).

Saccharinae includes 26 genera, of which we sampled *Eulalia* Kunth, *Saccharum* L., *Sorghum*, *Sorghastrum* Nash and *Imperata* Cirillo. In all but one plastome tree, *Eulalia* and *Sorghastrum* are sister taxa, with varying levels of support; this topology is congruent with the plastome tree in *Burke et al. (2016b)*. Most earlier studies did not identify a close relationship between these genera (*Skendzic, Columbus & Cerros-Tlatilpa, 2007*; *Grass Phylogeny Working Group II, 2012*), although they were included in the same clade (with other genera) in trees based on low-copy nuclear loci (*Estep et al., 2014*; *Hawkins et al., 2015*). In all but one plastome tree, *Saccharum* and *Sorghum* form a strongly supported clade, and *Imperata* is sister to this clade. The affinities of the *Eulalia* + *Sorghastrum* and *Imperata* + *Saccharum* + *Sorghum* clades, however, vary among trees; in other words, Saccharinae is not consistently resolved as monophyletic. *Burke et al. (2016b)* found *Eulalia* + *Sorghastrum* to be sister to Andropogoninae, with weak to strong support, and we find this same topology with weak to moderate support in five plastome trees, of which all but one are based on partitions excluding gapped sites. In four analyses including gapped sites, however, Saccharinae is monophyletic, and *Eulalia* + *Sorghastrum* and *Imperata* + *Saccharum* + *Sorghum* are sister clades.

The strongly supported clade comprising *I. afrum* (Ischaeminae; six genera) and *C. lacryma-jobi* (Coicinae; one genus) + *R. cochinchinensis* (Rottboelliinae; 16 genera) that is sister to Andropogoninae + Saccharinae in the plastome trees has also been found in other plastid studies (*Grass Phylogeny Working Group II, 2012*), but was not recovered in nuclear analyses, in which most deep branches within the tribe were poorly supported (*Estep et al., 2014*). At the base of the Andropogoneae clade in the plastome trees, subtribes Arthraxoninae (one genus) and Tripsacinae (seven genera, represented by *Zea*) diverge successively as sisters to the rest of the tribe, consistent with the better-sampled nuclear tree in *Estep et al. (2014)*.

*Paspaleae*

Paspaleae includes subtribes Paspalinae, Otachyriinae and Arthropogoninae and the *incertae sedis* genus *Reynaudia* Kunth. Our sampling includes two genera of Paspalinae (*Axonopus* P. Beauv. and *Paspalum*), three of Otachyriinae (*Otachyrium* Nees, *Plagiantha* Renvoize and *Steinchisma* Raf.) and two of Arthropogoninae (*Coleataenia* Griseb. and *Oncorachis* Morrone & Zuloaga). Monophyly of each subtribe is robustly supported in the plastome trees, and despite our limited taxon sampling, the strong support for a sister group relationship between Arthropogoninae and Otachyriinae is an improvement on earlier studies with greater taxon sampling but less sequence data per taxon, in which relationships among the three subtribes were unresolved (*Acosta et al., 2014*) or only weakly supported (*Grass Phylogeny Working Group II, 2012*). Relationships among the subtribes are similarly resolved in the plastome tree of *Piot et al. (2018)*. Relationships among the three sampled genera of Otachyriinae in our plastome trees are congruent with earlier plastid trees (*Grass Phylogeny Working Group II, 2012*; *Acosta et al., 2014*).

*Paniceae*

Paniceae includes seven subtribes and eight genera *incertae sedis* (*Soreng et al., 2017*). There has been considerable phylogenetic investigation of the tribe, mostly based on one or a few gene regions (reviewed in *Washburn et al. (2015)*). Numerous clades now recognized as tribes and subtribes were identified in a single-plastid-gene study with dense taxon sampling, in which most aspects of backbone relationships were unresolved (*Morrone et al., 2012*). Phylogenomic studies are providing new insights into relationships in this tribe. *Washburn et al. (2015)* analyzed 78 chloroplast, 22 mitochondrial and 2 nrDNA loci from 45 taxa of Paniceae, *Burke et al. (2016b)* analyzed complete plastomes from 16 taxa of Paniceae, and our sampling of the tribe builds slightly on the latter study by adding three additional plastomes. Plastid-based topologies in the two earlier studies and the current one are similar, with the following lineages diverging successively: Anthephorinae, Boivinellinae, Dichantheliinae + Neurachninae, Panicinae, Melinidinae and Cenchrinae. The strong support along the backbone of the Paniceae tree in the current and earlier plastome studies (*Washburn et al., 2015*; *Burke et al., 2016b*) is a substantial improvement on studies with considerably less genomic sampling (*Grass Phylogeny Working Group II, 2012*; *Morrone et al., 2012*; *Zuloaga, Salomón & Scataglini, 2014*). The moderately to maximally supported clade comprising Dichantheliinae + Neurachninae and Cenchrinae + Melinidinae + Panicinae is congruent with the plastome and combined plastome, mitochondrial and nuclear trees in *Washburn et al. (2015)*. They also found *Sacciolepis* Nash (*incertae sedis* within Paniceae and not sampled here) to be part of this clade and moderately to strongly supported as sister to Dichantheliinae + Neurachninae. In a combined nuclear (*phyB*) and plastid (*ndhF*) tree, however, *Sacciolepis* is embedded in Panicinae (*Vicentini et al., 2008*), possibly reflecting discordance between plastid and nuclear data due to hybridization. Relationships inferred among taxa of Melinidinae in *Washburn et al. (2015)*, *Burke et al. (2016b)* and the current trees are also congruent. In this study, *W. capillipes* is united with species of Panicinae in all trees from the 14 partitions (mean BP = 92%) as was inferred earlier (*Burke et al., 2016a*). However,

this result conflicts with those of two previous studies using incompletely sequenced plastid loci. The previous studies placed *W. capillipes* among Cenchrinae, but with marginal jackknife or BP (both <50%; *Grass Phylogeny Working Group II, 2012*; *Morrone et al., 2012*). The two earlier studies obtained DNA from the same plant (voucher: *J. Risler 1804*, MO) and the loci sequenced are among those with the greatest number of positively selected sites (*ndhF*, *matK* and *rbcL*); the plastome data were obtained from a different accession. The inflorescence and spikelet morphologies of *Whiteochloa* C.E. Hubb. are more similar to those of Panicinae than Cenchrinae. Setae, which are synapomorphic for Cenchrinae (hence the common name "Bristle clade"), are notably absent from *Whiteochloa* (*Morrone et al., 2012*).

Four possible explanations for this discrepancy are: (1) The use of plastid loci, which have high numbers of positively selected sites, skewed previous phylogenetic analyses. Such phylogenies, especially when based on a single gene, are susceptible to selection artifacts (see above). (2) One of the two plants was misidentified, although it is not clear which other Australian grasses might be mistaken for *W. capillipes*. When homologous regions from our complete plastome are aligned with the previously sequenced markers, nucleotide identities range only from 95% to 97%, which suggests that the two sources of DNA are not conspecific. The plant used to produce the complete plastome (*Duvall s.n.*, DEK) shows characters that are diagnostic for *Whiteochloa*. However, two duplicates of the voucher *J. Risler 1804* (DNA, MO) are also consistent with the current concept of *W. capillipes*. (3) *W. capillipes* is actually a complex of hybrids between species of Panicinae and Cenchrinae. The direction of the cross would determine which of two possible plastome haplotypes (Panicinae or Cenchrinae) was captured, depending on the female parent, whereas the morphological phenotypes of the reciprocal hybrids might be similar. Nuclear sequences of Panicinae and Cenchrinae, which have not been obtained to date, would be needed to test this third point and determine the identities of parent species. (4) There was a labeling mix-up or contamination somewhere in the extraction or sequencing process. This could be clarified by re-extracting and sequencing both specimens.

Subtribe Cenchrinae includes some 24 genera (*Soreng et al., 2017*) and most aspects of its phylogeny are poorly resolved (*Kellogg et al., 2009*; *Chemisquy et al., 2010*; *Morrone et al., 2012*). The two genera included here, *Cenchrus* L. and *Setaria*, of which we sampled three species, form a clade, as in *Burke et al. (2016b)*, but relationships among them conflict strongly. In eight trees, *Cenchrus* and *Setaria italica* + *Setaria viridis* form a maximally supported clade, whereas in two trees *Setaria* is strongly supported as monophyletic and sister to *Cenchrus*. Although the species treated here as *Setaria geminata* has been recognized in the genus *Paspalidium* Stapf, many authors have included it in *Setaria* (*Webster, 1993*, *1995*; *Veldkamp, 1994*; *Morrone et al., 2014*; *Soreng et al., 2017*). In our trees, inclusion of *Paspalidium* in *Setaria* is supported only by plastome coding data. In other studies, relationships among *Setaria* and related genera are unclear. For example, in a broadly sampled *ndhF* tree of *Setaria* and related genera, species of *Paspalidium* formed a clade that was part of a broader clade including a subset of *Setaria* species (including the type species) from China and South America, *Ixophorus unisetus* (J. Presl) Schltdl., *Zuloagaea bulbosa* (Kunth) E. Bess, *Stenotaphrum secundatum* (Walter)

Kuntze and *Uranthoecium truncatum* (Maiden & Betche) Stapf (*Kellogg et al., 2009*). Sampling of a much broader selection of genera and species in Cenchrinae will be needed to clarify the plastome phylogeny in this group.

Subtribe Boivinellinae includes 18–19 genera (*Silva et al., 2017*; *Soreng et al., 2017*). *Burke et al. (2016b)* sampled three of these and found them to form a maximally supported clade, but relationships among the genera differed among their MP (*Amphicarpum* sister to *Oplismenus* + *Echinochloa*), and ML and BI analyses (*Oplismenus* sister to *Amphicarpum* + *Echinochloa*). Relationships among these genera similarly differ among our plastome trees. In nuclear phylogenies, *Echinochloa* is not part of the Boivinellinae clade (*Christin et al., 2007*; *Vicentini et al., 2008*). In a better-sampled plastome tree, a maximally supported clade of six species of *Echinochloa* is weakly supported as sister to a maximally supported clade comprising *Brachiaria fragrans* A. Camus, *Panicum locopodioides* Bory ex Nees, *Chasechloa* A. Camus, *Lasiacis nigra* Davidse, *Oplismenus burmannii* (Retz.) P. Beauv. and *Pseudolasiacis leptolomoides* (A. Camus) A. Camus (*Piot et al., 2018*). *Washburn et al. (2015)* sampled *Alloteropsis*, *Echinochloa* (two species) and *Oplismenus*, and relationships among these taxa were weakly supported in their plastid tree. Our trees include *Amphicarpum*, *Oplismenus*, two species of *Echinochloa* and three species of *Alloteropsis*. The latter genus is of particular interest to evolutionary biologists (*Lundgren et al., 2015*) because one species, *A. semialata*, has two subspecies differing in photosynthetic pathway: one is $C_3$ and one is $C_4$ (*Gibbs Russell, 1983*; *Lundgren et al., 2016*). Monophyly of Boivinellinae is strongly supported in all but tree Y, in which *Amphicarpum* is strongly supported as sister to *T. xerophila* and the lineage is placed outside the main Boivinellinae clade; reasons for this alternative topology are unclear. Relationships among the three species of *Alloteropsis* are congruent with the tree in *Ibrahim et al. (2009)*, which identifies two major lineages in the genus: one comprising *Alloteropsis cimicina* (L.) Stapf sister to *Alloteropsis paniculata* (Benth.) Stapf + *Alloteropsis papillosa* Clayton; the other comprising *Alloteropsis angusta* and *Alloteropsis semialata*. Relationships among the four genera, however, vary among the plastome trees. Similar uncertainty at the base of the Boivinellinae tree is present in other studies based on fewer gene regions but greater taxon sampling (*Grass Phylogeny Working Group II, 2012*; *Morrone et al., 2012*; *Silva et al., 2017*).

### Chloridoideae

Chloridoideae includes five tribes (Centropodieae, Triraphideae, Eragrostideae, Cynodonteae and Zoysieae) and seven genera *incertae sedis* at tribal rank (*Soreng et al., 2017*). The successive branching order of Centropodieae, Triraphideae, Eragrostideae, Zoysieae and Cynodonteae is congruent with other studies of plastid data (*Columbus et al., 2007*; *Peterson, Romaschenko & Johnson, 2010*; *Peterson et al., 2011*, *Peterson, Romaschenko & Herrera Arrieta, 2016*; *Peterson, Romaschenko & Arrieta, 2014a*; *Duvall et al., 2016*; *Piot et al., 2018*). However, the relationship of Centropodieae to the rest of Chloridoideae is unclear in nuclear trees, which did not resolve relationships among Chloridoideae, Arundinoideae, Danthonioideae and Centropodieae (including *Centropodia* Rchb. and *Ellisochloa* P.M. Peterson & N.P. Barker) (*Fisher et al., 2016*).

Relationships among the other four tribes of Chloridoideae in nuclear trees, however, are congruent with the plastome trees (*Fisher et al., 2016*). Our sampling of Chloridoideae includes one of the three genera of Triraphideae (*Triraphis* R. Br.), two of the three subtribes of Eragrostideae (Eragrostidinae, Unioliinae) and both subtribes of Zoysieae (Sporobolinae, Zoysiinae). The relationships we find among the three sampled species of *Sporobolus* (Sporobolinae), of which two were previously recognized in the genus *Spartina* Schreb., are congruent with previous studies (*Peterson et al., 2014b*, *2014c*).

Cynodonteae is a large and variable tribe consisting of 94 genera (*Peterson, Romaschenko & Herrera Arrieta, 2016*; *Soreng et al., 2017*) for which there are no known morphological synapomorphies. Twenty-five subtribes of Cynodonteae are currently recognized (*Peterson, Romaschenko & Arrieta, 2014a*; *Peterson, Romaschenko & Herrera Arrieta, 2016*; *Peterson, Romaschenko & Herrera Arrieta, 2017a*; *Soreng et al., 2017*), some of which are combined in *Kellogg (2015)*, and eight of these are represented here. Sampling in Boutelouinae, Monanthochloinae and Hilariinae is identical to that in *Duvall et al. (2016)*, and relationships among these subtribes (Hilariinae sister to Boutelouinae + Monanthochloinae) are congruent with other plastid analyses (*Peterson, Romaschenko & Johnson, 2010*; *Peterson, Romaschenko & Arrieta, 2014a*, *2015*). They are also congruent with the 56 nuclear gene MP-EST tree in *Fisher et al. (2016)*, but not their 122 locus BI tree, in which *Distichlis* and *Hilaria* are sister taxa. Based on a phylogeny using seven plastid and ITS regions, these three subtribes are part of a larger clade recognized as supersubtribe Boutelouodinae, also including subtribes Allolepiinae, Jouveinae, Kaliniinae, Muhlenbergiinae, Scleropogoninae, Sohnsiinae and Traginae (*Peterson, Romaschenko & Herrera Arrieta, 2017a*). In *Peterson, Romaschenko & Herrera Arrieta (2017a)*, Boutelouinae + Monanthochloinae are moderately supported as sister to Kaliniinae and Hilariinae are unsupported as sister to Allolepiinae.

Compared to a previous plastome study of Chloridoideae (*Duvall et al., 2016*), we include new plastomes from Trichoneurinae (*Trichoneura grandiglumis*), Tripogoninae (*Halopyrum mucronatum* and *Melanocenchris abyssinica*), Triodiinae (*Triodia stipoides* (S.W.L. Jacobs) Crisp & Mant and *Triodia wiseana* C.A. Gardner), Eleusininae (*Astrebla pectinata, Chloris barbata, Eustachys glauca, Leptochloa pluriflora, Oxychloris scariosa*) and Dactylocteniinae (*Dactyloctenium aegyptium*). The sister group relationship between Trichoneurinae and Dactylocteniinae is congruent with and better supported than recent plastid trees in which *Neobouteloua* Gould + *Dactyloctenium* (L.) Willd. (Dactylocteniinae) are sister to a large but weakly supported clade including Orcuttiinae and a clade recognized as supersubtribe Gouiniodinae comprising subtribes Cteniinae, Farragininae, Gouiniinae, Hubbardochloinae, Perotidinae, Trichoneurinae and Zaqiqahinae, of which only Trichoneurinae is sampled here (*Peterson, Romaschenko & Arrieta, 2015*; *Peterson, Romaschenko & Herrera Arrieta, 2016*; *Soreng et al., 2017*). In a combined plastid and ITS tree, however, Dactylocteniinae and Eleusininae are sister taxa, albeit with weak support (*Peterson, Romaschenko & Herrera Arrieta, 2016*). The close relationship between the two species of *Triodia* (Triodiinae), one of which (*T. stipoides*) was previously recognized in *Monodia* S.W.L. Jacobs, is congruent with other studies (*Hilu & Alice, 2001*; *Grass Phylogeny Working Group II, 2012*). *Monodia* is nested within

*Triodia* (*Toon et al., 2015*), and this and the related genus *Symplectrodia* Lazarides have been synonymized under *Triodia* (*Crisp et al., 2015*; *Soreng et al., 2017*).

The branching order of the five sampled taxa of Eleusininae is congruent with the plastid tree that included 28 genera of the subtribe in *Peterson, Romaschenko & Arrieta (2015)*, but clear resolution of relationships awaits the inclusion of the remaining 143 species in 23 genera of Eleusininae not sampled in our study. Strong support in six trees for *A. pectinata* and *L. pluriflora* being successive sisters to the *O. scariosa* + *E. glauca* + *C. barbata* clade is congruent with other plastid trees (*Peterson, Romaschenko & Herrera Arrieta, 2016*). However, the strongly supported but discordant (reversed) branching order of *A. pectinata* and *L. pluriflora* in one plastome tree (noncoding regions including gapped sites) is congruent with the combined plastid + ITS tree in *Peterson, Romaschenko & Herrera Arrieta (2016)*, although the relevant deep branches in their tree only moderately supported. Relationships among Eleusininae, Triodiinae and Dactylocteniinae are poorly supported in the plastid trees in *Peterson, Romaschenko & Herrera Arrieta (2016)*, whereas we find these three lineages plus Trichoneurinae to be a clade with moderate to strong support in analyses of noncoding regions and complete plastomes. *Peterson, Romaschenko & Herrera Arrieta (2016)* found this same unsupported clade to include nine additional subtribes (Aeluropodinae, Cteniinae, Farragininae, Gouiniinae, Hubbardochloinae, Orcuttiinae, Orininae, Perotidinae, Zaqiqahinae). Nevertheless, interrelationships among these subtribes vary. The topology in two trees, in which Dactylocteniinae + Trichoneurinae and Eleusininae are sister groups, is congruent with the 56 nuclear gene MP-EST tree in *Fisher et al. (2016)*, who did not sample Trichoneurinae. However, topology of the nuclear tree in *Fisher et al. (2016)* was mostly incongruent with better sampled plastid trees (*Peterson, Romaschenko & Johnson, 2010*; *Peterson, Romaschenko & Arrieta, 2014a*), and they attributed these nuclear and plastid incongruences to incomplete lineage sorting and gene flow. We also find one moderately supported alternative topology in one plastome tree, in which Triodiinae and Dactylocteniinae + Trichoneurinae are sister groups. The deep branches in this part of the Cynodonteae clade are consistently short in all trees (Fig. S1), which is likely contributing to the topological uncertainty. Lack of representative sampling of related subtribes in the plastome tree is also likely contributing to the problematic reconstruction of deep relationships among these lineages.

*Peterson, Romaschenko & Herrera Arrieta (2016)* were the first to sample the monotypic *Halopyrum* Stapf in a molecular study and found it to be nested within Tripogoninae, where it is now classified. Our results are consistent with this, as we find *H. mucronatum* and *M. abyssinica* to be sister taxa, although we lack samples of five additional genera that have been placed in the subtribe (*Peterson et al., 2017b*; *Soreng et al., 2017*). The differing deep placements for Tripogoninae in the plastid trees, either in a robust clade with Eleusininae + Triodiinae + Dactylocteniinae + Trichoneurinae (the dominant topology) or sister to the remainder of the Cynodonteae (only in tree R), are likely the result of either conflicting signal or lack of signal in the noncoding partition excluding gapped sites. Moreover, in the single best ML tree from this analysis (Fig. S1), relationships among these lineages are unresolved and it is therefore surprising there is such high BP for this alternative topology. Overall, the strong support from plastome data for the mostly

non-conflicting relationships inferred among subtribal taxa of Cynodonteae is a considerable improvement on earlier plastid trees, in which relationships among these subtribal lineages were mostly poorly supported (*Grass Phylogeny Working Group II, 2012*; *Peterson, Romaschenko & Arrieta, 2014a*, *2015*), and denser plastome sampling across the tribe may lead to improved understanding of relationships, particularly of deep branches. However, studies of nuclear loci that have identified topologies that conflict with those from plastid data indicate that plastome analyses likely do not accurately reflect evolutionary history in this large clade (*Fisher et al., 2016*).

## FUTURE DIRECTIONS

Taxon density is key for resolving contradictory branching patterns. A robust plastome phylogeny of Poaceae with complete genus-level sampling is likely attainable within five years, dependent on the availability of rare species or those restricted to remote areas. Such a phylogenetic tree would contribute to further improvement, refinement and confidence in classification of grasses, and would facilitate broad characterization of the molecular evolutionary histories of plastomes in grasses, allow more precise divergence estimations, and produce complete descriptions of microstructural events and rare genomic changes in the plastome. We suggest that future work on plastome phylogeny of Poaceae should aim to (1) sample representatives from the few tribes (Guaduelleae, Duthieae, Brylkinieae, Cyperochloeae, Steyermarkochloeae) and many subtribes for which plastomes have not yet been published, as well as the few genera that remain unplaced to a tribe or a subtribe, most of which have no sequence data available; (2) complete genus-level sampling of plastomes in all tribes and subtribes, focusing first on genera whose affinities within tribes, subtribes and subfamilies are unclear in existing phylogenies (e.g., *Milium*, *Beckmannia*, *Cinna*) in Poeae group 2; *Avenella* in Airinae) and genera that have not been included in any phylogenetic study, such as *Phyllorachis* (Phyllorachideae); (3) sample plastomes representing all lineages of genera known to be para- or polyphyletic and which have not yet undergone taxonomic revision; (4) sample taxa of particular evolutionary interest, such as *Aristida longifolia*, the $C_3$ sister to all other species of *Aristida*, which are $C_4$. Attention should also be paid to stems and loop micro inversions and data bias from complementary base pair mutations in stems, in mechanical alignments. Attention to nuclear genomes is needed beyond all this, to clarify reticulation events, as those are not evident in plastid trees.

## CONCLUSION

High throughput sequencing, high speed computation, and big data science software tools together facilitate genome-scale systematic studies of plants. Here, we present the most comprehensive plastome phylogenomic study of Poaceae to date with specific emphasis on the effects of data partitioning. The plastome phylogeny is highly congruent with the latest classifications of Poaceae, with most branches that define tribes and subtribes strongly, and usually maximally supported, even though we have not sampled plastomes from all of the tribes. One interesting exception is the subtribal placement of *Whiteochloa* in Panicoideae, a result that should be confirmed through further study.

We demonstrate strong improvements in resolution and support in our plastome phylogenomic analyses of Poaceae, particularly when compared to single and few-gene plastid phylogenies. We recommend analyses of both coding and noncoding plastome regions while excluding regions that may be aligned ambiguously by removing all sites with gaps introduced by the alignment. Such ambiguous regions sometimes showed spurious "signal."

Although the plastid coding loci uniformly show a dominant signal of purifying selection, positively selected codons were also identified in most loci. We show that widely used loci in grass systematics, such as *ndhF*, *matK*, *rpoC2*, and *rbcL*, are particularly subject to selective effects and have the highest numbers of positively selected codons among plastid loci. Use of noncoding intergenic spacers, introns, and protein coding loci such as certain photosystem genes e.g., *psaA* and *psaB*, which have few positively selected codons, can reduce phylogenomic artifacts due to selection.

Relationships among PACMAD subfamilies were previously reported and widely cited in a benchmark paper of grass systematics (*Grass Phylogeny Working Group II, 2012*). That study was taxonomically denser than this study, but relied on considerably less molecular data, coincidentally analyzed the loci with the most positively selected codons, and, in the interest of retaining the most sequence data, included alignment gaps. Here we show that the positions of Panicoideae and Aristidoideae relative to the remaining PACMAD subfamilies is dependent on the data partition that is analyzed, and that earlier results should be viewed in the context of that information. This is a pivotal node in grass systematics that has broad historical significance for the adaptive changes that occurred during transitions from forests/forest margin habitats to open grasslands. Additional taxonomic sampling of plastomes and parallel nuclear studies obtained from transcriptomes or libraries enriched with specific targets will be needed to fully address these issues.

## ACKNOWLEDGEMENTS

We thank Dr. George Rogers, West Palm Beach State University, for help obtaining materials, Olivia East, Northern Illinois University, for technical assistance, and Marcial Escudero, Elizabeth Kellogg and an anonymous reviewer for constructive feedback on earlier versions of the manuscript. Any opinions, findings, and conclusions or recommendations expressed in this material are those of the authors and do not necessarily reflect the views of the National Science Foundation.

### Funding

This work was supported by the Plant Molecular and Bioinformatics Center and the Department of Biological Sciences at Northern Illinois University and grants from the National Science Foundation to Lynn G. Clark (DEB-1120750) and Melvin R. Duvall (DEB-1120761 and DEB-1342782). There was no additional external funding received for this study. The funders had no role in study design, data collection and analysis, decision to publish, or preparation of the manuscript.

## Grant Disclosures

The following grant information was disclosed by the authors:
Plant Molecular and Bioinformatics Center and the Department of Biological Sciences at Northern Illinois University.
National Science Foundation: DEB-1120761, DEB-1342782 and DEB-1120750.

## Competing Interests

Joseph M. Craine is an employee of Jonah Ventures and an Academic Editor for PeerJ.

## Author Contributions

- Jeffery M. Saarela conceived and designed the experiments, analyzed the data, wrote the paper, prepared figures and/or tables, reviewed drafts of the paper.
- Sean V. Burke conceived and designed the experiments, performed the experiments, analyzed the data, contributed reagents/materials/analysis tools, wrote the paper, prepared figures and/or tables, reviewed drafts of the paper.
- William P. Wysocki conceived and designed the experiments, performed the experiments, analyzed the data, contributed reagents/materials/analysis tools, reviewed drafts of the paper.
- Matthew D. Barrett contributed reagents/materials/analysis tools, reviewed drafts of the paper.
- Lynn G. Clark contributed reagents/materials/analysis tools, reviewed drafts of the paper.
- Joseph M. Craine contributed reagents/materials/analysis tools, reviewed drafts of the paper.
- Paul M. Peterson contributed reagents/materials/analysis tools, reviewed drafts of the paper.
- Robert J. Soreng reviewed drafts of the paper.
- Maria S. Vorontsova reviewed drafts of the paper.
- Melvin R. Duvall conceived and designed the experiments, performed the experiments, analyzed the data, contributed reagents/materials/analysis tools, wrote the paper, prepared figures and/or tables, reviewed drafts of the paper.

## DNA Deposition

The following information was supplied regarding the deposition of DNA sequences:
The new plastome sequences are accessible via GenBank accession numbers MF460970 to MF460984.

## Data Availability

The raw data is provided in the Supplemental Datasets Files.

## Supplemental Information

Supplemental information for this article can be found online at http://dx.doi.org/10.7717/peerj.4299#supplemental-information.

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
