# Peer review of "A 250 plastome phylogeny of the grass family (Poaceae): topological support under different data partitions"

_PeerJ, doi:10.7717/peerj.4299_

## Round 0.1 · original submission · Minor Revisions

Dear Jeff and Melvin,

The three reviewers and I agree that this paper is an important contribution to the systematics of grasses. There are many suggestions from the three reviewers. For example, the reviewers agree that the paper is too long. I hope you are able to include reviewers' suggestions and resubmit the paper for a second round of revisions.
Sincerely,

Marcial.

Ps. The full review from reviewer #3 is attached as a formatted PDF.

Reviewer 1 ·

Basic reporting

The paper is clear and well written, but too long, especially the Discussion section. The aims of the paper are clear but maybe too general. Perhaps the authors could indicate what specific topological problems in the grasses they hope to resolve with this new analysis. This I think could help better focus the discussion, especially considering that most topological relationships recovered are totally or partially consistent with previous results from other publications (very often authored by researchers also included in this study).

Sufficient field context is provided, and the literature review is very complete, but perhaps too exhaustive (please see general comments). The article is well structured and figures and tables are very good. Supplemental files are informative and in line with PeerJ shared information policy. Results are relevant and in line with the established aims.

There are no clear hypotheses in the paper. This is understandable given the nature and the aims of the study, but perhaps (as said above) the authors could indicate what major topological problems they want to solve. They could also indicate what differential behavior they expect from databases including or excluding gaps, sites under positive selection, etc (please see general comments)

Experimental design

The paper is clearly within the aims and scope of the journal. The methods are rigorous and they are described clearly and with sufficient detail. Research questions could perhaps be better defined (please see general comments and the basic reporting section). Results are relevant and mostly consistent with previous research.

Validity of the findings

Nothing to say here. Suggestions on original research questions are included in general comments

Additional comments

Comments on general research questions and hypotheses.
The phylogenetic relationships obtained by the authors are relevant and interesting, and their results show increased support for a lot of relationships that heretofore were poorly resolved. In phylogenetic analyses such as this it is often difficult to establish clear research questions (or starting hypotheses). However, perhaps the authors could consider indicating what poorly or ambiguously supported relationships in the grass phylogeny they aim at resolving. Once they have established that, perhaps it would be convenient to focus the discussion on those nodes in particular.
The sections of the article that deal with the effect of positively selected or gapped sites in the topology and resolution of the trees are also interesting. However, again the research question on this topic is not clearly stated. What did the authors expect to obtain? They made a big effort analysing different partitions, so surely they had some expectations about the results. At least as regards gapped sites they certainly had prior information on the effect of these sites (treated as missing data) on Maximum Likelihood (ML) analyses (they add several very relevant references on the issue). Considering this latter aspect, it is not clear to me why they included gapped sites (as missing data) in their analyses with ML if there is such strong evidence against it. Were they trying to confirm it? If so, this should be included as an aim of the paper. In any case, in my view more information should be added on why they included gapped sites.

Materials and methods.
The methodology of the paper is clear and sound. Some things however could perhaps be further clarified. The authors only used ML analyses, and do not perform Maximum Parsimony (MP) or Bayesian Inference (BI). The authors indicate that “We conducted only maximum likelihood (ML) analyses, given the large size of our datasets, in terms of both numbers of taxa and nucleotides” (lines 215-216). It is not clear to me why the other methods were excluded by the size of the matrices. Actually, several of the articles they cite that deal with plastome phylogenies use BI or MP methods (e.g. Burke et al., 2014. American Journal of Botany 101: 886-891). It could even be interesting to know if some of the nodes that receive differential support in the different matrices used by the authors are also ambiguous using other phylogenetic inference methods.
The authors include a lot of partitions in the analyses, which are very complete. Did they consider using best-fit partitioning schemes and models such as the ones implemented in partitionfinder v.1.1.1 (Lanfear et al. 2012. Molecular Biology and Evolution 29: 1695-1701)? (it is just a suggestion, the number of partitions tried is quite high).
As regards the methods used to build the libraries for NGS, both Nextera and Nextera XT are used. According to the methods section (the authors refer the reader to Burke et al., 2016b), plastids were not extracted prior to the building of the libraries in any of the samples, so why using Nextera in some samples and Nextera XT in others? Did the authors consider the genome size of the different grasses? Perhaps the authors could give and explanation for this in the paper (although both methods are valid).
Finally regarding methodology, a reference should be added to acknowledge the authorship of the MEME model beyond the HyPhy software (Murrell et al., 2012. PLOS Genetics 8: e1002764).

Results and discussion.
The results of the paper are clearly described. The discussion is also relevant and interesting, but I think some changes should be made to make it easier to follow. Some of the changes have to do with the inclusion of clearer research questions (see above). Perhaps the number of references that the authors include to support some of the phylogenetic relationships that are more clearly supported in the literature is a bit overwhelming. In my view, the authors should select the most relevant ones.
The discussion has two very clearly separated parts, one analyses the support of the different clades in the different partitions in a general fashion and the other focuses on the phylogeny of the different groups. However, the discussion becomes a bit repetitive in some parts (e.g. lines 1052-1129) where the authors re-discuss the effect of gapped and positively selected sites on topologies (this time focused on one particular node or group of nodes). Perhaps the discussion could be re-organised to avoid this (and to reduce its extension). Also in the discussion there is a part (“Rooting the grass phylogenetic tree”) that could, in my view, be eliminated. It is interesting, but it does not actually discuss the results of the authors, they use this section to justify a decision they made in the materials and methods section (rooting the tree with the Anomochlooideae instead of with the Joinvilleaceae). This chapter could be moved (much reduced) to the Materials and Methods.
Tables, figures and supplemental files are relevant and interesting. I have just a couple of comments here. First, regarding Table 2, I think there are problems with the numbers of positions used in the table. For instances (if I did it correctly), matrix Z, including coding and non-coding regions (and excluding gapped sites and positively selected sites) includes 71140 bp. Matrices H and R, including coding and non-coding regions (but not gapped or positively selected sites) have 44975 and 26307 bp, respectively (overall 71282). If I understood the method correctly, there are 142 bp missing in matrix Z (maybe I made a mistake with the numbers, if so I apologize to the authors...).

Phrasing and language edits (and other minor changes) to clarify the text:
Lines 41-42. Is it necessary to refer to these specific clades? especially considering that the results are in agreement with current classifications
Lines 73-75. The authors indicate the putative age of the fungal sclerotium, but make no reference to the ages of the phytoliths used by Prasad (2005, 2011). It is not very important, but perhaps both ages should be mentioned (or none)
Line 86. The authors write “520 tribes” instead of “52 tribes”
Line 141. Given that voucher specimens were deposited in one Herbarium only (DEK), it would be a good idea to use the complete name of the institution (at least the first time it is mentioned)
Lines 167-168. The authors indicate that “A published reference for each new plastome was obtained from a closely related grass species”. Perhaps the reference used could be added to table 1
Line 184: The authors should explain the meaning of dN and dS (it is never explained in the text)
Lines 185-186. The authors indicate that: “Each sequence was then tested for positive or purifying selection using the codon-based Z test of selection on default parameters”. A bibliographic reference on the method should be added here
Lines 180-202. Two tests of selection were conducted, but it is not clear to me why. Would it not be enough to use the second test (MEME)? Did they compare the results of both tests? This should be better explained
Line 231. “Data” should be written instead of “Datataset”
Lines 241-243. Please rephrase this third point
Lines 243-246. Please include figure numbers here
Line 291. Please write “by both” instead of “byboth”
Lines 531-534. Please simplify this sentence
Lines 674-675. “This pattern is indicative of character conflict between these data partitions that may be attributable to selection effects” A reference here would be convenient
Line 951. It should read “supported in” instead of “supportedin”
Line 958. It should read “this two genera lineage” instead of “this two genus lineage”
Line 1319. Please revise this sentence
Line 1656. Maximally?
Lines 1725-1726. Please revise this sentence
Table 2.- Caption. It should be “in each of these” instead of “in each these”
Figure 1. The caption should be extended to better explain the contents of the figure
Figure 7. Caption. Please revise “Panicoideae Pooideae”; what does “.st” mean?

·

Basic reporting

line 68 – what is meant by "dates of divergence"? Stem node or crown node?

Line 75 – there are some real questions about the validity of that "fungal sclerotium" and whether the floret is from a grass. If you choose to cite it, I would express some caution. Also most of the dating studies that you cite did not include that particular calibration point; if included it would make the grasses even older than the Indian phytoliths do.

Line 124 and Table S1. I assume the paper by Teisher et al. came out too recently for the plastomes to be included in this analysis, but it would be good to say that here.

Line 133 – It is good to have this identification error corrected in such a prominent way. Hopefully it will prevent future authors from being misled.

Line 260, discussion of positive selection – I'm surprised about the high number of positively selected codons in rpoC2. This protein includes a variable number of repeat sequences that are difficult to align (see papers by Nigel Barker) and almost certainly lead to alignment gaps in this analysis, in which case they would be largely omitted I think. Is it possible that the evidence for positive selection comes from the alignment problems? Likewise, aligning matK across many taxa usually requires gaps and can be difficult. Could the positive selection be an artifact here as well?

Line 277 and following – This paragraph is hard to read and some of the sentences are hard to follow. Can it be shortened? As an example of the confusion, the first sentence notes 242 clades with support ≥50 in tree X. 309 clades with support ≥50 in at least one tree. The next sentence (line 279) says "of these"… Are "these" the 309 clades? 82 have full support in all 14 trees but only 59 have full support in the 10 plastome trees. Is this an additional 59? Otherwise why is the number (59) fully supported in the plastome trees less than the number in all the trees? The rest of the paragraph is similarly confusing.

Line 291 – need space between "by" and "both"

Lines 299-300 – I don't understand why there is a standard deviation reported on both the maximum and minimum values. Shouldn't it be reported as range, mean, and standard deviation? Same issue on line 306.

Lines 300 and 303 – The number of lines for which support increases is listed as 11-46 and then later as 10-58.

Line 486 Gaoligongshania is misspelled.

Line 531-534 – I understand what the authors are trying to say here, but this sentence is somewhat awkward. How about just saying "The deepest split in the PACMAD clade varies among the trees." Or, "It is unclear whether Aristidoideae, Panicoideae, or both, is sister to the remaining PACMAD taxa."

Lines 534-539 – These sentences describe three possible topologies in 6 trees. These are supposedly the "complete plastome" trees, but there are four of these, not 6. If plastome coding and non-coding are included (which they seem to be since the paragraph includes tree R), then there are 10 trees. Something's missing.

Line 591 – "and" should not be capitalized.

Lines 631-633 – The numbers of trees here do not make sense. Triodiinae and Dactylocteniinae+Trichoneurinae form a clade in two trees, and Eleusininae and Datylo+Trichoneur form a clade in two other trees. There are 14 trees, so what about the other 10?

Line 640 – by "included" here do you mean "combined"?

Line 643-646 – The first authors to place the early diverging grasses correctly were Clark et al. in 1995; they should be cited here.

Lines 694 and following – This discussion of gaps is fine but I'm sure it applies more to some gaps than others. Is it possible to identify which specific genes or regions seem to be responsible for most of the conflicting signal? This section would be a lot more useful if it focused on specific sets of gaps in specific genes or plastome regions.

Line 826 – Aren't all bamboos polyploid? I don't think any diploids are known.

Line 951 – need a space between "supported" and "in"

Lines 964-965 – These sentences do not match the tree in Figure 5. In that tree P. aurea + propinqua is sister to the clade of P. nigra + sulphurea, and then P. edulis sister to the other four. Although the text here refers to five species only three are mentioned.

Line 1016 – Why is this topology likely an artifact? In some cases the authors give equal weight to clades supported by non-coding data, but here it is dismissed. Are there other non-molecular data that argue strongly for the other topology?

Lines 1193 and following – Much of the discussion here relies on unpublished data, in particular that of Saarela et al. (in review). While it is probably OK to cite some unpublished data for a single point, here the authors seem to be trying to describe a set of trees that the reader has no access to. I suggest removing most of the references to the unpublished Saarela paper, write the discussion as it pertains to the trees in this paper, and then insert a sentence somewhere to the effect that these results are extended and explored in more depth in the unpublished paper.

Line 1319 – remove open parentheses

Lines 1451-1453 – It is worth noting that Teisher et al. recovered all three possible topologies for the base of PACMAD, depending on assumptions. None was particularly well supported and they concluded that plastome data may be insufficient to resolve that particular set of relationships. The conflicting topologies of Teisher et al. are cited here but not until several lines below.

Line 1469 – The words "ambiguously aligned" are included here, whereas in most other parts of the manuscript there is no particular value judgement applied to the sites with gaps. There is also no documentation that the alignment is ambiguous – i.e., the fact that an alignment includes gaps does not make it ipso facto ambiguous.

Line 1498 and following – Teisher et al. (2017) should be cited in several places in this paragraph. They found the same topology as recovered here, including the strongly supported sister relationship of Arundinoideae and Micrairoideae.

Line 1579 - Cyperochloeae is misspelled. Insert "and" before Steyermarkochloeae.

Line 1580 – Lecomtella is misspelled.

Line 1581 – Lecomtelleae is misspelled.

Line 1690 – "include" should be "includes"

Experimental design

Strengths: The taxon sampling is comprehensive and the use of complete plastomes includes the maximum possible amount of sequence information. The Results and Discussion are placed in a broad context and consider all currently available data.
The data analysis is exhaustive and explores all relevant methods. Exploring the effect of including and excluding gapped sites and sites under positive selection is useful and gives confidence in the results.

Weaknesses: No serious weaknesses. My comments have to do with details and nuance and should be easy to address before publication. My first main concern is lack of evaluation of the various trees; they are described in detail but the reader is left on her own to decide which topologies to consider the most plausible. The Results section is a narrative description of the trees, duplicating what is shown in the figures. The Discussion section continues this approach but adds comparisons with previously published trees. These lengthy descriptions are presented as though all tree topologies have equal merit and as though there are no other biological characters involved. The Results section would be easier to read if it focused on the most robust results and the ones of most importance for classification or the evolution of major characters. The Discussion section would benefit from a few mentions of some characters that might be consistent with one or the other of the competing trees.

A second general point is that gapped sites are sometimes dismissed as being ambiguously aligned and sometimes they are accepted as being valid. Since some gaps are unambiguous (e.g. a well documented indel in ndhf) and some are ambiguous (e.g. runs of As or Ts in non-coding regions), the blanket statement seems not to be fully justified. Although I'm sure it is unintentional, the reader gets the sense (e.g. line 1469, line 1855-1857) that the gapped sites are being used to dismiss particular relationships on a somewhat ad hoc basis. For example, on lines 44-45, the statement is made that inclusion of gapped sites 'might introduce false signal". If the gapped sites are intrinsically unreliable, then they should be omitted throughout the paper. Conversely if the authors consider that they do provide some signal in some cases, then perhaps the gaps need to be divided into ambiguous and unambiguous sets, or at least consider where and when gapped sites might provide useful information. It is equally possible that exclusion of sites with obvious indels is misleading and reduces critical signal. I'd be more comfortable if the authors either a) took a more neutral tone, stating simply that in a few cases tree topologies are sensitive to inclusion or exclusion of sites with gaps, or b) evaluated the gapped sites themselves as to how helpful they are.

Validity of the findings

Nothing to add to comments above.

Additional comments

Nothing to add to comments above.

Reviewer 3 ·

Basic reporting

The paper is an important contribution and a first step to obtaining a tree-of-life of grasses using plastome data analysis. The authors have used their own data and data generated by previous authors to build an updated plastome-based phylogeny of Poaceae. The background and literature provided is sufficient, but sometimes it is not discussed appropriately. The English style, figures and tables are correct and the authors share the raw data. The paper is extremely long (101 pages manuscript, excluding tables and figures) and basically descriptive. The authors describe, too profusely and sometimes redundantly, the phylogenetic relationships among taxa within each subfamily, tribe and other taxonomic ranks in the Results and Discussion sections; these two sections could probably be re-organized and reduced to make a shorter paper keeping only essential novel information. Some parts of the paper and even subsections deal with aspects/taxa that have not been analysed in this paper and are therefore superfluous.

Experimental design

The research is primarily original as the authors have used methodological approaches (e. g., phylogenetic searches using 14 alternative data partitions and a basic test for detection of purifying/non-purifying selection within coding regions) not assayed before in grass plastomes. The methods have been described in detail and the performed analyses are overall correct. However, some procedures could be questionable. For example, 1) the alignments were not curated manually, at the risk of leaving some microstructural mutations/indels to be misaligned or deleting valuable information; it would, indeed, have a minimum effect at deep-level phylogenies, but perhaps not at shallow-level phylogenies, especially for very closely related taxa, though the authors solved partially the problem by discarding the gaps in some partition analyses; 2) the authors replaced stop codons with gaps in codon data if they were present; it could leave putative non-functional copies (e. g., pseudogenes) in the CDS data set. Despite the number of pseudogenes is low in the grass plastomes, it would be desirable to discard them from the CDS-based analysis. The authors have not commented the presence of potential pseudogenes in the data set (nor if some of them were present in one of the discarded IR); 3) the authors used a codon-based Z test to test for purifying vs positive selection in coding regions across the 250 grass plastomes using the default options in MEGA and Positive selection options in HyPhy (using MEME, mixed effects model of evolution to search for episodic selection at individual sites), and provide a Fig. 1 with number and proportion of codons under purifying/positive selection for each CDS and a Suppl. Table 1 with information on dN, dS, omega and p and test values for purifying/positive selection for each CDS; it is not clear, however, which codons of the genes show positive selection (those located in the more conserved 5’-end or those located in the more variable 3’-end of the genes?) and which values of the tests support one hypothesis over the other (in Table S1). Also, and most importantly, the authors have not tested if episodic positive selection is distributed across all grass lineages or, more likely, only across a subset of them (or even a few of them) in each case. To do it the authors should run a complementary approach to find selected branches (under positive selection) by pooling information over sites as indicated by Kosakovsky Pond et al. (2011) Mol Biol Evol 28: 3033–3043 (branch-site REL model of evolution).
The hypothesis testing is mainly based in a classification of grasses proposed by several of the current authors (Soreng et al. 2017) which, in turn, compiles molecular and morphological data from different researchers (see General comments to author).

Validity of the findings

The data is robust and statistically sound, though most of the novelty resides in the re-analysis of an enlarged data matrix using different data partitions (coding and non-coding regions, full plastome, selected genes) with and without gaps and positively selected codons, and the comparisons of the topologies and support of clades between partitions. Discussions with respect to previous phylogenetic works and classifications are exhaustively, but some of them are misleading or are not limited to supporting results (see General comments to author). The interpretation of the potential negative impact of positively selected sites should be re-evaluated with more precise data on distribution of sites across lineages, as well as that of the informative value of coding and non-coding regions with respect to the evolutionary depth of the groups under study.

Additional comments

The paper is an important contribution to the tree-of-life of grasses using plastome data analysis. It deals with an evolutionary systematic study of the grass family based on phylogenomic analysis of a large collection of plastome data (250 taxa, representing 180 genera and 44 tribes of Poaceae) generated by the authors and by previous researchers that have been jointly analysed for the first time using 14 different data partitions. The data, methodology and results are overall sound; however, the paper presents some flaws and misinterpretations that should be corrected and results that should be properly addressed. Additionally, the paper is extremely long, and parts of the Results and Discussion sections could probably be deleted or summarized to make a more readable paper.

I acknowledge the valuable efforts made by the authors to compile and analyse a large amount of current plastome data and to use it to help to resolve the phylogeny and systematics of Poaceae. Nonetheless, I am concerned about several issues that require throughout revision:

1.The paper is extremely long (101 pages manuscript, excluding tables and figures) and basically descriptive. The authors describe, too profusely and sometimes redundantly, the phylogenetic relationships among taxa within each subfamily, tribe and other taxonomic ranks of Poaceae in the Results and Discussion sections. These two sections could probably be re-organized and reduced to make a shorter paper keeping only essential novel information. Some parts of the paper deal with aspects/taxa that have not been analysed in this paper. For example, “Rooting the grass phylogenetic tree” subsection (of Discussion) is out of place here as the authors did not root the Poaceae tree with any close outgroup (but with basal Anomochlooideae lineages). In several instances, the authors comment grass subtribes for which plastomes have not been published yet. It is superfluous, as the current work is not intended to be a full revision.
2. The hypothesis testing is mainly based in a classification of grasses proposed by several of the current authors (Soreng et al. 2017) which, in turn, compiles molecular and morphological data from different researchers. It is appropriate to use it as a baseline hypothesis and to compare the new results to it; however, the authors should give the credits to the researchers that contributed most to the phylogeny/classification of each group, trying to avoid undermining or skewed statements. For example, the authors indicate that “Danthonioideae includes a single tribe comprising 18 genera …. (Soreng et al., 2017)” and that “the plastome trees” (-based on 7 species representing 6 genera) “are better supported than trees based on a few plastid regions (Linder et al. 2010)”. However, systematic circumscription of Danthonioideae is mostly based in the work of Linder et al. (2010), who recognized 17 genera after their comprehensive molecular and morphological study of 281 danthonioid species, and the additional contribution of Teisher et al. (2017) with 1 genus. It is not surprising to know that the 7 species plastome trees of Danthonioideae are better supported than the >200 species plastid trees of Linder et al. (2010) but the two studies and their respective sampling sizes are not comparable.
3. The authors used a codon-based Z test to test for purifying vs positive selection in coding regions across the 250 grass plastomes using the default options in MEGA and Positive selection options in HyPhy (using MEME, mixed effects model of evolution to search for episodic selection at individual sites), and provide a Fig. 1 with number and proportion of codons under purifying/positive selection for each CDS and a Suppl. Table 1 with information on dN, dS, omega and p and test values for purifying/positive selection for each CDS. It is an interesting approach though it is not clear, however, which codons of the genes show positive selection (those located in the more conserved 5’-end or those located in the more variable 3’-end of the genes?) and which values of the tests support one hypothesis over the other (in Table S1). Also, and most importantly, the authors have not tested if episodic positive selection is distributed across all grass lineages or, more likely, only across a subset of them in each case. It would be advisable to conduct a complementary approach in order to find selected branches (under positive selection) by pooling information over sites as indicated by Kosakovsky Pond et al. (2011) Mol Biol Evol 28: 3033–3043 (branch-site REL model of evolution). It is highly relevant because some of the conclusions drawn by the authors about the potential effect that genes containing positively selected codons can have on phylogenetic disturbance (or “systematic error”) should be carefully checked.
4. The authors have built a grass phylogeny showing different evolutionary depths (from subfamily (deep phylogeny) to species/variety (shallow phylogeny) levels and have compared the levels of resolution and support of each of the 14 data partitions across the main clades. However, it would be worth to know which data partition (or even subpartition) is most adequate for solid reconstruction of the deep and shallow branches of the tree clades. Additionally, they indicate that widely used loci in grass systematics, such as ndhF, matK, rpoC2 and rbcL, particularly prone to positive selection in some codons, could cause phylogenetic artifacts. I am not convinced about this conclusion. These genes (or parts of them) are more variable than other plastid genes, and therefore more useful to resolve deep vs shallow phylogenies, as demonstrated in early studies of plastid phylogenies of angiosperms. All of them encode functional proteins, and episodic positive selection in some codons may occur only in some lineages. At least two of them (rbcL, matK) have been selected as barcoding molecules for plants (ToL project). The authors should make more solid arguments about the phylogenetic value/failure of these genes before recommending caution about them. Moreover, the suggestion of using the highly conserved psaA and psaB genes for phylogenetic reconstruction is not very useful for shallow phylogenies of grasses; they may content very few positively selected codons but do not contain enough phylogenetic signal for recently evolved lineages.
5. The core Pooideae clade should not include Brachypodieae. Several evidences support it. The sister but non-inclusive relationship of Brachypodium to the core pooid clade [Triticodae (Triticeae+Bromeae)/Poodae (former Poeae+Aveneae; now Poeae s. l.)], originally proposed by Davis and Soreng (1993), was abandoned in favor of the inclusion of Brachypodium within the ‘core pooids’, a non-taxonomic but independently evolved natural group, in some recent analyses (Davis & Soreng, 2007; Saarela et al., 2015; Soreng et al., 2015, 2017). However, recent studies (Minaya et al. 2015; pooids, b-amylase; Sancho et al., 2017, New Phytol. (accepted paper, phylogenomics of Brachypodium and grass plastomes) support the sister relationship proposed by Davis and Soreng as well as divergence times intermediate between those of the basal ancestral pooids and the recently evolved core pooids (Sancho et al. 2017). Additionally, pairwise plastome genetic and patristic distances have further confirmed that Brachypodium is closer to some basal pooid lineages than to the core pooid lineages (Sancho et al., 2017), corroborating similar results based on nuclear single copy genes (Minaya et al., 2015) and functional genomic studies of regulation of vernalization and flowering time genes (Fjellheim et al. 2014; Front. Plant Sci. 5: 431;
Woods et al. 2016, Plant Physiol. 170:2124-2135). Brachypodium is a highly isolated lineage, and the large length of its stem branch (compared to the short lengths of its crown branches) has been recovered in all phylogenetic studies conducted with representative species of this genus and other Pooideae lineages (see, for example, Minaya et al. 2015; Catalan et al. 2016 (not 2015); Sancho et al. 2017). It would be advisable to recognize that Brachypodieae is not part of the core Pooideae from the very beginning; the authors could explain better their current results based on this hypothesis.
6. The authors have misinterpreted the results of Christin et al. (2007, 2008) about convergent evolution of C4 photosynthetic pathway genes. The sentence (ls. 758-760) “For example, in grasses in which photosynthetic genes, such as rbcL or PEPC, converge under selection for C4 photosynthesis, misleading phylogenies can result (Christin et al., 2007, 2008)" is incorrect. Convergent evolution caused by selection resulting in (false) monophyly of C4 grasses was found in analysis of some coding positions of the PEPC gene (Christin et al. 2007), whereas polyphyletic origin of C4 grass lineages was recovered in analyses of non-coding positions of the same PEPC gene (Christin et al. 2007) as well as in plastid rbcL and ndhF genes (Christin et al. 2008). Christin et al. never indicated that the plastid rbcL gene was under convergent selection; by contrast, they used this gene and the plastid ndhF gene to construct a reliable phylogeny to estimate the divergence times of the polyphyletic C4 grass lineages.

Additional points:

7. The alignments were not curated manually, at the risk of leaving some microstructural mutations/indels to be misaligned or deleting valuable information; it would, indeed, have a minimum effect at deep-level phylogenies, but perhaps not at shallow-level phylogenies, especially for very closely related taxa, though the authors solved partially the problem by discarding the gaps in some partition analyses.
8. The authors replaced stop codons with gaps in codon data if they were present; it could leave putative non-functional copies (e. g., pseudogenes) in the CDS data set. Despite the number of pseudogenes is low in the grass plastomes, it would be desirable to discard them from the CDS-based analysis. The authors have not commented the presence of potential pseudogenes in the data set (nor if some of them were present in one of the discarded IR).
9. Despite their critics about genes showing positively selected codons the authors chose as their reference tree, tree X, based on plastome data including positively selected codons.
10. l. 250. ‘Unsupported’ should not be used for clade support < 50%, you could use very low or very weak support instead.
11. ls. 304-306. What is the sentence for?
12. l. 663. Please give details on number of clades and percentages.
13. ls. 667-668. Please identify the 30 clades. Do they correspond to shallow clades?
14. ls. 674-675. The argument is not convincing; character conflict does not necessarily to be connected with selection effect. The authors should test this hypothesis.
15. l. 821, 836-841. Bambusoideae. Most of the Discussion on bamboos is based on previous plastid-based phylogenetic studies, despite the fact that many bamboos are polyploids. What is the conclusion here? Can the authors identify hybrid/allopolyploid clades or introgression events that could explain the conflicting topologies obtained from plastomes and from nuclear data?
16. ls. 964-966. Confusing sentence, the authors have not described properly the relationships among the 5 Phyllostachys species.
17. ls. 1003-1008. Unnecessary paragraph. The current study could not resolve it as Neohouzeaua has not been included in the study.
18. ls. 1053-1054. The pivotal paper that proposed Brachypodium distachyon as model system for grasses was that of IBI (International Brachypodium Initiative) or Vogel et al. 2010. Nature 463: 763-768.
19. 1066-1069. A sister relationship of Diarrhena and Brachypodium was recovered in the beta-amylase tree of Minaya et al. (2015) though some sequences (Diarrhena, B. distachyon clone 2-3) could be recombinant. The authors cite this reference in the manuscript but do not comment these results.
20. ls. 1085-1098. See comments above about phylogenetic studies of Brachypodium (Minaya et al. 2015; Catalan et al. 2016 (book chapter), and the accepted work by Sancho et al. 2017, that could be discussed by the authors.
21. ls. 1102-1109. I recommend the authors to read the recent work by Sancho R, Cantalapiedra CP, López-Álvarez D, Gordon SP, Vogel JP, Catalan P, Contreras-Moreira B. 2017. Comparative plastome genomics and phylogenomics of Brachypodium: flowering time signatures, introgression and recombination in recently diverged ecotypes. New Phytologist (in press) for a robust dating analysis of the Brachypodium lineages within the grass phylogeny framework based on plastome analysis and nesting dated approaches. The paper will be published as early view soon.
22. ls. 1197-1198 and more. Please indicate if the work by Saarela et al. (in review) is already published.
23. ls. 1205-1207. Pimentel et al. (2017) found a sister relationship of Lagurus to Aveninae-Koeleriinae in their 5-genes plastid tree. The authors should comment it. This paper [Pimentel M, Escudero M, Sahuquillo E, Minaya MA, Catalán P. 2017. Diversification rates and chromosome evolution in the temperate grasses (Pooideae) are associated with major environmental changes in the Oligocene-Miocene. PeerJ] has been accepted for publication and will be published soon.
24. ls. 1236-1243. Ammophila has not been included in this study. These sentences are unnecesary here.
25. 1245-1247. Pimentel et al. (2017) in their 5-genes plastid phylogeny do not recover a sister relationship of Anthoxanthiinae to Agrostidinae+Brizinae but that of Anthoxanthinae to Aveninae-Koeleriinae-Lagurus. These authors have a larger sampling of Agrostidinae, and Aveninae-Koeleriinae than the one presented here. The authors should discuss it and be cautious about their conclusion.
26. ls. 1275-1278. Minaya et al. (2015) included Avenula bromoides (Gouan) H. Scholz in their study (not A. hookeri). It is not surprising to know that the plastome tree does not agree with the ITS (nuclear)+plastid tree of Minaya et al. (2015) for these particular taxa, considering the high reticulation of the groups involved.
27. 1278-1279. The authors do not explain the conflicts of their plastome tree with those of the b-amylase tree of Minaya et al. (2015). They should comment them. Again, it would not be surprising to find differences between the nuclear beta-amylase tree and a plastome tree in highly reticulate groups. The authors do not extract conclusions about the observed differences. Moreover, Minaya et al. (2015) detected incongruent resolutions for some beta-amylase sequences, caused by recombination or selection, and more specifically they explained the cases of Avenula bromoides, Desmazeria rigida, Deschampsia antarctica, Alopecuros arundinaceus, Colpodium drakensbergense, Ammophila arenaria, Vulpia alopecuros, and Festuca ovina. It looks as if the authors have not paid attention to the paper by Minaya et al. (2015), in which they throughly investigated the potential origins of phylogenetic incongruence in cloned copies of the single copy gene beta-amylase.
28. 1279-1282. The sentence is superfluous; it is not currently supported by the data.
29. ls. 1288-1289. This relationship is only supported by 8-9 trees and, as stated by the authors, they haven't sampled other representatives of Airinae (apart from H. hookeri), Holcineae (apart from H. lanatus) and Deschampsia s. s. (apart from D. antarctica). In the 5-genes plastid phylogeny of Pimentel et al. (2017) those relationships are not well supported.
30. ls. 1299-1305. The 5-genes plastid phylogeny of Pimentel et al. (2017) (including matK) does not recover a strong support for the Cynosurinae+Dactylidinae+Parapholiinae+Loliinae clade. A tree based solely in matK might not be an improvement to the resolution of the phylogeny of the group.
31. ls. 1306-1307. 98% in tree X (Fig. 4) for the branch showing the sister relationship of Dactylidinae/Cynorusinae+Parapholiinae to Loliinae. Taxon sampling has increased in Dactylidinae, Cynosurinae and Parapholiinae but not much in other close groups (Airinae, Holcinae, Deschampsia s.s.).
32. ls. 1318-1319. Incomplete sentence.
33. ls. 1467-1469. The authors recognize that their plastome gapped regions could be ambiguously aligned. It could be avoided in part through manual alignment curation.
34. ls. 1478-1479. “lack of support”. Replace with very low support.
35. l. 1511. The authors recognize here the potential influence of nuclear genes in combined nuclear+plastid topologies and their contrasting resolution with respect to plastid or plastome topologies alone, but they haven't done it with respect to Minaya et al. (2017) analyses (see comments above).
36. ls. 1634-1638. Please compare levels of support with sampling sizes.
37. ls. 1653-1656. Confusing sentence. It is not clear which studies show stronger support than others and which studies were considerably less sampled than others.
38. ls. 1656-1661. Please indicate if the Washburn et al. (2015) tree is their only-plastid tree, -which would be congruent with the plastome tree-, and if conflict with the Vicentini et al. (2008) tree is because the latter is a nuclear-based tree (as indicated by Washburn et al. 2015).
39. ls. 1683-1687. Please indicate the ploidy level of W. capillipes. Bidirectional crosses are common in grass allopolyploids (e. g. Brachypodium hybridum, Lopez-Alvarez et al. 2012, 2017; Catalan et al. 2016) and allopolyploidy and bidirectional origin could explain the coexistence of two plastid types within the same taxon (see discussions in Catalan et al. 2016 (Phylogeny and evolution of the genus Brachypodium, book chapter); Lopez-Alvarez et al. 2017, Annals of Botany 119: 545-561.).
40. ls. 1694-1698. The authors should discuss their results regarding Setaria and Paspalidium with respecto to the broadly sampled study of both genera and close allies conducted by Kellogg et al. (2009) using plastid ndhF data. Please specify the two plastome trees that support the monophyly of Setaria.
41. ls. 1816-1817. Speculative assessment. Denser plastome sampling could also reduce the resolution or the support of some evolutionary relationships.
42. ls. 1836-1839. The analysis of perennial Brachypodium spp plastomes is currently underway (Sancho et al. unpub. data). Nonetheless, resolution of Brachypodieae with respect to Diarrheneae and the core Pooideae (Poeae+Bromeae+Triticeae) do not depend on it. Brachypodium is a strongly supported monophyletic lineage in all evolutionary analyses.
43. ls. 2023-2025. The correct reference is as follows: Catalán P, López-Alvarez D, Díaz-Pérez A, Sancho R, López-Herranz ML. 2016. Phylogeny and evolution of the genus Brachypodium. In Vogel J (ed.). Genetics and genomics of Brachypodium. pp. 9-38. Series Plant Genetics and Genomics: Crops Models. Springer. New York.

---

## Round 0.2 · Minor Revisions

Dear Jeff and Melvin,

The two reviewers agree that the authors have made a substantial effort to address their concerns of the previous version of the manuscript. I agree this version of the manuscript is much improved and easier to read. There are still many minor suggestions from the two reviewers. I hope you are able to include reviewers' suggestions and resubmit the paper.

Sincerely,

Marcial.

·

Basic reporting

The authors have made a substantial and conscientious effort to address the concerns of the previous reviewers. This version of the manuscript is much improved and appreciably more readable.

The following comments all note trivial editorial changes:

Line 147 – the cited reference here is incorrect. We used seed from that same accession (not really equivalent to the same individual) in Estep et al. 2014, which I think is cited elsewhere.

Lines 202, 287 – "predominate" (verb) should be "predominant" (adjective)

Line 327 – I'm having trouble understanding the syntax here. Is a word missing, or an extraneous phrase included?

Lines 319-363 – This section is a narrative report of the data in Table 3 and is very hard to read. I appreciate the efforts to provide a summary, but it still feels long. How about starting the paragraph at line 319 as "Excluding positively selected codons had relatively little effect on support values overall, with average support differences ranging from 1 to 4%. The largest effect of positively selected sites is in comparison B vs. D. This is not surprising since these are three-gene data sets, with all genes known to have positively selected sites."
The next paragraph could be: "Exclusion of gapped sites also had relatively little effect on average support values. … (two sentences similar to above).

Line 573 – "Panicininae" should be "Panicinae"

Lines 649-651 – There's an arithmetic problem here: 23 clades in the three gene trees differ from those in the plastome coding trees; 13 are only weakly supported, but the remaining 9 are moderately supported. 9 + 13 = 22, so one is missing.

Lines 789-791 – Is the rate of evolution in Olyreae accelerated or is it retarded in the woody bamboos? Unless there's been a formal test that Olyreae have been speeding up relative to some background rate, it might be better just to state that Olyreae plastomes mutate faster than those in the woody bamboos, and leave open the question of whether it's a speed up in one or a slow down in the other.

Line 864 – move parentheses in Attigala et al. (2016)

Section on Poeae, lines 1056 and following – in a number of places in this section, species are listed as "newly sampled here". I'm not sure what this means, since I think a number of these taxa are in other studies, just not with complete plastomes. Also, these "newly sampled" statements appear to be mostly in the Pooideae rather than elsewhere in the manuscript. I'd suggest either deleting them or making clear why they are particularly important.

Line 1080 – delete "or not"

Line 1248 – I think chloroplast capture is a form of introgression, so these two terms are redundant.

Line 1361 – add a period after "sampled"

Line 1478 – There is a fourth explanation, which is a labeling mix-up or contamination somewhere in the extraction or sequencing process. The solution (which I’m not recommending for the current paper!) would be to re-extract and sequence both specimens.

Line 1507 – "unisetis" should be "unisetus"

Line 1554-1555 – There is a problem with the sentence beginning "Twenty one five subtribes …." Probably a typographical error somewhere.

Line 1607 - I think I know what you mean by "all plastid and plastome trees", but plastid = plastome. Presumably this is intended to compare few-gene vs. complete plastome trees?

Lines 1664-1665 – I'd be cautious about any biological inferences about Whiteochloa until the possibility of technical error can be ruled out.

Experimental design

Meets all journal standards. The analyses are careful and detailed.

Validity of the findings

Meets all journal standards. The conclusions are well supported.

Additional comments

This is a very useful contribution to the literature and I'm looking forward to seeing it in print.

Reviewer 3 ·

Basic reporting

see General comments

Experimental design

see General comments

Validity of the findings

see General comments

Additional comments

This is a resubmission of a manuscript I reviewed before. The authors have satisfactorily addressed most of the issues raised in my first review and have considerably shortened the length of the paper, making it a more readable work.

I have few additional issues about the new version that should be considered as they could help to improving/clarifying the current knowledge on the evolutionary history of some particular grass lineages.

Introduction:

Ls. 78-86: “Recent grass plastome sequences have been variously published …”: Please cite the recent publication by Sancho et al. 2017 (Comparative plastome genomics and phylogenomics of Brachypodium: flowering time signatures, introgression and recombination in recently diverged ecotypes. New Phytologist, early view, doi: 10.1111/nph.14926). This work includes a family-wide plastome-based phylogenomic dating analysis of the main lineages of grasses.

Discussion

BOP clade:

Ls. 766-768: “However, in a recent plastid study Oryzoideae and Pooideae were recovered as sister taxa, although with uneven sampling throughout the family (Pimentel et al., 2017)”: This sentence seems to be out of place here and could be deleted. The study of Pimentel et al (2017) was exclusively focused on Pooideae; sampling outside the subfamily was very scarce and the few non-pooid representatives incorporated into the analyses were used as outgroups.

Pooideae:

Ls. 972-980: “Tribes Brachypodieae, Diarrheneae, Bromeae, Poeae and Triticeae form a maximally supported clade …”: Please cite here the recent plastome-based study of grasses conducted by Sancho et al. (2017). It would be worth mentioning here that these authors found intermediate consecutively diverging positions of Diarrheneae and Brachypodieae between the basal pooid lineages and the recently evolved lineages of the 'core pooids' clade and that their study supports the evolutionary circumscription of the ‘core pooid’ clade to the recently evolved Triticeae+Bromeae/Poeae lineage, as initially suggested by David & Soreng (1993). It could be also mentioned that Sancho et al. (2017) demonstrated through raw pariwise genetic distances and patristic genetic distances (on their ML tree) that Brachypodieae are not closely related to core pooids (Triticeae, Bromeae and Poeae), in fact they are as close to Triticeae-Bromeae as to the basal pooid lineages and even less close to Poeae than to the studied basal pooids.

Ls. 1009-1011: “Evidence does not support an older age for the Brachypodium crown clade than for the Poeae + Triticeae crown clade (Catalán et al.,2012)”: Please mention the recent plastome-based nested dating analysis of the grass family and Brachypodium presented by Sancho et al. (2017). The authors estimated the age of the Brachypodium crown node at 10.1 Ma.

Ls. 1011_1013: “Plastomes of the other species of Brachypodium (Catalán et al., 2012, 2016a, 2016b), once sampled, might break up the long branch and help clarify relationships among Brachypodieae, Diarrheneae and Bromeae–Poeae–Triticeae”: This statement is incorrect. Plastomes of other species of Brachypodium (e. g. the early diverging B. stacei) do not affect the evolutionary position of Brachypodium within the plastome grass tree (cf. Sancho et al. 2017). Relationships of all these tribal taxa have been clearly shown in this study.

Poeae

Ls. 1057-1059: Please cite the plastome-based study of Sancho et al. (2017).

Ls 1070-1071: “In a recent five region plastid study, however, Anthoxanthinae is strongly supported as sister to Aveninae (Pimentel et al., 2017)”: This statement is incorrect. In the study of Pimentel et al (2017) the Anthoxanthinae lineage is sister to a ((Aveninae/Koeleriinae), Lagurus) clade. A similar relationship was also found by Quintanar et al. (2007).

Future directions

Ls 1649-1651: “(4) sample perennial species of Brachypodium (Brachypodieae), which may help resolve problematic relationships among Brachypodieae, Diarrheneae and Poeae + Bromeae + Triticeae in subfamily Pooideae”: This statement is out of place here. Sancho et al. (2017) have clarified the evolutionary relationships of Brachypodieae and the other tribes in their plastome-based study sampling the three annual species of Brachypodium and dating the grass plastome tree. Inclusion of perennial species of Brachypodium will not change the relationships (most Brachypodium perennial species are of more recent origin than the annuals, e. g. those of the core perennial clade (see plastid and nuclear trees in Catalan et al. 2016), and only one short-rhizomatous perennial (B. mexicanum) is approximately as old as the annuals or of intermediate age between the annuals and the core perennial clade (B. retusum) (cf. Catalan et al. 2012, 2016). New plastome data from perennial Brachypodium taxa corroborate these findings (Sancho et al. unp. data).

---

## Round 0.3 · accepted · Accept

Dear Jeff and Melvin,

You have added reviewers suggestions. As result your study has been improved. I am glad to inform you that your study is now suitable for publication in PeerJ.

Congratulations!

Cheers,

Marcial.